# Overflow metabolism originates from growth optimization and cell heterogeneity

Xin Wang*

School of Physics, Sun Yat-sen University, Guangzhou, China

## eLife Assessment

This **valuable** study tackles the well-established overflow metabolism issue by applying a coarse-grained metabolic flux model to predict how individual cells execute various energy strategies, such as respiration versus fermentation. The model's population average is **convincing** enough to align with experimental observations on overflow metabolism. The potential source of metabolic or proteomic heterogeneity of individual cells remains an open question to be studied. How individual cells adjust their metabolic strategies also requires future study of the underlying mechanisms. Overall, this work provides a key aspect on cell-to-cell variability on general metabolic response.

***For correspondence:**
wangxin36@mail.sysu.edu.cn

**Competing interest:** The author declares that no competing interests exist.

**Abstract** A classic problem in metabolism is that fast-proliferating cells use seemingly wasteful fermentation for energy biogenesis in the presence of sufficient oxygen. This counterintuitive phenomenon, known as overflow metabolism or the Warburg effect, is universal across various organisms. Despite extensive research, its origin and function remain unclear. Here, we show that overflow metabolism can be understood through growth optimization combined with cell heterogeneity. A model of optimal protein allocation, coupled with heterogeneity in enzyme catalytic rates among cells, quantitatively explains why and how cells choose between respiration and fermentation under different nutrient conditions. Our model quantitatively illustrates the growth rate dependence of fermentation flux and enzyme allocation under various perturbations and is fully validated by experimental results in *Escherichia coli*. Our work provides a quantitative explanation for the Crabtree effect in yeast and the Warburg effect in cancer cells and can be broadly used to address heterogeneity-related challenges in metabolism.

## Introduction

A prominent feature of cancer metabolism is that tumor cells excrete large quantities of fermentation products in the presence of sufficient oxygen (*Hanahan and Weinberg, 2011*; *Liberti and Locasale, 2016*; *Vander Heiden et al., 2009*). This process, discovered by Otto Warburg in the 1920s (*Warburg, 1924*) and known as the Warburg effect, aerobic glycolysis, or overflow metabolism (*Basan et al., 2015*; *Hanahan and Weinberg, 2011*; *Liberti and Locasale, 2016*; *Vander Heiden et al., 2009*), is ubiquitous among fast-proliferating cells across a broad spectrum of organisms (*Vander Heiden et al., 2009*), ranging from bacteria (*Basan et al., 2015*; *Holms, 1996*; *Meyer et al., 1984*; *Nanchen et al., 2006*; *Neidhardt et al., 1990*) and fungi (*De Deken, 1966*) to mammalian cells (*Hanahan and Weinberg, 2011*; *Liberti and Locasale, 2016*; *Vander Heiden et al., 2009*). For microbes, cells use standard respiration when nutrients are scarce, while they use the counterintuitive aerobic glycolysis when nutrients are adequate, just analogous to normal tissues and cancer cells, respectively (*Vander Heiden et al., 2009*).

Over the past century, and particularly through extensive studies in the last two decades (*Liberti and Locasale, 2016*), various rationales for overflow metabolism have been proposed (*Basan et al., 2015*; *Chen and Nielsen, 2019*; *Majewski and Domach, 1990*; *Molenaar et al., 2009*; *Niebel et al., 2019*; *Peebo et al., 2015*; *Pfeiffer et al., 2001*; *Shlomi et al., 2011*; *Vander Heiden et al., 2009*; *Varma and Palsson, 1994*; *Vazquez et al., 2010*; *Vazquez and Oltvai, 2016*; *Zhuang et al., 2011*). Notably, *Basan et al., 2015* provided a systematic characterization of this process, including various types of experimental perturbations. Currently, prevalent explanations (*Basan et al., 2015*; *Chen and Nielsen, 2019*) hold that overflow metabolism arises from the proteome efficiency in fermentation being consistently higher than that in respiration. However, recent studies have shown that the measured proteome efficiency in respiration is actually higher than in fermentation for many yeast and cancer cells (*Shen et al., 2024*), even though these cells generate fermentation products through aerobic glycolysis. This finding (*Shen et al., 2024*) apparently contradicts the prevalent explanations (*Basan et al., 2015*; *Chen and Nielsen, 2019*). Furthermore, most explanations (*Basan et al., 2015*; *Chen and Nielsen, 2019*; *Majewski and Domach, 1990*; *Shlomi et al., 2011*; *Varma and Palsson, 1994*; *Vazquez et al., 2010*; *Vazquez and Oltvai, 2016*; *Zhuang et al., 2011*) rely on the assumption that cells optimize their growth rate for a given rate of carbon influx (i.e. nutrient uptake rate) under each nutrient condition (or its equivalents). However, this assumption remains open to further scrutiny, as the given factors in a nutrient condition are the identities and concentrations of the carbon sources (*Molenaar et al., 2009*; *Scott et al., 2010*; *Wang et al., 2019*), rather than the carbon influx. Therefore, the origin and function of overflow metabolism still remain unclear (*DeBerardinis and Chandel, 2020*; *Hanahan and Weinberg, 2011*; *Liberti and Locasale, 2016*; *Vander Heiden et al., 2009*).

Why have microbes and cancer cells evolved to possess the seemingly wasteful strategy of aerobic glycolysis? For unicellular organisms, there is evolutionary pressure (*Vander Heiden et al., 2009*) to optimize cellular resources for rapid growth (*Dekel and Alon, 2005*; *Edwards et al., 2001*; *Hui et al., 2015*; *Li et al., 2018*; *Scott et al., 2010*; *Towbin et al., 2017*; *Wang et al., 2019*; *You et al., 2013*). In particular, it has been shown that cells allocate protein resources for optimal growth (*Hui*

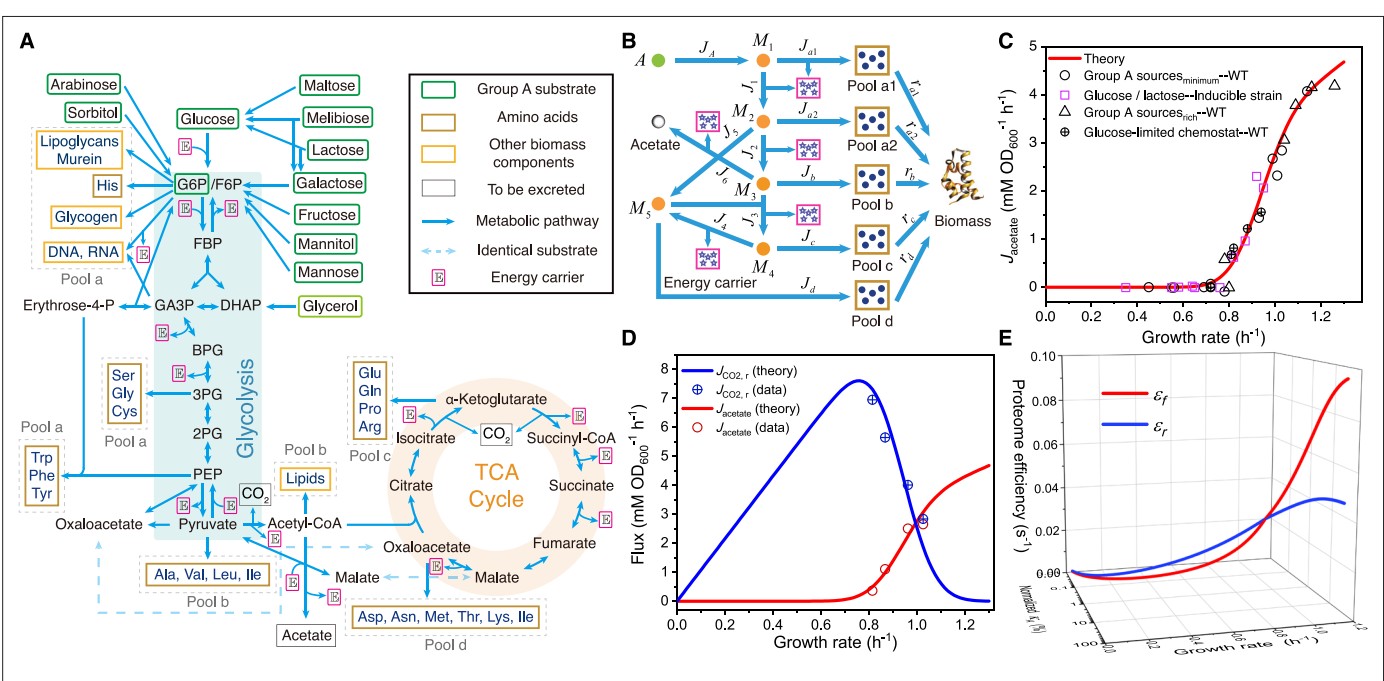

**Figure 1.** Model and results of overflow metabolism in *E. coli*. (**A**) The central metabolic network of carbon source utilization. The Group A carbon sources (*Wang et al., 2019*) are labeled with green squares. (**B**) Coarse-grained model for Group A carbon source utilization. (**C**) Model predictions (see *Equations S47 and S160*) and experimental results (*Basan et al., 2015*; *Holms, 1996*) of overflow metabolism, covering the data for all the Group A carbon sources shown in (**A**). (**D**) Growth rate dependence of respiration and fermentation fluxes (see *Equations S47 and S160*). (**E**) The proteome efficiencies for energy biogenesis in the respiration and fermentation pathways vary with growth rate as functions of the nutrient quality of a Group A carbon source (see *Equations S31 and S36*). See Appendices 9 and 11 for model parameter settings and experimental data sources (*Basan et al., 2015*; *Holms, 1996*; *Hui et al., 2015*) for *Figures 1–4* of *E. coli*.

et al., 2015; Scott et al., 2010; Wang et al., 2019; You et al., 2013), and the most efficient protein allocation corresponds to elementary flux mode (Müller et al., 2014; Wortel et al., 2014). For cancer cells, disrupting the growth control system and evading immune destruction from the host are prominent hallmarks of their survival (Hanahan and Weinberg, 2011), which in certain ways mimic the evolutionary pressure on microbes to optimize cell growth rate. In this study, we apply the optimal growth principle of microbes, which also roughly holds for cancer cells, to a heterogeneous framework to address the puzzle of aerobic glycolysis. We use *Escherichia coli* as a typical example to show that overflow metabolism can be understood from optimal protein allocation combined with heterogeneity in enzyme catalytic rates. The optimal growth strategy varies between respiration and fermentation depending on the concentration and type of the nutrient, and the combination with cell heterogeneity results in the standard picture (Basan et al., 2015; Holms, 1996; Meyer et al., 1984; Nanchen et al., 2006; van Hoek et al., 1998) of overflow metabolism. Our model quantitatively illustrates the growth rate dependence of fermentation/respiration flux and enzyme allocation under various types of perturbations in *E. coli*. Furthermore, it provides a quantitative explanation for the data on the Crabtree effect in yeast and the Warburg effect in cancer cells (Bartman et al., 2023; Shen et al., 2024).

## Results

### Coarse-grained model

Based on the topology of the metabolic network (Neidhardt et al., 1990; Nelson and Cox, 2008) (see *Figure 1A*), we classify the carbon sources that enter from the upper part of glycolysis into Group A (Wang et al., 2019) and the precursors of biomass components (such as amino acids) into five pools. Specifically, each pool is designated according to its entry point (see *Figure 1A* and Appendix 2.2 for details): a1 (entry point: G6P/F6P), a2 (entry point: GA3P/3PG/PEP), b (entry point: pyruvate/acetyl-CoA), c (entry point: $\alpha$-ketoglutarate), and d (entry point: oxaloacetate). Pools a1 and a2 are also combined as Pool a due to the joint synthesis of precursors. Then, the metabolic network for Group A carbon source utilization (see *Figure 1A*) can be coarse-grained into a model shown in *Figure 1B* (see Appendix 3.1 for details), where node $A$ represents an arbitrary carbon source of Group A. Evidently, *Figure 1B* is topologically identical to *Figure 1A*. Each coarse-grained arrow in *Figure 1B* represents a stoichiometric flux $J_i$, which delivers carbon flux and may be accompanied by energy consumption or biogenesis (e.g. $J_1$, $J_{a1}$; see *Figure 1A–B* and *Appendix 1—figure 1A*).

In fact, the stoichiometric flux $J_i$ scales with the cell population. For comparison with experiments, we define the normalized flux $J_i^{(\text{N})} \equiv J_i \cdot m_0 / M_{\text{carbon}}$, which can be regarded as the flux per unit of biomass (the superscript '(N)' stands for normalized; see Appendix 2.3–2.4 for details). Here, $M_{\text{carbon}}$ represents the carbon mass of the cell population, and $m_0$ is the weighted average carbon mass of metabolite molecules at the entry of precursor pools (see *Equation S17*). Then, the cell growth rate $\lambda$ can be represented by the total outflow of the normalized fluxes: $\lambda = \sum_i^{a1,a2,b,c,d} J_i^{(\text{N})}$ (see Appendix 2.4). The normalized fluxes of respiration and fermentation are $J_r^{(\text{N})} \equiv J_4^{(\text{N})}$ and $J_f^{(\text{N})} \equiv J_6^{(\text{N})}$, respectively (see *Figure 1A and B*). In practice, each $J_i^{(\text{N})}$ is characterized by two quantities: the proteomic mass fraction $\phi_i$ of the enzyme dedicated to carrying the flux and the substrate quality $\kappa_i$, such that $J_i^{(\text{N})} = \phi_i \cdot \kappa_i$. We take the Michaelis-Menten form for the enzyme kinetics (Nelson and Cox, 2008), and then $\kappa_i \equiv k_i \cdot \frac{[S_i]}{[S_i]+K_i}$ (see *Equation S12* and Appendix 2.4 for details), where $[S_i]$ is the concentration of substrate $S_i$, and $K_i$ is the Michaelis constant. For each intermediate node and reaction along the pathway (e.g. node $M_1$ in $J_{a1}$), the substrate quality $\kappa_i$ can be approximated as a constant (see Appendix 2.5): $\kappa_i \equiv k_i \cdot \frac{[S_i]}{[S_i]+K_i} \approx k_i$, where $[S_i] \geq K_i$ generally holds true in bacteria (Bennett et al., 2009; Park et al., 2016). However, the nutrient quality $\kappa_A$ is a variable that depends on the nutrient type and concentration of a Group A carbon source (see *Equation S27*).

Generally, there are three independent fates for a Group A carbon source in the metabolic network (Chen and Nielsen, 2019): fermentation, respiration, and biomass generation (see *Appendix 1—figure 1C-E*). Each draws a distinct proteome fraction of $\phi_f$, $\phi_r$ and $\phi_{\text{BM}}$, with no overlap between them (see Appendix 3.1). The net effect of the first two fates is energy biogenesis, while the last one generates precursors for biomass, accompanied by energy biogenesis. By applying the proteomic

constraint that there is a maximum fraction, $\phi_{\max}$, for proteome allocation: $\phi_{\max} \approx 0.48$ (*Scott et al., 2010*), we have:

$$\phi_f + \phi_r + \phi_{\mathrm{BM}} = \phi_{\max}. \tag{1}$$

In fact, *Equation 1* is equivalent to $\phi_{\mathrm{R}} + \phi_A + \sum_{j=1}^{6} \phi_j + \sum_{i}^{a1,a2,b,c,d} \phi_i = \phi_{\max}$ (see Appendix 3.1 for derivation details), where $\phi_{\mathrm{R}}$ and $\phi_A$ represent the proteomic mass fractions of the active ribosome-affiliated proteins and the cargo proteins responsible for the uptake of the Group A carbon source, respectively. During cell proliferation, ribosomes serve as the factories for protein synthesis and are primarily composed of proteins (*Neidhardt et al., 1990*; *Nelson and Cox, 2008*), while other biomass components, such as RNA, are optimally produced (*Kostinski and Reuveni, 2020*) in accordance with the growth rate determined by protein synthesis. Thus, the cell growth rate is proportional to $\phi_{\mathrm{R}}$: $\lambda = \phi_{\mathrm{R}} \cdot \kappa_{\mathrm{t}}$, where $\kappa_{\mathrm{t}}$ is a parameter set by the translation rate (*Scott et al., 2010*) (see Appendix 2.1 for details), which can be approximated as a constant within the growth rate range of interest (*Dai et al., 2017*).

For balanced cell growth in bacteria, the energy demand $J_{\mathrm{E}}$, expressed as the stoichiometric energy flux in ATP, is generally proportional to the biomass production rate (*Ebenhöh et al., 2024*), since the proportion of maintenance energy is roughly negligible (*Locasale and Cantley, 2010*) (see Appendix 10 for the cases of yeast and tumor cells). Thus, the normalized flux of energy demand in ATP, denoted as $J_{\mathrm{E}}^{(\mathrm{N})}$, representing the energy demand per unit of biomass, is proportional to the growth rate $\lambda$ (see Appendix 3.1 for details):

$$J_{\mathrm{E}}^{(\mathrm{N})} = \eta_{\mathrm{E}} \cdot \lambda, \tag{2}$$

where $\eta_{\mathrm{E}}$ is an energy coefficient (see *Equations S25 and S26* for details). By converting all energy currencies (such as NADH, FADH2, etc.) into ATP, the normalized energy fluxes for respiration and fermentation are given by $J_r^{(\mathrm{E})} = \beta_r^{(A)} \cdot J_r^{(\mathrm{N})}/2$ and $J_f^{(\mathrm{E})} = \beta_f^{(A)} \cdot J_f^{(\mathrm{N})}/2$, where $\beta_r^{(A)}$ and $\beta_f^{(A)}$ are the stoichiometric coefficients of ATP production per glucose in each pathway (see *Appendix 1—figure 1C-E* and Appendix 3.1 for details). The denominator coefficient of '2' is derived from the stoichiometry of the coarse-grained reaction $M_1 \to 2M_2$ (see *Figure 1A and B*). Applying the criteria of flux balance (i.e. mass conservation; see Appendix 2.3) at each intermediate node ($M_i$, $i = 1, \ldots, 5$) and precursor pool (Pool $i$, $i = $ a1, a2, b, c, d), along with the constraints of proteome allocation (see *Equation 1*) and energy demand (see *Equation 2*), we obtain the relations between normalized energy fluxes and growth rate for a given nutrient condition with a fixed $\kappa_A$ (see Appendix 3.1 for details):

$$\begin{cases} J_r^{(\mathrm{E})} + J_f^{(\mathrm{E})} = \varphi \cdot \lambda, \\ \dfrac{J_r^{(\mathrm{E})}}{\varepsilon_r} + \dfrac{J_f^{(\mathrm{E})}}{\varepsilon_f} = \phi_{\max} - \psi \cdot \lambda, \end{cases} \tag{3}$$

where $\varphi$ is a constant coefficient primarily determined by the coefficient $\eta_{\mathrm{E}}$ (see *Equation S33*), and $\varphi \cdot \lambda$ represents the normalized flux of energy demand, excluding energy biogenesis from the biomass synthesis pathway. The coefficients $\psi$, $\varepsilon_r$, and $\varepsilon_f$ are functions of $\kappa_A$, such that their values are highly dependent on nutrient conditions. $\psi^{-1}$ denotes the proteome efficiency for biomass generation in the biomass synthesis pathway (see *Equation S32*), defined as $\psi^{-1} \equiv \lambda/\phi_{\mathrm{BM}}$ (see Appendix 3.1). $\varepsilon_r$ and $\varepsilon_f$ represent the proteome efficiencies for energy biogenesis in the respiration and fermentation pathways, respectively, defined as the normalized energy fluxes expressed in ATP generated per proteomic mass fraction, with $\varepsilon_r \equiv J_r^{(\mathrm{E})}/\phi_r$ and $\varepsilon_f \equiv J_f^{(\mathrm{E})}/\phi_f$. Hence,

$$\begin{cases} \varepsilon_r = \dfrac{\beta_r^{(A)}}{1/\kappa_A + 1/\kappa_r^{(A)}}, \\ \varepsilon_f = \dfrac{\beta_f^{(A)}}{1/\kappa_A + 1/\kappa_f^{(A)}}, \end{cases} \tag{4}$$

where both $\kappa_r^{(A)}$ and $\kappa_f^{(A)}$ are composite parameters that can be approximated as constants, with $1/\kappa_r^{(A)} \equiv 1/\kappa_1 + 2/\kappa_2 + 2/\kappa_3 + 2/\kappa_4$ and $1/\kappa_f^{(A)} \equiv 1/\kappa_1 + 2/\kappa_2 + 2/\kappa_6$ (see Appendices 2.5 and 3.1 for details).

## Origin of overflow metabolism

The standard picture of overflow metabolism (**Basan et al., 2015**; **Holms, 1996**; **Meyer et al., 1984**; **Nanchen et al., 2006**; **van Hoek et al., 1998**) is exemplified by the experimental data (**Basan et al., 2015**) presented in **Figure 1C**, where the fermentation flux exhibits a threshold-analog dependence on the growth rate $\lambda$. It is well established that respiration is significantly more efficient than fermentation in terms of energy biogenesis per unit of carbon (i.e. $\beta_r^{(A)} > \beta_f^{(A)}$) (**Nelson and Cox, 2008**; **Vander Heiden et al., 2009**). Then, why do cells bother to use the seemingly wasteful fermentation pathway? We proceed to address this issue by applying optimal protein allocation (**Scott et al., 2010**; **Wang et al., 2019**) within the framework of optimal growth.

For cell proliferation in a given nutrient condition (i.e. with a fixed $\kappa_A$), the values of $\varepsilon_r$, $\varepsilon_f$, and $\psi$ are determined (see **Equations 4 and S32**). However, the growth rate $\lambda$ can be influenced by protein allocation between respiration and fermentation, specifically $\phi_r$ and $\phi_f$, according to the governing equation (**Equation 3**). If $\varepsilon_r > \varepsilon_f$, that is, if the proteome efficiency in respiration is higher than that in fermentation, then $\lambda = \frac{\phi_{\max} - J_f^{(E)}\left(1/\varepsilon_f - 1/\varepsilon_r\right)}{\psi + \varphi/\varepsilon_r} \leq \frac{\phi_{\max}}{\psi + \varphi/\varepsilon_r}$. The optimal growth strategy is $\phi_f = J_f^{(E)} = 0$, meaning that the cell exclusively uses respiration. Conversely, if $\varepsilon_f > \varepsilon_r$, then $\phi_r = J_r^{(E)} = 0$ is optimal, and the cell solely uses fermentation. In either case, the choice between respiration and fermentation for growth optimization is determined by comparing their proteome efficiencies.

In practice, both proteome efficiencies $\varepsilon_r$ and $\varepsilon_f$ are functions of nutrient quality $\kappa_A$, which can be significantly influenced by the nutrient type and concentration of the carbon source (see **Equations 4 and S27**). Therefore, the optimal growth strategy may vary depending on the nutrient conditions. In nutrient-poor conditions where $\kappa_A \ll \kappa_r^{(A)}$ and $\kappa_A \ll \kappa_f^{(A)}$, the proteome efficiencies can be approximated by $\varepsilon_r \approx \beta_r^{(A)} \cdot \kappa_A$ and $\varepsilon_f \approx \beta_f^{(A)} \cdot \kappa_A$ (see **Equation 4**), and hence $\varepsilon_r\left(\kappa_A\right) > \varepsilon_f\left(\kappa_A\right)$ (since $\beta_r^{(A)} > \beta_f^{(A)}$), meaning that the proteome efficiency of respiration is higher than that of fermentation under these conditions. In contrast, in rich media, using parameters for $\kappa_i$ derived from in vivo/in vitro experimental data for *E. coli* (see **Appendix 1—table 1**, **Appendix 1—table 2** and Appendix 7.1–7.2), we obtain $\varepsilon_r\left(\kappa_{\text{glucose}}^{(\text{ST})}\right) < \varepsilon_f\left(\kappa_{\text{glucose}}^{(\text{ST})}\right)$ with **Equation 4** (see also **Equations S39-S40**), where $\kappa_{\text{glucose}}^{(\text{ST})}$ represents the substrate quality of glucose at saturated concentration (abbreviated as 'ST' in the superscript). This indicates that the proteome efficiency in fermentation is higher than that in respiration for bacteria in rich media. Indeed, recent studies have validated that the measured proteome efficiency in fermentation is higher than in respiration for *E. coli* in lactose at saturated concentration (**Basan et al., 2015**), i.e., $\varepsilon_r\left(\kappa_{\text{lactose}}^{(\text{ST})}\right) < \varepsilon_f\left(\kappa_{\text{lactose}}^{(\text{ST})}\right)$. In **Figure 1E**, we present the growth rate dependence of proteome efficiencies $\varepsilon_r$ and $\varepsilon_f$ in a three-dimensional (3D) format using the collected data shown in **Appendix 1—table 1**, where $\varepsilon_r$, $\varepsilon_f$ and the growth rate $\lambda$ all vary as functions of nutrient quality $\kappa_A$. Furthermore, the ratio $\Delta$ (defined as $\Delta\left(\kappa_A\right) \equiv \varepsilon_f\left(\kappa_A\right)/\varepsilon_r\left(\kappa_A\right)$) is a monotonically increasing function of $\kappa_A$, and there exists a critical value of $\kappa_A$ (denoted as $\kappa_A^{(C)}$; see Appendix 3.2 for details) satisfying $\Delta\left(\kappa_A^{(C)}\right) = 1$. Below $\kappa_A^{(C)}$, where the nutrient is poorer and the cell grows slowly, the proteome efficiency of fermentation is lower than that of respiration (i.e. $\varepsilon_f < \varepsilon_r$), hence respiration is the optimal choice (with $\lambda = \phi_{\max} \cdot \left(\psi + \varphi/\varepsilon_r\right)^{-1}$). Above $\kappa_A^{(C)}$, where the nutrient is richer and the cell grows faster, fermentation is more efficient than respiration in terms of proteome efficiency (i.e. $\varepsilon_f > \varepsilon_r$) and becomes the optimal growth strategy (with $\lambda = \phi_{\max} \cdot \left(\psi + \varphi/\varepsilon_f\right)^{-1}$). This analysis qualitatively explains the phenomenon of aerobic glycolysis.

For a quantitative understanding of overflow metabolism, let us first consider the homogeneous case, where all cells share identical biochemical parameters. For optimal protein allocation, the relation between fermentation flux and growth rate under nutrient variation (with significantly varying $\kappa_A$) is given by $J_f^{(E)} = \varphi \cdot \lambda \cdot \theta\left(\lambda - \lambda_C\right)$, where '$\theta$' represents the Heaviside step function, and $\lambda_C$ denotes the critical

growth rate corresponding to the nutrient condition with nutrient quality $\kappa_A^{(C)}$ (i.e. $\lambda_C \equiv \lambda\left(\kappa_A^{(C)}\right)$).

Similarly, the growth rate dependence of respiration flux is $J_r^{(E)} = \varphi \cdot \lambda \cdot [1 - \theta\left(\lambda - \lambda_C\right)]$. These digital response outcomes are consistent with the numerical simulation findings of *Molenaar et al., 2009*. However, they are clearly incompatible with the threshold-analog response observed in the standard picture of overflow metabolism (*Basan et al., 2015*; *Holms, 1996*; *Meyer et al., 1984*; *Nanchen et al., 2006*; *van Hoek et al., 1998*).

To address this issue, we take into account cell heterogeneity, which is ubiquitous in both microbes (*Ackermann, 2015*; *Bagamery et al., 2020*; *Balaban et al., 2004*; *Nikolic et al., 2013*; *Solopova et al., 2014*; *Wallden et al., 2016*; *Yaginuma et al., 2014*; *Zhang et al., 2018*) and tumor cells (*Duraj et al., 2021*; *Shibao et al., 2018*; *Hanahan and Weinberg, 2011*; *Hensley et al., 2016*). In the context of the Warburg effect or overflow metabolism, experimental studies have reported significant metabolic heterogeneity in the choice between respiration and fermentation within a cell population (*Bagamery et al., 2020*; *Duraj et al., 2021*; *Shibao et al., 2018*; *Hensley et al., 2016*; *Nikolic et al., 2013*). Motivated by the observation that the turnover number ($k_{cat}$ value) of a catalytic enzyme varies considerably between in vitro and in vivo measurements (*Davidi et al., 2016*; *García-Contreras et al., 2012*), we note that the concentrations of potassium and phosphate, which vary from cell to cell, have a significant impact on the $k_{cat}$ values of metabolic enzymes (*García-Contreras et al., 2012*). Therefore, within a cell population, there is a distribution of $k_{cat}$ values for a catalytic enzyme, commonly referred to as extrinsic noise (*Elowitz et al., 2002*). For simplicity, we assume that the $k_{cat}$ values for each enzyme follow a Gaussian distribution. Consequently, the proteome efficiencies $\varepsilon_r$ and $\varepsilon_f$, which are crucial for determining the choice between respiration and fermentation, also follow Gaussian distributions (see Appendix 8 for details). This variability leads to diverse distributions of single-cell growth rates across different carbon sources (see *Equations S155-S157 and S163-S165*), which has been fully verified by recent experiments using isogenic *E. coli* at single-cell resolution (*Wallden et al., 2016*; see *Appendix 1—figure 2B*). Accordingly, the critical growth rate $\lambda_C$ is expected to follow a Gaussian distribution $\mathcal{N}\left(\mu_{\lambda_C}, \sigma_{\lambda_C}^2\right)$ within a cell population (see Appendix 8 for details), where $\mu_{\lambda_C}$ is approximated by the deterministic result of $\lambda_C$ (*Equation S43*). Assuming the coefficient of variation (CV) of $\lambda_C$ is $\sigma_{\lambda_C}/\mu_C = 12\%$, or equivalently that the CV for the catalytic rate of each metabolic enzyme is 25%, we derive the growth rate dependence of fermentation and respiration fluxes (see Appendix 3.3 for details):

$$
\begin{cases}
J_f^{(N)}\left(\lambda\right) = \dfrac{\varphi \cdot \lambda}{\beta_f^{(A)}} \cdot \left[\mathrm{erf}\left(\dfrac{\lambda - \mu_{\lambda_C}}{\sqrt{2}\sigma_{\lambda_C}}\right) + 1\right], \\
J_r^{(N)}\left(\lambda\right) = \dfrac{\varphi \cdot \lambda}{\beta_r^{(A)}} \cdot \left[1 - \mathrm{erf}\left(\dfrac{\lambda - \mu_{\lambda_C}}{\sqrt{2}\sigma_{\lambda_C}}\right)\right],
\end{cases}
\tag{5}
$$

where 'erf' represents the error function. The fermentation flux exhibits a threshold-analog relation with the growth rate (the red curves in *Figures 1C–D, 2B–C, 3B, D and F*), while the respiration flux (the blue curve in *Figure 1D*) decreases as the fermentation flux increases. In *Figure 1C–D*, we observe that the model results (see *Equation 5* and Appendix 9 for details; parameters are set based on the experimental data shown in *Appendix 1—table 1*) quantitatively agree with the experimental data from *E. coli* (*Basan et al., 2015*; *Holms, 1996*). The fermentation flux is represented by the acetate secretion rate $J_{actate}^{(M)} = 2J_f^{(N)}$, and the respiration flux is exemplified by the carbon dioxide flux $J_{CO_2,r}^{(M)} = 6J_r^{(N)}$ (the superscript '(M)' represents the measurable flux in the unit of mM/OD600/h; see Appendix 9.1 for details). By incorporating cell heterogeneity, our model of optimal protein allocation quantitatively explains overflow metabolism.

## Testing the model through perturbations

To further test our model, we systematically investigate its predictions under various types of perturbations and compare them with experimental data from existing studies (*Basan et al., 2015*; *Holms, 1996*) (see Appendices 4 and 5.1 for details).

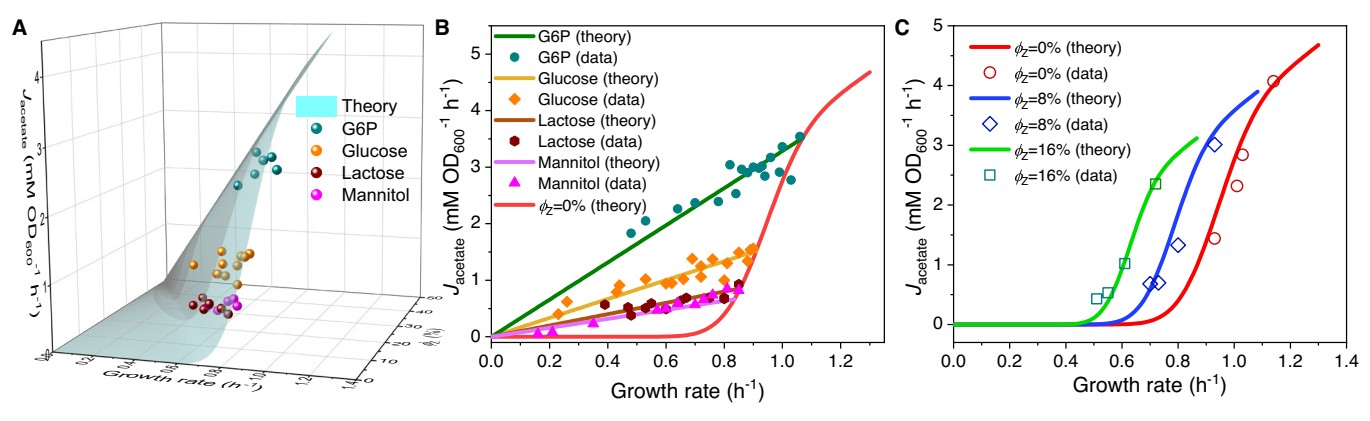

**Figure 2.** Influence of protein overexpression on overflow metabolism in *E. coli*. (**A**) A 3D plot of the relations among fermentation flux, growth rate, and the expression level of useless proteins. In this plot, both the acetate excretion rate and growth rate vary as bivariate functions of the nutrient quality of a Group A carbon source (denoted as $\kappa_A$) and the useless protein expression encoded by *lacZ* gene (denoted as $\phi_Z$ perturbation; see *Equations S57 and S160*). (**B**) Growth rate dependence of the acetate excretion rate upon $\phi_Z$ perturbation for each fixed nutrient condition (see *Equations S58 and S160*). (**C**) Growth rate dependence of the acetate excretion rate as $\kappa_A$ varies (see *Equations S58 and S160*), with each fixed expression level of LacZ.

First, we consider the proteomic perturbation caused by overexpression of useless proteins encoded by the *lacZ* gene (i.e. $\phi_Z$ perturbation) in *E. coli*. The net effect of the $\phi_Z$ perturbation is that the maximum fraction of the proteome available for resource allocation changes from $\phi_{\max}$ to $\phi_{\max} - \phi_Z$ (*Basan et al., 2015*), where $\phi_Z$ is the proteomic mass fraction of useless proteins. In a cell population, the critical growth rate $\lambda_C(\phi_Z)$ still follows a Gaussian distribution $\mathcal{N}\left(\mu_{\lambda_C}(\phi_Z), \sigma_{\lambda_C}(\phi_Z)^2\right)$, where the CV of $\lambda_C(\phi_Z)$ remains unchanged. Consequently, the growth rate dependence of fermentation flux changes to $J_f^{(N)} = \frac{\varphi \cdot \lambda}{\beta_f^{(A)}} \cdot \left[\mathrm{erf}\left(\frac{\lambda - \mu_{\lambda_C}(\phi_Z)}{\sqrt{2}\sigma_{\lambda_C}(\phi_Z)}\right) + 1\right]$ (see Appendix 4 for model perturbation results regarding respiration flux), where both the growth rate $\lambda(\kappa_A, \phi_Z)$ and the normalized fermentation flux $J_f^{(N)}(\kappa_A, \phi_Z)$ are bivariate functions of $\kappa_A$ and $\phi_Z$ (see *Equations S49, S56 and S57*). For each degree of LacZ expression (with fixed $\phi_Z$), similar to wild-type strains, the fermentation flux exhibits a threshold-analog response to growth rate as $\kappa_A$ varies (see *Figure 2C*), which agrees quantitatively with experimental results (*Basan et al., 2015*). The shifts in the critical growth rate $\lambda_C(\phi_Z)$ are fully captured by $\mu_{\lambda_C}(\phi_Z) = \mu_{\lambda_C}(0)(1 - \phi_Z/\phi_{\max})$. In contrast, for nutrient conditions with each fixed $\kappa_A$, since the growth rate changes with $\phi_Z$ just like $\lambda_C(\phi_Z)$ : $\lambda(\kappa_A, \phi_Z) = \lambda(\kappa_A, 0)(1 - \phi_Z/\phi_{\max})$, the fermentation flux is then proportional to the growth rate for the varying levels of LacZ expression: $J_f^{(N)} = \frac{\varphi}{\beta_f^{(A)}} \cdot \left[\mathrm{erf}\left(\frac{\lambda(\kappa_A, 0) - \mu_{\lambda_C}(0)}{\sqrt{2}\sigma_{\lambda_C}(0)}\right) + 1\right] \cdot \lambda$, where the slope is a monotonically increasing function of the substrate quality $\kappa_A$. These scaling relations are well validated by the experimental data (*Basan et al., 2015*) shown in *Figure 2B*. Finally, in the case where both $\kappa_A$ and $\phi_Z$ are free to vary, the growth rate dependence of fermentation flux presents a threshold-analog response surface in a 3D plot, where $\phi_Z$ appears explicitly as the *y*-axis (see *Figure 2A*). Experimental data points (*Basan et al., 2015*) lie right on this surface, which is highly consistent with the model predictions.

Next, we study the influence of energy dissipation, which introduces an energy dissipation coefficient $w$ to *Equation 2*: $J_E^{(N)} = \eta_E \cdot \lambda + w$. Similarly, the critical growth rate in this case, $\lambda_C(w)$, follows a Gaussian distribution $\mathcal{N}\left(\mu_{\lambda_C}(w), \sigma_{\lambda_C}(w)^2\right)$ in a cell population. The relation between the growth rate and fermentation flux can be characterized by: $J_f^{(N)} = \frac{\varphi \cdot \lambda + w}{\beta_f^{(A)}} \cdot \left[\mathrm{erf}\left(\frac{\lambda - \mu_{\lambda_C}(w)}{\sqrt{2}\sigma_{\lambda_C}(w)^2}\right) + 1\right]$ (see Appendix 4.2 for details). In *Figure 3A–B*, we present a comparison between the model results and experimental data (*Basan et al., 2015*) in 3D and 2D plots, which demonstrate good agreement. A notable characteristic of energy dissipation, as distinguished from $\phi_Z$ perturbation, is that the fermentation flux increases despite a decrease in the growth rate when $\kappa_A$ is fixed.

We proceed to analyze the impact of translation inhibition with different sub-lethal doses of chloramphenicol on *E. coli*. This type of perturbation introduces an inhibition coefficient $\iota$ to the translation rate, thus turning $\kappa_t$ into $\kappa_t/\iota + 1$. Still, the critical growth rate $\lambda_C(\iota)$ follows a Gaussian

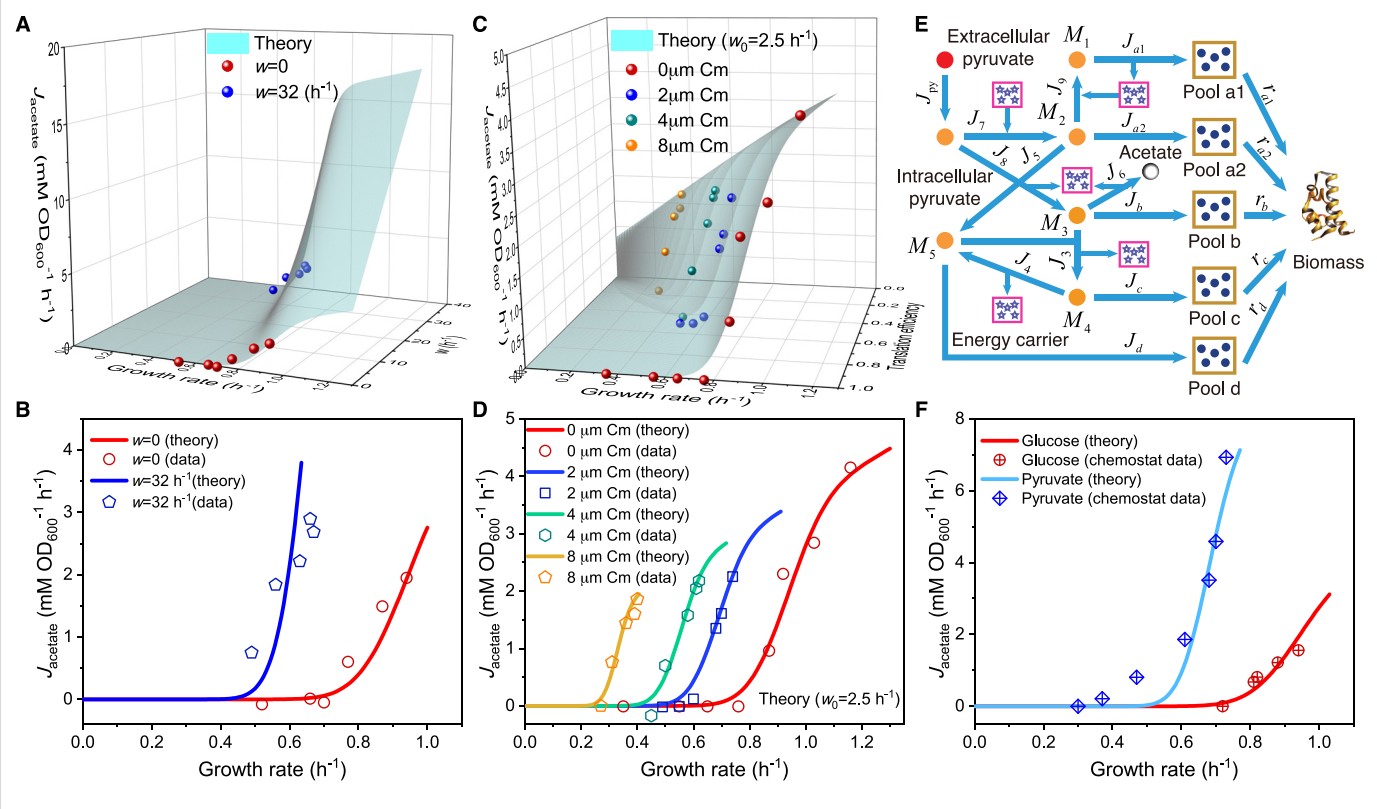

**Figure 3.** Influence of energy dissipation, translation inhibition, and carbon source category alteration on overflow metabolism in *E. coli*. (**A**) A 3D plot of the relations among fermentation flux, growth rate, and the energy dissipation coefficient (see *Equations S70 and S160*). (**B**) Growth rate dependence of the acetate excretion rate as the nutrient quality $\kappa_A$ varies, with each fixed energy dissipation coefficient determined by or fitted from experimental data. (**C**) A 3D plot of the relations among fermentation flux, growth rate, and the translation efficiency (see *Equations 85 and S160*). Here, the translation efficiency is adjusted by the dose of chloramphenicol (Cm). (**D**) Growth rate dependence of the acetate excretion rate as $\kappa_A$ varies, with each fixed dose of Cm. (**E**) Coarse-grained model for pyruvate utilization. (**F**) The growth rate dependence of fermentation flux in pyruvate (see *Equations 105 and S160*) significantly differs from that of the Group A carbon sources (see *Equations 47 and S160*).

distribution $\mathcal{N}\left(\mu_{\lambda_C}(\iota), \sigma_{\lambda_C}(\iota)^2\right)$, and then, the growth rate dependence of fermentation flux is given by: $J_f^{(N)} = \frac{\varphi \cdot \lambda}{\beta_f^{(A)}} \cdot \left[\text{erf}\left(\frac{\lambda - \mu_{\lambda_C}(\iota)}{\sqrt{2}\sigma_{\lambda_C}(\iota)}\right) + 1\right]$ (see Appendix 4.3 for details). In *Appendix 1—figure 2D and E*, we observe that the model predictions are generally consistent with the experimental data (*Basan et al., 2015*). However, a noticeable systematic discrepancy arises when the translation rate is low. Therefore, we consider maintenance energy, which is typically tiny and generally negligible for bacteria over the growth rate range of interest (*Basan et al., 2015*; *Locasale and Cantley, 2010*; *Neidhardt, 1996*). Encouragingly, by assigning a very small value to the maintenance energy coefficient $w_0$ (where $w_0 = 2.5\left(\text{h}^{-1}\right)$), the model results for the growth rate-fermentation flux relation $J_f^{(N)} = \frac{\varphi \cdot \lambda + w_0}{\beta_f^{(A)}} \cdot \left[\text{erf}\left(\frac{\lambda - \mu_{\lambda_C}(\iota)}{\sqrt{2}\sigma_{\lambda_C}(\iota)}\right) + 1\right]$ quantitatively agree with experiments (*Basan et al., 2015*) (see *Figure 3C–D* and Appendix 4.3 for details).

Finally, we consider the alteration of nutrient categories by switching to a non-Group A carbon source: pyruvate, which enters the metabolic network from the endpoint of glycolysis (*Neidhardt et al., 1990*; *Nelson and Cox, 2008*). The coarse-grained model for pyruvate utilization is shown in *Figure 3E* (see also *Figure 1A*), which shares identical precursor pools with those for Group A carbon sources, yet has several differences in the coarse-grained reactions. The growth rate dependencies of both the proteome efficiencies (see *Appendix 1—figure 2H*) and energy fluxes (see *Figure 3F*) are qualitatively similar to those of Group A carbon source utilization, while there are quantitative differences in the coarse-grained parameters (see Appendices 5.1 and 9 for derivation details).

Most notably, the critical growth rate $\lambda_C^{(py)}$ and the ATP production per glucose in the fermentation pathway $\beta_f^{(py)}$ for pyruvate utilization are noticeably smaller than those for Group A sources (i.e. $\lambda_C$ and $\beta_f^{(A)}$, respectively). Consequently, the growth rate dependence of fermentation flux in pyruvate should present a distinctly different curve from that of Group A carbon sources (see *Equations 5 and S105*), which is fully validated by experimental results (*Holms, 1996*; see *Figure 3F*).

## Enzyme allocation under perturbations

As mentioned above, our coarse-grained model is topologically identical to the central metabolic network (see *Figure 1A*) and can thus predict enzyme allocation for each gene in glycolysis and the TCA cycle (see *Appendix 1—figure 1B* and *Appendix 1—table 1*) under various types of perturbations. In *Figure 1B*, the intermediate nodes $M_1$, $M_2$, $M_3$, $M_4$, and $M_5$ represent G6P, PEP, acetyl-CoA, $\alpha$-ketoglutarate, and oxaloacetate, respectively. Therefore, $\phi_1$ and $\phi_2$ correspond to enzymes involved in glycolysis (or at the junction of glycolysis and the TCA cycle), while $\phi_3$ and $\phi_4$ correspond to enzymes in the TCA cycle (see *Figure 1A–B* and Appendix 3.1).

We first consider enzyme allocation under carbon limitation by varying the nutrient type and concentration of a Group A carbon source (i.e. $\kappa_A$ perturbation). This has been extensively studied in more simplified models (*Hui et al., 2015*; *You et al., 2013*), where the growth rate dependence of enzyme allocation under $\kappa_A$ perturbation is generally described by a C-line response (*Hui et al., 2015*; *You et al., 2013*). Specifically, the genes responsible for digesting carbon compounds exhibit a linear increase in gene expression as the growth rate decreases (*Hui et al., 2015*; *You et al., 2013*). However, when it comes to enzymes catalyzing reactions between intermediate nodes, we gathered experimental data from existing studies (*Hui et al., 2015*) and found that the enzymes in glycolysis exhibit a completely different response pattern compared to those in the TCA cycle (see *Appendix 1—figure 3A and B*). This discrepancy cannot be explained by the C-line response. To address this issue, we

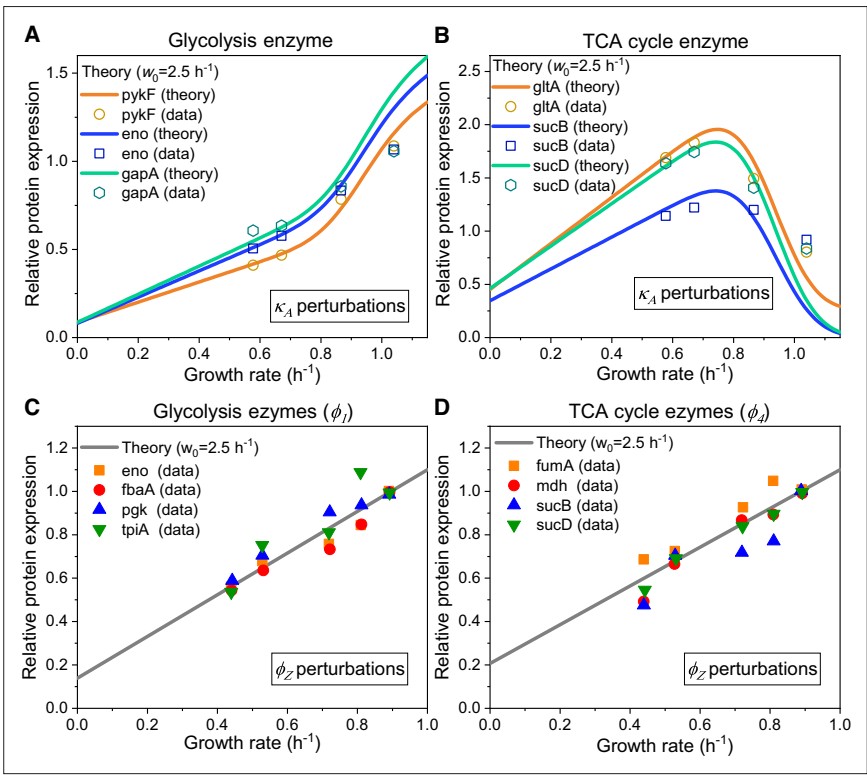

**Figure 4.** Relative protein expression of central metabolic enzymes in *E. coli* under carbon limitation and proteomic perturbation. (**A, C**) Relative protein expression of representative genes from glycolysis. (**B, D**) Relative protein expression of representative genes from the TCA cycle. (**A, B**) Results of the perturbation through changes in nutrient quality $\kappa_A$ (see *Equation S119*). (**C, D**) Results of proteomic perturbation via varied levels of expression of the useless protein LacZ (i.e. $\phi_Z$ perturbation; see *Equation S121*).

apply the coarse-grained model described above (see *Figure 1B*) to calculate the growth rate dependence of enzyme allocation for each $\phi_i$ ($i = 1, 2, 3, 4$) using model settings for wild-type strains, with no fitting parameters influencing the shape (see *Equations S118-S119* and Appendix 9). In *Figure 4A–B* and *Appendix 1—figure 3C-D*, we see that the model predictions overall match with the experimental data (*Hui et al., 2015*) for representative genes from either glycolysis or the TCA cycle, and maintenance energy (with $w_0 = 2.5 \left(\text{h}^{-1}\right)$) has a negligible effect on this process. Still, there are minor discrepancies that arise from the basal expression of metabolic genes, which may be attributed to the fact that our model deals with relatively stable growth conditions while microbes need to be prepared for fluctuating environments (*Basan et al., 2020*; *Kussell and Leibler, 2005*; *Mori et al., 2017*).

We proceed to analyze the influence of $\phi_Z$ perturbation and energy dissipation. In both cases, our model predicts a linear response to growth rate reduction for all genes in either glycolysis or the TCA cycle (see Appendix 6.2–6.3 for details). For $\phi_Z$ perturbation, all predicted slopes are positive, and there are no fitting parameters involved (*Equations S120-S121*). In *Figure 4C–D* and *Appendix 1—figure 3E-J*, we show that our model quantitatively illustrates the experimental data (*Basan et al., 2015*) for representative genes in the central metabolic network, and there is a better agreement with experiments (*Basan et al., 2015*) by incorporating the maintenance energy (with $w_0 = 2.5 \left(\text{h}^{-1}\right)$ as aforementioned). For energy dissipation, however, the predicted slopes of the enzymes corresponding to $\phi_4$ are negative, and there is a constraint that the slope signs of the enzymes corresponding to the same $\phi_i$ ($i = 1, 2, 3$) should be the same. In *Appendix 1—figure 3K-N*, we see that the model results (*Equations S127 and S123*) are consistent with experiments (*Basan et al., 2015*).

## Explanation of the Crabtree effect in yeast and the Warburg effect in cancer cells

We proceed to apply our model to explain the Crabtree effect in yeast (*Bagamery et al., 2020*; *De Deken, 1966*; *Shen et al., 2024*) and the Warburg effect in tumors (*Bartman et al., 2023*; *Duraj et al., 2021*; *Hanahan and Weinberg, 2011*; *Shen et al., 2024*; *Vander Heiden et al., 2009*) with slight modifications using the optimal growth principle combined with cell heterogeneity (see Appendix 10 and *Appendix 1—figure 5*). For yeast and tumors, similar to the case of *E. coli*, the proteome efficiencies $\varepsilon_r$ and $\varepsilon_f$ are both increasing functions of nutrient quality $\kappa_A$ (see *Equation S170*). Under poor nutrient conditions (i.e. $\kappa_A$ is small), the proteome efficiency in respiration is higher than that in fermentation: $\varepsilon_r > \varepsilon_f$ (see *Equations S174-S175*), making respiration the optimal choice for growth optimization (see *Equation S171*). Conversely, when nutrients are abundant and $\varepsilon_f > \varepsilon_r$, aerobic glycolysis (i.e. fermentation) becomes the optimal growth strategy (see *Equation S172*). Further combination with cell heterogeneity results in the standard picture of overflow metabolism, which has indeed been observed in yeast (*van Hoek et al., 1998*). However, it remains challenging to tune the growth rate of cancer cells in vivo.

Recently, *Shen et al., 2024* discovered that the proteome efficiency measured at the cell population level in respiration (i.e. $\langle \varepsilon_r \rangle$; where '$\langle \rangle$' denotes the population average) is higher than that in fermentation (i.e. $\langle \varepsilon_f \rangle$) for many yeast and cancer cells, despite the presence of fermentation fluxes through aerobic glycolysis. Evidently, this finding (*Shen et al., 2024*) contradicts prevalent explanations (*Basan et al., 2015*; *Chen and Nielsen, 2019*), which hold that overflow metabolism arises because the proteome efficiency in fermentation is consistently higher than in respiration. Nevertheless, our model may resolve this puzzle due to the incorporation of two important features. First, our model predicts that the proteome efficiency in respiration is larger than that in fermentation when nutrient quality is low (see *Equations S174-S175*). Second, and crucially, by accounting for cell heterogeneity, our model allows a proportion of cells to have a higher proteome efficiency in fermentation than in respiration, even when the overall proteome efficiency in respiration at the cell population level is greater than that in fermentation (i.e. $\langle \varepsilon_r \rangle > \langle \varepsilon_f \rangle$).

To compare our model results quantitatively with experimental data on yeast and tumors (*Shen et al., 2024*), we define $\text{Pr}_f \equiv \frac{J_f^{(\text{E})}}{J_f^{(\text{E})} + J_r^{(\text{E})}}$ as the fraction of ATP produced through fermentation. To account for cell heterogeneity, we apply Gaussian distributions to enzyme turnover numbers, as described above. This yields the relationship between $\text{Pr}_f$ (i.e. $\frac{J_f^{(\text{E})}}{J_f^{(\text{E})} + J_r^{(\text{E})}}$) and $\langle \varepsilon_r \rangle$ and $\langle \varepsilon_f \rangle$ through derivations (see *Equations S180-S190* and Appendix 10 for details):

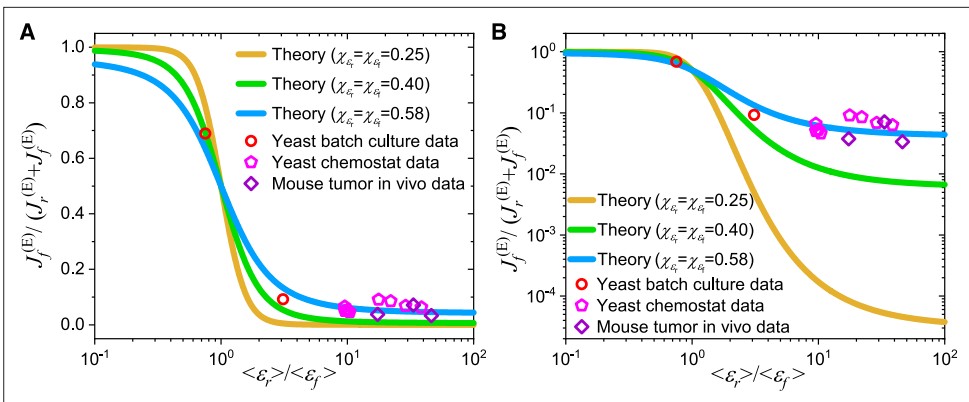

**Figure 5.** Model comparison with data on the Crabtree effect in yeast and the Warburg effect in tumors. (**A**) A linear scale representation on the *y*-axis. (**B**) A log scale representation on the *y*-axis. In (**A–B**), $\langle \varepsilon_r \rangle$ and $\langle \varepsilon_f \rangle$ represent the population averages of $\varepsilon_r$ and $\varepsilon_f$, while $\chi_{\varepsilon_r}$ and $\chi_{\varepsilon_f}$ are the coefficients of variation (CVs) of $\varepsilon_r$ and $\varepsilon_f \cdot \langle \varepsilon_r \rangle / \langle \varepsilon_f \rangle$ represents the ratio of proteome efficiency between respiration and fermentation at the population-averaged level, while $J_f^{(E)} / \left( J_f^{(E)} + J_r^{(E)} \right)$ stands for the fraction of energy flux generated by the fermentation pathway (see *Equation 6*). The data for yeast in batch culture and chemostat were calculated from experimental data of *S. cerevisiae* and *I. orientalis* (***Shen et al., 2024***). The data for mouse tumors were calculated from in vivo experimental data of pancreatic ductal adenocarcinoma (PDAC) and leukemic spleen of mice (***Bartman et al., 2023***; ***Shen et al., 2024***). See Appendix 11 for detailed information on the experimental data sources (***Bartman et al., 2023***; ***Shen et al., 2024***).

$$\frac{J_f^{(E)}}{J_f^{(E)} + J_r^{(E)}} = \frac{1}{2} \left[ \mathrm{erf} \left( \frac{1 - \langle \varepsilon_r \rangle / \langle \varepsilon_f \rangle}{\sqrt{2} \cdot \sqrt{\chi_{\varepsilon_r}^2 + \chi_{\varepsilon_f}^2 \cdot \left( \langle \varepsilon_r \rangle / \langle \varepsilon_f \rangle \right)^2}} \right) + 1 \right], \tag{6}$$

where $\chi_{\varepsilon_r}$ and $\chi_{\varepsilon_f}$ represent the CVs of proteome efficiencies $\varepsilon_r$ and $\varepsilon_f$, respectively. Due to the higher levels of cell heterogeneity in yeast (***Bagamery et al., 2020***) and cancer cells (***Duraj et al., 2021***; ***Shibao et al., 2018***; ***Hanahan and Weinberg, 2011***; ***Hensley et al., 2016***), the CVs of $\varepsilon_r$ and $\varepsilon_f$ (i.e. $\chi_{\varepsilon_r}$ and $\chi_{\varepsilon_f}$) in these cells are expected to be significantly higher than those in *E. coli*, although their precise values are unknown. The values for the variables shown in *Equation 6* can be obtained from experiments. Therefore, we plot the theoretical results from *Equation 6* using $\chi_{\varepsilon_r}$ and $\chi_{\varepsilon_f}$ values of 0.25, 0.40, and 0.58 to compare with experimental data from yeast and in vivo mouse tumors (***Bartman et al., 2023***; ***Shen et al., 2024***). As shown in *Figure 5A–B*, the theoretical results with $\chi_{\varepsilon_r} = \chi_{\varepsilon_f} = 0.58$ align quantitatively with the experimental data (***Bartman et al., 2023***; ***Shen et al., 2024***) on both logarithmic and linear scales, demonstrating that our model has the potential to quantitatively explain the Crabtree effect in yeast and the Warburg effect in cancer cells.

## Discussion

The phenomenon of overflow metabolism, or the Warburg effect, has been a long-standing puzzle in cell metabolism. Although many rationales have been proposed over the past century (***Basan et al., 2015***; ***Chen and Nielsen, 2019***; ***Majewski and Domach, 1990***; ***Molenaar et al., 2009***; ***Niebel et al., 2019***; ***Peebo et al., 2015***; ***Pfeiffer et al., 2001***; ***Shlomi et al., 2011***; ***Vander Heiden et al., 2009***; ***Varma and Palsson, 1994***; ***Vazquez et al., 2010***; ***Zhuang et al., 2011***), contradictions persist (***Shen et al., 2024***), leaving the origin and function of this phenomenon unclear (***DeBerardinis and Chandel, 2020***; ***Hanahan and Weinberg, 2011***; ***Vander Heiden et al., 2009***). In this study, we use *E. coli* as a typical example and demonstrate that overflow metabolism can be understood through optimal protein allocation combined with cell heterogeneity. Under nutrient-poor conditions, the proteome efficiency of respiration is higher than that of fermentation (see *Figure 1E*), and thus the cell uses respiration to optimize growth. In rich media, however, the proteome efficiency of fermentation increases more rapidly and surpasses that of respiration (see *Figure 1E*), leading the cell to adopt fermentation as the optimal growth strategy. In further combination with cell heterogeneity in enzyme

catalytic rates (*Davidi et al., 2016*; *García-Contreras et al., 2012*), our model quantitatively illustrates the threshold-analog response (*Basan et al., 2015*; *Holms, 1996*) in overflow metabolism (see *Figure 1C*). Furthermore, it quantitatively explains the data on the Crabtree effect in yeast and the Warburg effect in cancer cells (*Bartman et al., 2023*; *Shen et al., 2024*).

Mechanistically, the optimal growth strategy for the binary choice between respiration and fermentation can be facilitated by the direct sensing and comparison of proteome efficiencies between the two pathways (see Appendix 3.4). A growing body of evidence suggests that the cyclic AMP (cAMP)-cAMP receptor protein (CRP) system plays a crucial role in sensing proteome efficiency and executing the optimal strategy (*Basan et al., 2015*; *Towbin et al., 2017*; *Valgepea et al., 2010*; *Wehrens et al., 2023*). However, it has also been suggested that the cAMP-CRP system alone is insufficient, and that additional regulators remain to be identified to fully elucidate this mechanism (*Basan et al., 2015*; *Valgepea et al., 2010*). Furthermore, since the binary choice between respiration and fermentation is driven by the comparison of proteome efficiencies, the optimal growth principle in our model can be relaxed to the case where efficient protein allocation is required only for enzymes, rather than ribosomes. This allows our model to remain applicable under suboptimal growth conditions (see Appendix 3.4 for details), where recent experimental studies have shown that the inactive portion of ribosomes (i.e. ribosomes not bound to mRNAs) may vary with culturing conditions (*Dai et al., 2017*; *Li et al., 2018*) and between individual cells within the same culture (*Pavlou et al., 2025*), despite an overall trend toward growth optimization.

In existing rationales (*Basan et al., 2015*; *Chen and Nielsen, 2019*; *Majewski and Domach, 1990*; *Shlomi et al., 2011*; *Varma and Palsson, 1994*; *Vazquez et al., 2010*; *Vazquez and Oltvai, 2016*), the standard picture of overflow metabolism (*Basan et al., 2015*; *Holms, 1996*; *Meyer et al., 1984*; *Nanchen et al., 2006*; *van Hoek et al., 1998*) has primarily been illustrated by a threshold-linear response, which largely relies on the assumption that cells optimize their growth rate for a given rate of carbon influx under each nutrient condition (or similar equivalents; see Appendix 7.3). However, in practice, for microbes or tumor cells grown in vitro or in vivo, the given factors are the identity and concentration of the nutrient (*Molenaar et al., 2009*; *Scott et al., 2010*; *Wang et al., 2019*), rather than the rate of carbon influx. Additionally, prevalent explanations (*Basan et al., 2015*; *Chen and Nielsen, 2019*) suggest that overflow metabolism originates from the proteome efficiency in fermentation always being higher than that in respiration (see Appendix 7.3 for details). While it has been observed in *E. coli* that proteome efficiency in fermentation is higher than that in respiration for cells cultured in lactose at saturated concentration (*Basan et al., 2015*), *Shen et al., 2024* reported that for many yeast and cancer cells, the proteome efficiency in fermentation is noticeably lower than that in respiration, despite the presence of aerobic glycolytic fermentation flux. This observation (*Shen et al., 2024*) evidently contradicts the prevalent explanations (*Basan et al., 2015*; *Chen and Nielsen, 2019*). Our model resolves this puzzle by significantly differing from existing rationales in its optimization principle, where we optimize cell growth rate purely through protein allocation without imposing a special constraint on carbon influx (see Appendix 7.3 for details). More importantly, our model incorporates cell heterogeneity, which is crucial for both explaining the threshold-analog response in overflow metabolism and for resolving this puzzle raised by *Shen et al., 2024*.

In the homogeneous case, the optimal growth strategy for growth rate dependent fermentation flux results in a digital response (see *Equation S44*), corresponding to an elementary flux mode (*Müller et al., 2014*; *Wortel et al., 2014*), which aligns with the numerical study by *Molenaar et al., 2009* but is incompatible with the standard picture of overflow metabolism (*Basan et al., 2015*; *Holms, 1996*; *Meyer et al., 1984*; *Nanchen et al., 2006*; *van Hoek et al., 1998*). Furthermore, in this case, cells would not generate fermentation flux if the proteome efficiency in fermentation were lower than that in respiration, under the optimal growth framework. By incorporating heterogeneity in enzyme catalytic rates (*Davidi et al., 2016*; *García-Contreras et al., 2012*), the critical growth rate (i.e. threshold) shifts from a single value to a Gaussian distribution (see *Equation S45* and Appendix 8 for details; see also *Appendix 1—figure 4*) across a cell population, thereby turning a digital response into the threshold-analog response observed in overflow metabolism (see *Figure 1C*). Moreover, cell heterogeneity allows a fraction of cells to possess a larger proteome efficiency in fermentation than in respiration despite the overall proteome efficiency in respiration at the cell population level is higher than in fermentation. This mechanism facilitates the fermentation flux in yeast and cancer cells observed by *Shen et al., 2024* (see *Figure 5A–B*).

Our model results, based on cell heterogeneity, are further supported by observed distributions of single-cell growth rates in *E. coli* (*Wallden et al., 2016*) (see *Appendix 1—figure 2B*), as well as by experiments involving various types of perturbations (*Basan et al., 2015*; *Holms, 1996*; *Hui et al., 2015*), both in terms of acetate secretion patterns and gene expression in the central metabolic network (see *Figures 2–4*, *Appendix 1—figures 2D and E and 3*). Furthermore, the heterogeneity patterns predicted by our model for fermentation and respiration modes in an isogenic cell population under the same culturing conditions are highly consistent with the non-genetic heterogeneity observed in single-cell experiments with *E. coli* (*Nikolic et al., 2013*) and *S. cerevisiae* (*Bagamery et al., 2020*), and align with experiments on intra-tumor heterogeneity in glioblastoma (*Duraj et al., 2021*; *Shibao et al., 2018*). Finally, our model can be broadly applied to address heterogeneity-related challenges in metabolism on a quantitative basis, including diverse metabolic strategies of cells in various environments (*Bagamery et al., 2020*; *Duraj et al., 2021*; *Escalante-Chong et al., 2015*; *Hensley et al., 2016*; *Liu et al., 2015*; *Solopova et al., 2014*; *Wang et al., 2019*).

## Acknowledgements

The author thanks Chao Tang, Qi Ouyang, Yang-Yu Liu, and Kang Xia for helpful discussions. This work was supported by the National Natural Science Foundation of China (Grant Nos.12004443 and 12474207), Guangzhou Municipal Innovation Fund (Grant No.202102020284), and the Hundred Talents Program of Sun Yat-sen University.

## Additional information

### Funding

| Funder | Grant reference number | Author |
| --- | --- | --- |
| National Natural Science Foundation of China | 12004443 | Xin Wang |
| National Natural Science Foundation of China | 12474207 | Xin Wang |
| Guangzhou Municipal Science and Technology Bureau | 202102020284 | Xin Wang |
| Sun Yat-sen University | The Hundred Talents Program | Xin Wang |

The funders had no role in study design, data collection and interpretation, or the decision to submit the work for publication.

### Author contributions

Xin Wang, Conceptualization, Resources, Data curation, Software, Formal analysis, Supervision, Funding acquisition, Validation, Investigation, Visualization, Methodology, Writing – original draft, Project administration, Writing – review and editing

### Author ORCIDs

Xin Wang ⓘ https://orcid.org/0000-0001-6479-395X

Reviewer #1 (Public review): https://doi.org/10.7554/eLife.94586.4.sa1
Author response https://doi.org/10.7554/eLife.94586.4.sa2

## Additional files

### Supplementary files

Source data 1. Source data for the theoretical results generated in this study and the experimental data from prior studies, as shown in *Figures 1–5*.

Source data 2. Source data for the theoretical results generated in this study and the experimental data from prior studies, as shown in *Appendix 1—figures 2–4*.

MDAR checklist

## Data availability

All study data are included in the manuscript and supporting files. All model results were generated using analytical formulas, with the relevant formulas and parameters specified in the manuscript and appendices. Source data files have been provided for Figures 1–5 and Appendix 1—figures 2–4.

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

# Appendix 1

**Appendix 1—table 1.** Molecular weight (MW) and in vivo/in vitro $k_{cat}$ data for *E. coli*.

| No.[*] | Reaction | Enzyme | Gene name | EC | MW (kDa) | In vitro $k_{cat}$ (s$^{-1}$) | References | In vivo[†] $k_{cat}$ (s$^{-1}$) | Selected $k_{cat}$ (s$^{-1}$) |
|---|---|---|---|---|---|---|---|---|---|
| | Glucose-6P ↔ Fructose-6P | Glucose-6-phosphate isomerase | pgi | EC:5.3.1.9 | $1.2\times10^2$ | $2.6\times10^2$ | PMID:7004378; DOI:https://doi.org/10.1016/j.ijms.2004.09.017 | $8.7\times10^2$ | $8.7\times10^2$ |
| | Fructose-6P → Fructose-1,6P | Phosphofructokinase | pfkA[‡] | EC:2.7.1.11 | $1.4\times10^2$ | $4.4\times10^2$ | PMID:6218375; 70226 | $1.7\times10^3$ | $1.7\times10^3$ |
| | Fructose-1,6P ↔ Glyceraldehyde 3-phosphate+Dihydroxyacetone phosphate | Fructose-bisphosphate aldolase | fbaA[†] | EC:4.1.2.13 | $7.8\times10$ | $1.4\times10$ | PMID:8939754; 15531627 | $1.6\times10^2$ | $1.6\times10^2$ |
| | Dihydroxyacetone phosphate ↔ Glyceraldehyde 3-phosphate | Triosephosphate Isomerase | tpiA | EC:5.3.1.1 | $5.4\times10$ | $4.3\times10^2$ | PMID:3887397; 6092857 | $2.7\times10^2$ | $2.7\times10^2$ |
| | Glyceraldehyde 3-phosphate ↔ 1,3-Bisphosphoglycerate | Glyceraldehyde-3-phosphate dehydrogenase | gapA | EC:1.2.1.12 | $1.4\times10^2$ | $9.5\times10$ | PMID:4932978; 2200929 | $1.5\times10^2$ | $1.5\times10^2$ |
| | 1,3-Bisphosphoglycerate ↔ 3-Phosphoglycerate | Phosphoglycerate kinase | pgk | EC:2.7.2.3 | $4.4\times10$ | $3.5\times10^2$ | PMID:367367; 166274 | $1.9\times10^2$ | $1.9\times10^2$ |
| | 3-Phosphoglycerate ↔ 2-Phosphoglycerate | Phosphoglycerate mutase | gpmA[‡] | EC:5.4.2.11 | $4.9\times10$ | $3.3\times10^2$ | PMID:10437801 | $4.5\times10^2$ | $4.5\times10^2$ |
| $J_1$ | 2-Phosphoglycerate ↔ Phosphoenolpyruvate | Enolase | eno | EC:4.2.1.11 | $9.0\times10$ | $2.2\times10^2$ | PMID:1094232; 4942326 | $1.7\times10^2$ | $1.7\times10^2$ |
| | Phosphoenolpyruvate → Pyruvate | Pyruvate kinase | pykF[‡] | EC:2.7.1.40 | $2.4\times10^2$ | $5.0\times10^2$ | PMID:6759852 | $1.6\times10^3$ | $1.6\times10^3$ |
| $J_2$ | Pyruvate → Acetyl-CoA | Pyruvate dehydrogenase | aceE[‡] | EC:1.2.4.1 | $1.0\times10^2$ | $1.2\times10^2$ | PMID:23088422 | $3.4\times10^2$ | $3.4\times10^2$ |
| | Oxaloacetate +Acetyl CoA → Citrate | Citrate synthase | gltA | EC:2.3.3.1 | $9.7\times10$ | $2.4\times10^2$ | PMID:4900996; 23954305 | $7.1\times10$ | $7.1\times10$ |
| | Citrate ↔ Isocitrate | Aconitate hydratase | acnB[‡] | EC:4.2.1.3 | $9.4\times10$ | $7.0\times10$ | PMID:15963579; 15963579 | $6.3\times10$ | $6.3\times10$ |
| $J_3$ | Isocitrate→ α-Ketoglutarate | Isocitrate dehydrogenase | icd | EC:1.1.1.42 | $9.5\times10$ | $2.0\times10^2$ | PMID:8141; 36923; 2200929 | $3.3\times10$ | $3.3\times10$ |
| | α-Ketoglutarate → Succinyl-CoA | α-Ketoglutarate dehydrogenase complex E1 component | suc A suc B[‡] | EC:1.2.4.2, EC:2.3.1.61 | $1.9\times10^2$ | $1.5\times10^2$ | PMID:6380583; 4588679 | $1.3\times10^2$ | $1.3\times10^2$ |
| | Succinyl-CoA ↔ Succinate | Succinyl-CoA synthetase | suc C suc D | EC:6.2.1.5 | $1.6\times10^2$ | $9.1\times10$ | PMID:5338130 | $1.0\times10^2$ | $1.0\times10^2$ |
| | Succinate → Fumarate | Succinate dehydrogenase | sdh A sdh B[‡] | EC:1.3.5.1 | $1.0\times10^2$ | $1.1\times10^2$ | PMID:4334990; 16484232 | $1.1\times10^2$ | $1.1\times10^2$ |
| | Fumarate ↔ Malate | Fumarase | fumA[‡] | EC:4.2.1.2 | $2.0\times10^2$ | $1.2\times10^3$ | PMID:3282546; 12021453 | $4.9\times10^2$ | $4.9\times10^2$ |
| $J_4$ | Malate ↔ Oxaloacetate | Malate dehydrogenase | mdh | EC:1.1.1.37 | $6.1\times10$ | $5.5\times10^2$ | doi:https://doi.org/10.1016/0076-6879(69)13029-3 | $6.6\times10$ | $6.6\times10$ |
| $J_5$ | Phosphoenolpyruvate →Oxaloacetate | Phosphoenolpyruvate carboxylase | ppc | EC:4.1.1.31 | $4.0\times10^2$ | $1.5\times10^2$ | PMID:9927652; 4932977 | / | $1.5\times10^2$ |
| | Acetyl-CoA ↔ Acetyl phosphate | Phosphate acetyltransferase | pta | EC:2.3.1.8 | $7.7\times10$ | $3.0\times10$ | PMID:20236319 | $3.7\times10^2$ | $3.7\times10^2$ |
| | Acetyl phosphate↔ Acetate | Acetate kinase | ackA | EC:2.7.2.1 | $4.3\times10$ | $3.6\times10^3$ | EcoCyc: EG10027; PMID:24801996 | $3.3\times10^2$ | $3.3\times10^2$ |
| $J_6$ | Acetate (intracellular) ↔ Acetate (extracellular) | Acetate transporter | actP | / | $2\times10$ | $4.7\times10^2$ | PMID:31405984 (Estimated) | / | $4.7\times10^2$ |

*Appendix 1—table 1 Continued on next page*

| No.[*] | Reaction | Enzyme | Gene name | EC | MW (kDa) | In vitro $k_{cat}$ (s⁻¹) | References | In vivo[†] $k_{cat}$ (s⁻¹) | Selected $k_{cat}$ (s⁻¹) |
|---|---|---|---|---|---|---|---|---|---|
| $J_7$ | Pyruvate → Phosphoenolpyruvate | Pyruvate, water dikinase | ppsA | EC:2.7.9.2 | $2.5×10^2$ | $3.5×10$ | PMID:4319237 | / | $3.5×10$ |
| | Glucose-6P (extracellular) → Glucose-6P (intracellular) | Glucose-6-phosphate transporter | UhpT | / | $5×10$ | $2×10^2$ | PMID:3283129; 2197272; 20018695 (Estimated) | / | $2×10^2$ |
| | Glucose (extracellular) → Glucose-6P | Glucose-specific PTS enzyme | ptsG | EC: 2.7.1.199 | $5×10$ | $1×10^2$ | PMID:9575173; 20018695; 12146972 | / | $1×10^2$ |
| | Lactose (extracellular) → Lactose (intracellular) | Lactose transporter | lacY | / | $4.6×10$ | $6×10$ | PMID:6444453; 20018695 | / | $6×10$ |
| $J_A$ | Lactose →Glucose +Galactose | β-galactosidase | lacZ | EC:3.2.1.23 | $4.6×10^2$ | $6.4×10^2$ | PMID:8008071; 23011886 (Estimated) | / | $6.4×10^2$ |
| $J_{py}$ | Pyruvate (extracellular) → Pyruvate (intracellular) | Pyruvate transporter | btsT CstA | / | $8×10$ | $6×10$ | PMID:20018695; 33260635; EcoCyc: G7942; EG10167 (Estimated) | / | $6×10$ |

[*]The classification of $J_i$ follows the coarse-grained models shown in **Figures 1B and 3E**.

[†]In vivo $k_{cat}$ values were obtained using the experimental data shown in **Appendix 1—table 2**, combined with **Equations S134-S135**.

[‡]See **Appendix 1—figure 1B** for additional genes that may play a secondary role.

**Appendix 1—table 2.** Proteome and flux data (**Basan et al., 2015**) used to calculate the in vivo $k_{cat}$ of *E. coli*.

| | Culture 1 | Culture 2 | Culture 3 | Culture 4 |
|---|---|---|---|---|
| Growth rate $\lambda$ (h⁻¹)[*] | 0.82 | 0.87 | 0.97 | 1.03 |
| $J_{acetate}$ (mM OD$_{600}$⁻¹ h⁻¹)[†] | 0.39 | 1.18 | 2.68 | 2.84 |
| $J_{CO2, r}$ (mM OD$_{600}$⁻¹ h⁻¹) [†] | 7.44 | 6.05 | 4.30 | 3.04 |
| Gene name | Proteomic mass fractions obtained using absolute abundance ($\phi_i$) | | | |
| pgi | 0.09% | 0.09% | 0.10% | 0.11% |
| pfkA | 0.06% | 0.06% | 0.06% | 0.06% |
| fbaA | 0.32% | 0.35% | 0.35% | 0.39% |
| tpiA | 0.12% | 0.15% | 0.13% | 0.18% |
| gapA | 1.19% | 1.29% | 1.33% | 1.47% |
| pgk | 0.30% | 0.31% | 0.32% | 0.36% |
| gpmA | 0.15% | 0.15% | 0.15% | 0.16% |
| eno | 0.63% | 0.70% | 0.75% | 0.83% |
| pykF | 0.15% | 0.15% | 0.18% | 0.21% |
| aceE | 0.30% | 0.32% | 0.34% | 0.41% |
| gltA | 0.88% | 0.80% | 0.61% | 0.48% |
| acnB | 0.92% | 0.84% | 0.66% | 0.57% |
| icd | 1.55% | 1.55% | 1.31% | 1.39% |
| suc A suc B | 0.71% | 0.75% | 0.64% | 0.55% |
| suc C suc D | 0.88% | 0.84% | 0.66% | 0.52% |
| sdh A sdh B | 0.49% | 0.45% | 0.42% | 0.35% |
| fumA | 0.24% | 0.21% | 0.17% | 0.13% |
| mdh | 0.45% | 0.45% | 0.41% | 0.39% |

*Appendix 1—table 2 Continued on next page*

Appendix 1—table 2 Continued

|  | Culture 1 | Culture 2 | Culture 3 | Culture 4 |
|---|---|---|---|---|
| pta | 0.10% | 0.10% | 0.10% | 0.10% |
| ackA | 0.06% | 0.07% | 0.06% | 0.06% |

*For calibration purposes, a factor of 1.03/0.97 was multiplied by the reference data (**Basan et al., 2015**)[‡].

[†]For calibration purposes, a factor of 2.84/3.24 was multiplied by the reference data (**Basan et al., 2015**)[‡].

[‡]Here, (1.03, 2.84) and (0.97, 3.24) are both the data points for ($\lambda$ h$^{-1}$, $J_{acetate}$ mM OD$_{600}^{-1}$ h$^{-1}$) for *E. coli* strain NCM3722 cultured with lactose in the same reference (**Basan et al., 2015**). The former is specified in the source data of the reference's figure 1 (**Basan et al., 2015**), while the latter is recorded in the reference's extended data figure 3a (**Basan et al., 2015**). With the calibrations above, the data for the $J_{acetate}^{(M)} - \lambda$ relation shown here align with the curve depicted in **Figure 1C**.

**Appendix 1—table 3.** Illustrations of symbols in this manuscript.

| Symbols | Illustrations/Definitions | Model variable/parameter settings for *E. coli*[*] |
|---|---|---|
| *A* (in the figures) | A Group *A* carbon source joining the metabolic network from the upper part of glycolysis. | NA [†] |
| $M_i$ (in the figures) | A metabolite in the metabolic network that serve as intermediate node. | NA |
| $J_i$ (in the figures) | The stoichiometric flux delivering carbon flux, an extensive variable‡; see **Equation S7**. | see **Equations S7-S8**. |
| $r_i$ (in the figures) | The mass fraction of carbon flux drawn from a precursor pool. | $r_{a1}$=24%; $r_{a2}$=24%; $r_b$ = 28%; $r_c$ = 12%; $r_d$ = 12% (**Nelson and Cox, 2008**). |
| $\lambda$ | Growth rate of the cell population; see **Equation S36** for the optimal model solution. | see **Equations S4 and S36**. |
| $J_r$, $J_f$ | $J_r$ and $J_f$ are stoichiometric fluxes of respiration and fermentation, extensive variables. | $J_r = J_4$; $J_f = J_6$ (see **Equation S22**) |
| $m_0$ | The weighted average carbon mass of metabolite molecules at the entrance of precursor pools. | See **Equation S17**. |
| $M_{carbon}$ | The carbon mass of the cell population, an extensive variable. | NA |
| $M_{protein}$ | The protein mass of the cell population; an extensive variable. | NA |
| $M_Q^{(P)}$, $M_R^{(P)}$, $M_C^{(P)}$ | The mass of Q-class, R-class, or C-class proteome. | See **Equation S2**. |
| $f_Q$, $f_R$, $f_C$ | The ribosome allocation fraction for protein synthesis of Q-class, R-class, or C-class. | $f_Q = \phi_Q$. |
| $m_{AA}$ | The average molecular weight of amino acids. | A reducible parameter for the results. |
| $k_T$ | Translation speed of ribosomes. | $k_T$=20.1 aa/s (**Scott et al., 2010**). |
| $\phi_Q$, $\phi_R$, $\phi_C$ | The mass fraction of Q-class, R-class, or C-class proteome; see Appendix 2.1. | $\phi_Q$=52% (**Scott et al., 2010**). |
| $\phi_{max}$ | The maximum proteomic mass fraction of proteome allocation for fermentation, respiration, and biomass generation, with $\phi_{max} \equiv 1 - \phi_Q$. | $\phi_{max}$=48% (**Scott et al., 2010**). |

*Appendix 1—table 3 Continued on next page*

*Appendix 1—table 3 Continued*

| Symbols | Illustrations/Definitions | Model variable/parameter settings for *E. coli*[*] |
|---|---|---|
| $m_R$ | The protein mass of a single ribosome. | $m_R = 7336\, m_{AA}$ (**Neidhardt et al., 1990**). |
| $V_{cell}$ | The cell volume of the cell population (the 'big cell'); an extensive variable. | NA |
| $N_R$, $M_{rp}^{(P)}$ | The number or the total protein mass of ribosomes in the big cell; extensive variables. | NA |
| $\varsigma$ | The ratio of the mass of R-class proteome to the protein mass of ribosomes: $\varsigma \equiv M_R^{(P)}/M_{rp}^{(P)}$. | $\varsigma = 1.67$ (**Scott et al., 2010**). |
| $[E_i]$, $[S_i]$ | The concentration of enzyme $E_i$ or substrate $S_i$; intensive variables. | NA |
| $a_i$, $d_i$, $b_i$, $c_i$ | $a_i$ and $d_i$ are reaction parameters; $b_i$ and $c_i$ are stoichiometric coefficients. See Appendix 2.3. | NA |
| $K_i$ | The Michaelis constant, defined as $K_i \equiv (d_i + k_i^{cat})/a_i$. | Obtainable from **Bennett et al., 2009**, yet unused in practice since $[Si] > K_i$ (see Appendix 2.5). |
| $v_i$ | The reaction rate per volume of a biochemical reaction catalyzed by $E_i$; an intensive variable. | See **Equation S6**. |
| $N_{E_i}$, $M_{E_i}$ | The copy number or the total weight enzyme $E_i$ in the cell population; extensive variables. | $N_{E_i} = V_{cell} \cdot [E_i]$; $M_{E_i} = N_{E_i} \cdot m_{E_i}$. |
| $m_{carbon}$ | The mass of a carbon atom. | $m_{carbon} = \frac{12}{N_{Avogadro}}$ g, where g represents gram and $N_{Avogadro}$ is the Avogadro constant. |
| $\Phi_i$ | The enzyme cost of all $E_i$ molecules in the cell population; an extensive variable. | $\Phi_i \equiv N_{E_i} \cdot n_{E_i}$. |
| $\xi_i$ | $\xi_i$ is defined such that $\xi_i = J_i/\Phi_i$. | $\xi_i \equiv \frac{k_i^{cat}}{n_{E_i}} \cdot \frac{[S_i]}{[S_i]+K_i}$. |
| $J_i^{(N)}$ | The normalized flux, i.e., flux per unit of biomass; an intensive variable[§] | $J_i^{(N)} \equiv J_i \cdot m_0/M_{carbon}$ see **Equations S15-S16**. |
| $J_r^{(N)}$, $J_f^{(N)}$ | $J_r^{(N)}$ and $J_f^{(N)}$ are the normalized fluxes of respiration and fermentation, intensive variables. | $J_r^{(N)} = J_4^{(N)}$; $J_f^{(N)} = J_6^{(N)}$. |
| $N_{EP_i}^{carbon}$ | The number of carbon atoms in the entry point metabolite molecule of Precursor Pool $i$. | $N_{EP_{a1}}^{carbon} = 6$; $N_{EP_{a2}}^{carbon} = 3$; $N_{EP_b}^{carbon} = 3$; $N_{EP_c}^{carbon} = 5$; $N_{EP_d}^{carbon} = 4$ (**Nelson and Cox, 2008**). |
| $k_{cat}$, $k_i^{cat}$ | The turnover number of a catalytic enzyme. | See **Appendix 1—table 1**. |
| $m_{E_i}$, $n_{E_i}$ | $m_{E_i}$ and $n_{E_i}$ are the molecular weight and the enzyme cost of an $E_i$ molecule, respectively. | See **Appendix 1—table 1**. |

*Appendix 1—table 3 Continued on next page*

*Appendix 1—table 3 Continued*

| Symbols | Illustrations/Definitions | Model variable/parameter settings for *E. coli*[*] |
|---|---|---|
| $r_{\mathbf{carbon}}$, $r_{\mathbf{protein}}$ | $r_{carbon}$ and $r_{protein}$ are the mass fractions of all carbon and protein within a cell, respectively. | $r_{protein}=0.55$; $r_{carbon}=0.48$ (***Neidhardt et al., 1990***). |
| $\kappa_i$ | Substrate quality of a metabolite in a biochemical reaction; see ***Equation S12 and S20***. | Calculated from the values of $k_i^{cat}$, $m_{E_i}$, $m_0$, $r_{protein}$, $r_{carbon}$. |
| $\kappa_A$ | Substrate quality of a Group A carbon source; see ***Equation S27***. | Calculated from the values of $k_A^{cat}$, $m_{E_A}$, $m_0$, $r_{protein}$, $r_{carbon}$, $K_A$ and the concentration of the Group A carbon source [A]. |
| $\phi_i$ | The proteomic mass fraction of enzyme $E_i$: $\phi_i \equiv M_{E_i}/M_{protein}$; an intensive variable. | See ***Equation S9***. |
| $\eta_i$ | The fraction of stoichiometric flux drawn from a precursor pool; see ***Equations S13, S14 and S18***. | $\eta_{a1}=15\%$; $\eta_{a2}=30\%$; $\eta_b=35\%$; $\eta_c=9\%$; $\eta_d=11\%$ (calculated from the values of $r_i$ and $N_{EP_i}^{carbon}$). |
| $\phi_r$, $\phi_f$, $\phi_{\mathbf{BM}}$ | $\phi_f$, $\phi_f$, $\phi_{BM}$ are the proteomic mass fraction of enzymes dedicated to fermentation, respiration, and biomass generation, respectively. | NA |
| $\kappa_{\mathbf{t}}$ | A parameter determined by the translation rate, defined as $\kappa_t \equiv k_T \cdot m_{AA}/\left(\varsigma \cdot m_R\right)$. | $\kappa_t = 1/610$ (s$^{-1}$) (calculated from the values of $k_T$, $\varsigma$ and $m_R$). |
| $J_{BM}$ | The carbon flux of biomass production; an extensive variable. | See ***Equation S10***. |
| $J_E$ | The energy demand for cell growth, expressed as the stoichiometric energy flux in ATP; an extensive variable. | See ***Equation S25***. |
| $J_{\mathbf{E}}^{(\mathbf{N})}$ | The normalized flux of energy demand in ATP; an intensive variable. | $J_E^{(N)} \equiv J_E \cdot m_0/M_{carbon}$. |
| $r_{\mathrm{E}}$, $\eta_{\mathbf{E}}$ | $r_E$ and $\eta_E$ are energy coefficients. $r_E$ is the slope of $J_E$ versus $J_{BM}$; $$\eta_E = r_E \cdot \left[\sum_i r_i/N_{EP_i}^{carbon}\right].$$ | See Appendix 9.2. |
| $\beta_i$ | The stoichiometric coefficient of ATPs in biochemical reactions shown in ***Figures 1B and 3E*** (for *E. coli*) or ***Appendix 1—figure 5E and F*** (for yeast and mammalian cells). | $\beta_1 = 4$, $\beta_2 = 3$, $\beta_3 = 2$, $\beta_4 = 6$, $\beta_6 = 1$, $\beta_{a1} = 4$, $\beta_7 = 1$, $\beta_8 = 2$, $\beta_9 = 6$ (*E. coli*); $\beta_1 = 5$, $\beta_2 = 1$, $\beta_3 = 5$, $\beta_4 = 7.5$, $\beta_6 = 2.5$, $\beta_{a1} = 5$ (eukaryotic cells) (***Neidhardt et al., 1990***; ***Sauer et al., 2004***). |
| $\beta_r^{(A)}$, $\beta_f^{(A)}$ | $\beta_r^{(A)}$ and $\beta_f^{(A)}$ are the stoichiometric coefficients of ATP production per glucose in respiration and fermentation, respectively. | $\beta_r^{(A)} = 26$, $\beta_f^{(A)} = 12$ (*E. coli*); $\beta_r^{(A)} = 32$, $\beta_f^{(A)} = 2$ (eukaryotic cells) (***Neidhardt et al., 1990***). |
| $J_r^{(\mathbf{E})}$, $J_f^{(\mathbf{E})}$ | $J_r^{(E)}$ and $J_f^{(E)}$ are normalized energy fluxes of respiration and fermentation, intensive variables. | $J_r^{(E)} \equiv \frac{\beta_r^{(A)}}{2} \cdot J_r^{(N)}$; $J_f^{(E)} \equiv \frac{\beta_f^{(A)}}{2} \cdot J_f^{(N)}$. |

*Appendix 1—table 3 Continued on next page*

*Appendix 1—table 3 Continued*

| Symbols | Illustrations/Definitions | Model variable/parameter settings for *E. coli*[*] |
|---|---|---|
| $\varepsilon_r$, $\varepsilon_f$ <br> $\varepsilon_r^{(dt)}$, $\varepsilon_f^{(dt)}$ | $\varepsilon_r$ (or $\varepsilon_r^{(dt)}$) and $\varepsilon_f$ (or $\varepsilon_f^{(dt)}$) are the proteome efficiencies for energy biogenesis in the respiration and fermentation pathways: <br> $\varepsilon_r \equiv J_r^{(E)} / \phi_r$ and $\varepsilon_f \equiv J_f^{(E)} / \phi_f$. | Calculated from the values of $\kappa_A$, $\kappa_i$, $\beta_r^{(A)}$ and $\beta_f^{(A)}$ with **Equations S132 and S161**. |
| $\varphi$ | $\varphi$ is an energy demand coefficient, defined in **Equation S33** and mainly determined by $\eta_E$. | Calculated from the values of $\eta_E$, $\beta_i$, $\eta_i$ with **Equation S33**. See Appendix 9.2. |
| $\psi$, $\psi_{dt}$ | $\psi^{-1}$ (or $\psi_{dt}^{-1}$) is the proteome efficiency for biomass generation in the biomass pathway, with <br> $\psi^{-1} \equiv \big/ \lambda/\phi_{BM}$. | Calculated from the values of $\eta_i$, $\kappa_A$, $\kappa_i$, $\Omega$, $\kappa_t$ with **Equations S133 and S162**. |
| $\kappa_r^{(A)}$, $\kappa_f^{(A)}$ | $\kappa_r^{(A)}$ and $\kappa_f^{(A)}$ are parameters defined as <br> $\kappa_r^{(A)} \equiv \left[\frac{1}{\kappa_1} + \frac{2}{\kappa_2} + \frac{2}{\kappa_3} + \frac{2}{\kappa_4}\right]^{-1}$ and <br> $\kappa_f^{(A)} \equiv \left[\frac{1}{\kappa_1} + \frac{2}{\kappa_2} + \frac{2}{\kappa_6}\right]^{-1}$. | Calculated from the values of $\kappa_i$. |
| $\Omega$ | $\Omega$ is a composite parameter defined as $\Omega \equiv 1/\kappa_t + \sum_{i}^{a1,a2,b,c,d} \eta_i/\kappa_i$. | See Appendix 9.2. |
| $\kappa_{glucose}^{(ST)}$, $\kappa_{lactose}^{(ST)}$ | The substrate quality of glucose or lactose at saturated concentration. | Calculated using **Equation S27** and the approximation used in **Equation S20**. |
| $\Delta$ | $\Delta$ is a function of $\kappa_A$ defined as $\Delta(\kappa_A) \equiv \varepsilon_f(\kappa_A) / \varepsilon_r(\kappa_A)$. | $\Delta \equiv \varepsilon_f/\varepsilon_r$. |
| $\kappa_A^{(C)}$ | The critical value of $\kappa_A$ which satisfy $\Delta(\kappa_A) = 1$ and thus $\varepsilon_f(\kappa_A) = \varepsilon_r(\kappa_A)$; See **Equation S42** (for *E. coli*) and S176 (for yeast and mammalian cells). | Calculated from the values of $\beta_i$ and $\kappa_i$ with **Equation S42**. |
| $\lambda_C$ | The critical growth rate at the transition point: $\lambda_C \equiv \lambda\left(\kappa_A^{(C)}\right)$; See **Equations S43 and S177**. | Calculated from the values of $\phi_{max}$, $\varphi$, $\beta_i$, $\kappa_i$, $\kappa_A^{(C)}$, $\Omega$, $\eta_i$ with **Equations S43, S32 and S162**. |
| $\theta$ | The Heaviside step function. | NA |
| $J_{acetate}$, $J_{CO_2,r}$ | $J_{acetate}$ and $J_{CO_2,r}$ are the stoichiometric fluxes of acetate from the fermentation pathway and CO2 from the respiration pathway; extensive variables. | $J_{acetate} = J_f$; $J_{CO_2,r} = 3 \cdot J_r$. See Appendix 9.1 and **Equations S158**. |
| $J_{acetate}^{(M)}$, $J_{CO_2,r}^{(M)}$ | $J_{acetate}^{(M)}$ and $J_{CO_2,r}^{(M)}$ are the fluxes of $J_{acetate}$ and $J_{CO_2,r}$ (per biomass) in the unit of mM/OD600/h, which are measurable in experiment. Intensive variables. | $J_{acetate}^{(M)} \approx 2 \cdot J_f^{(N)}$; $J_{CO_2,r}^{(M)} \approx 6 \cdot J_r^{(N)}$. See Appendix 9.1 and **Equation S160**. |

*Appendix 1—table 3 Continued on next page*

*Appendix 1—table 3 Continued*

| Symbols | Illustrations/Definitions | Model variable/parameter settings for *E. coli*[*] |
|---|---|---|
| $\kappa_A^{\max}$ | The maximum value of $\kappa_A$ available across different Group A carbon sources. | Approximated by the max $\kappa_A$ across Group A carbon sources, calculated with **Equation S27** and the approximation used in **Equation S20**. |
| $\lambda_{\max}$ | The population cell growth rate for the maximum value of $\kappa_A$: $\lambda_{\max} = \lambda\left(\kappa_A^{\max}\right)$. | Calculated from the maximum of **Equation S36** with the values of $\beta_i$, $\kappa_i$, $\kappa_A^{\max}$, $\varphi$, $\Omega$, $\kappa_t$, and **Equations S32, S132, Equation S161 and S162**. |
| $\mathcal{N}\left(\mu, \sigma^2\right)$ | A Gaussian distribution with a mean of μ and a standard deviation of $\sigma$. | The probability density function is $f(x) = \frac{1}{\sigma\sqrt{2\pi}}e^{-\frac{1}{2}\left(\frac{x-\mu}{\sigma}\right)^2}$. |
| $\mu_{\lambda_C}, \sigma_{\lambda_C}$ | $\mu_{\lambda_C}$ and $\sigma_{\lambda_C}$ are the mean and standard deviation of $\lambda_C$, respectively. | $\mu_{\lambda_C}$ is approximated by the deterministic value of $\lambda_C$; see Appendix 3.3 for $\sigma_{\lambda_C}$ settings. See Appendix 9.2 for the values. |
| erf | The error function in mathematics. | $\mathrm{erf}(x) = \frac{2}{\sqrt{\pi}} \int_0^x \exp\left(-t^2\right) dt$ |
| $\phi_Z$ | The proteomic mass fraction of useless proteins encoded by the LacZ gene. | See Appendix 4.1. |
| $w$ | An energy dissipation coefficient. | See Appendix 4.2. |
| $w_0$ | The maintenance energy coefficient. | $w_0$=0 or 2.5 ($h^{-1}$) as specified in **Figures 3–4, Appendix 1—figures 2 and 3**. See Appendices 4.3 and 9.2. |
| $\iota$ | $\iota$ is the inhibition coefficient such that $\left(1+\iota\right)^{-1}$ represents the translation efficiency. | See Appendices 4.3 and 9.2 |
| $\iota_{w_0=0}^{(2\mu m\,Cm)}, \iota_{w_0=0}^{(4\mu m\,Cm)}, \iota_{w_0=0}^{(8\mu m\,Cm)}, \iota_{w_0=2.5}^{(2\mu m\,Cm)}, \iota_{w_0=2.5}^{(4\mu m\,Cm)}, \iota_{w_0=2.5}^{(8\mu m\,Cm)}$ | The values for $\iota$ in the cases with 2 μm , 4 μm, or 8 μm of chloramphenicol and the maintenance energy coefficient $w_0$ chosen as 0 or 2.5 ($h^{-1}$). | $\iota_{w_0=0}^{(2\mu m\,Cm)} = 1.15; \iota_{w_0=0}^{(4\mu m\,Cm)} = 2.33; \iota_{w_0=0}^{(8\mu m\,Cm)} = 6.25; \iota_{w_0=2.5}^{(2\mu m\,Cm)} = 1.05; \iota_{w_0=2.5}^{(4\mu m\,Cm)} = 2.00; \iota_{w_0=2.5}^{(8\mu m\,Cm)} = 5.40$. See Appendix 9.2. |
| $\kappa_{py}$ | The substrate quality of pyruvate; see **Equation S89**. | Calculated from the values of $k_{py}^{cat}$, $m_{E_{py}}$, $m_0$, $r_{protein}$, $r_{carbon}$, $K_{py}$ and the external concentration of pyruvate [py]. |
| $\beta_r^{(py)}, \beta_f^{(py)}$ | $\beta_r^{(py)}$ and $\beta_f^{(py)}$ are the stoichiometric coefficients of ATP production per pyruvate in respiration and fermentation, respectively. | $\beta_r^{(py)} = 10; \beta_f^{(py)} = 3$. (**Neidhardt et al., 1990**). |
| $J_r^{(E,py)}, J_f^{(E,py)}$ | $J_r^{(E,py)}$ and $J_f^{(E,py)}$ are the normalized energy fluxes of respiration and fermentation for pyruvate utilization; intensive variables. | The corresponding variables of $J_r^{(E)}$ and $J_f^{(E)}$ in the case of pyruvate utilization. |

*Appendix 1—table 3 Continued on next page*

Appendix 1—table 3 Continued

| Symbols | Illustrations/Definitions | Model variable/parameter settings for *E. coli** |
|---|---|---|
| $\varepsilon_r^{(py)}, \varepsilon_f^{(py)}$ | $\varepsilon_r^{(py)}$ and $\varepsilon_f^{(py)}$ are the proteome efficiencies for energy biogenesis using pyruvate in the respiration and fermentation pathways. | The corresponding variables of $\varepsilon_r$ and $\varepsilon_f$ in the case of pyruvate utilization. |
| $\Omega'_{Gg}$ | $\Omega'_{Gg}$ is a composite parameter defined as $\Omega'_{Gg} \equiv \left(\eta_b + \eta_c\right)/\kappa_8 + \eta_{a1}/\kappa_9.$ | See Appendix 9.2. |
| $\psi_{py}, \varphi_{py}, \kappa_{py}^{(ST)}, \kappa_{py}^{(C)}, \lambda_{max}^{(py)}$ | $\psi_{py}, \varphi_{py}, \kappa_{py}^{(ST)}, \kappa_{py}^{(C)}$ and $\lambda_{max}^{(py)}$ are the corresponding variables/ parameters of $\psi, \varphi, \kappa_A^{max}, \kappa_A^{(C)}$ and $\lambda_{max}$ in the case of pyruvate utilization. | See Appendices 5.1 and 9.2. |
| $\lambda_C^{(py)}, \mu_{\lambda_C^{(py)}}, \sigma_{\lambda_C^{(py)}}$ | $\lambda_C^{(py)}, \mu_{\lambda_C^{(py)}}$ and $\sigma_{\lambda_C^{(py)}}$ are the corresponding variables/parameters of $\lambda_C, \mu_{\lambda_C}$ and $\sigma_{\lambda_C}$ in the case of pyruvate utilization. | See Appendices 5.1 and 9.2. |
| $N_{P_i}^{carbon}$ | The number of carbon atoms in a molecule of Pool $i$. | The value of $N_{P_i}^{carbon}$ is approximated by $N_{EP_i}^{carbon}$ (*Equation S107*). |
| $\kappa_i^{(21AA)}$ | The substrate quality of the external supplied amino acids identical to those in Pool $i$. | See Appendices 5.2 and 9.2. |
| $\Omega_{21AA}$ | $\Omega_{21AA}$ is a composite parameter defined as $\Omega_{21AA} \equiv 1/\kappa_t + \eta_{a1}/\kappa_{a1} + \sum_{i}^{a2,b,c,d} \eta_i/\kappa_i^{(21AA)}$. | See Appendices 5.2 and 9.2. |
| $\psi_{21AA}, \varphi_{21AA}, \lambda_{max}^{(21AA)}, \lambda_C^{(21AA)}, \mu_{\lambda_C^{(21AA)}}, \sigma_{\lambda_C^{(21AA)}}$ | $\psi_{21AA}, \varphi_{21AA}, \lambda_{max}^{(21AA)}, \lambda_C^{(21AA)}, \mu_{\lambda_C^{(21AA)}}$ and $\sigma_{\lambda_C^{(21AA)}}$ are the corresponding variables/ parameters of $\psi, \varphi, \lambda_{max}, \lambda_C, \mu_{\lambda_C}$ and $\sigma_{\lambda_C}$ in the case of a Group A carbon source is mixed with 21 types of amino acids at saturated concentrations. | See Appendices 5.2 and 9.2. |
| $\Omega_{7AA}, \varphi_{7AA}, \mu_{\lambda_C^{(7AA)}}, \sigma_{\lambda_C^{(7AA)}}$ | $\Omega_{7AA}, \varphi_{7AA}, \mu_{\lambda_C^{(7AA)}}$ and $\sigma_{\lambda_C^{(7AA)}}$ are the corresponding parameters of $\Omega, \varphi, \mu_{\lambda_C}$ and $\sigma_{\lambda_C}$ in the case of a Group A carbon source is mixed with 7 types of amino acids. | See Appendices 5.2 and 9.2. |
| $J_{in}^{(N)}, \vartheta$ | $J_{in}^{(N)}$ is the normalized stoichiometric influx of a Group A carbon source (*Equation S136*). $\vartheta$ is a parameter defined as $\vartheta = \eta_{a1} + \eta_c + \left(\eta_{a2} + \eta_b + \eta_d\right)/2$ for the model shown in *Figure 1B*. | See Appendix 7.3 |

Appendix 1—table 3 Continued on next page

*Appendix 1—table 3 Continued*

| Symbols | Illustrations/Definitions | Model variable/parameter settings for *E. coli** |
|---|---|---|
| $\chi_{\text{ext}}$, $\chi_{\text{int}}$, $\chi_{\text{tot}}$ | $\chi_{\text{ext}}$, $\chi_{\text{int}}$ and $\chi_{\text{tot}}$ are the level of extrinsic noise, intrinsic noise and total noise in a system. | See Appendix 8.1 |
| $\mu_{k_i^{\text{cat}}}$, $\sigma_{k_i^{\text{cat}}}$, $\mu_{1/k_i^{\text{cat}}}$, $\sigma_{1/k_i^{\text{cat}}}$, $\mu'_{1/k_i^{\text{cat}}}$, $\sigma'_{1/k_i^{\text{cat}}}$ | $\mu_{k_i^{\text{cat}}}$ and $\sigma_{k_i^{\text{cat}}}$ are the mean and standard deviation of $k_i^{\text{cat}}$. $\mu_{1/k_i^{\text{cat}}}$ (or $\mu'_{1/k_i^{\text{cat}}}$) and $\sigma_{1/k_i^{\text{cat}}}$ (or $\sigma'_{1/k_i^{\text{cat}}}$) are the mean and standard deviation of $1/k_i^{\text{cat}}$. See Appendix 8.1. | $\mu_{k_i^{\text{cat}}}$ is approximated by the deterministic value of $k_i^{\text{cat}}$. The CV of $k_i^{\text{cat}}$ is set to 25%. $\mu_{1/k_i^{\text{cat}}} \approx 1/\mu_{k_i^{\text{cat}}}$; $\sigma_{1/k_i^{\text{cat}}}/\mu_{1/k_i^{\text{cat}}} \approx \sigma_{k_i^{\text{cat}}}/\mu_{k_i^{\text{cat}}}$. |
| $\text{IG}\left(x; \mu, \zeta\right)$ | The inverse Gaussian (IG) distribution: variable $x>0$ with parameters $\mu$ and $\zeta$. See **Equation S142**. | The probability density function is $\sqrt{\dfrac{\zeta}{2\pi x^3}} \exp\left(-\dfrac{\zeta(x-\mu)^2}{2\mu^2 x}\right)$. |
| $\text{IOG}\left(x; \mu, \zeta\right)$ | The positive inverse of Gaussian (IOG) distribution: variable $x>0$ with parameters $\mu$ and $\zeta$. See **Equation S140** and Appendix 8.1. | The probability density function is $\sqrt{\dfrac{\zeta}{2\pi x^4}} \exp\left(-\dfrac{\zeta(x-\mu)^2}{2\mu^2 x^2}\right)$. |
| $\zeta_{1/k_i^{\text{cat}}}$, $\zeta'_{1/k_i^{\text{cat}}}$ | Distributional parameters of $1/k_i^{\text{cat}}$ corresponding to $\zeta$ in an IG or IOG distribution. | See Appendix 8.1 |
| $G\left(k\right)$ | The characteristic function of IG distribution. See **Equation S147**. | $G\left(k\right) = \displaystyle\int_{-\infty}^{\infty} e^{ikx} \cdot IG\left(x; \mu, \zeta\right) dx$ |
| $X_i$, $\alpha_i$, $\Theta$, $T_\Theta$, $\Gamma_i\left(t\right)$ | $X_i$, $\alpha_i$, $\Theta$ and $\Gamma_i\left(t\right)$ are variables and parameters used to calculate the first passage time $T_\Theta$ of a stochastic process that mimics the duration of an enzyme to finishing a catalytic job. | See Appendix 8.1. |
| $\gamma_i$, $\Xi$, $\mu_\Xi$, $\sigma_\Xi$ | $\gamma_i$ is a real number; $\Xi$ is a variable defined as $\Xi \equiv \displaystyle\sum_{i=1}^{n} \gamma_i/k_i^{\text{cat}}$; $\mu_\Xi$ and $\sigma_\Xi$ are the mean and standard deviation of $\Xi$. | See **Equation S153** and Appendix 8.1. |
| $\mu_{\kappa_i}$, $\sigma_{\kappa_i}$, $\mu_{1/\kappa_i}$, $\sigma_{1/\kappa_i}$ | $\mu_{\kappa_i}$ and $\sigma_{\kappa_i}$ are the mean and standard deviation of $\kappa_i$; $\mu_{1/\kappa_i}$ and $\sigma_{1/\kappa_i}$ are the mean and standard deviation of $1/\kappa_i$. | See **Equation S154** and Appendices 8.1 and 9.2. |
| $\lambda_r$, $\lambda_f$, $\mu_{\lambda_r}$, $\sigma_{\lambda_r}$, $\mu_{\lambda_f}$, $\sigma_{\lambda_f}$, $\rho_{rf}$ | $\lambda_r$ and $\lambda_f$ are the growth rates when cells choose respiration or fermentation; $\mu_{\lambda_r}$, $\mu_{\lambda_f}$ and $\sigma_{\lambda_r}$, $\sigma_{\lambda_f}$ are the means and standard deviations of $\lambda_r$ and $\lambda_f$; $\rho_{rf}$ is the correlation of $\lambda_r$ and $\lambda_f$. | See **Equation S36** and Appendices 8.1 and 9.2. |
| $\lambda_{\text{succinate}}^{(21\text{AA})}$, $\lambda_{\text{acetate}}$, $\mu_{\lambda_{\text{succinate}}^{(21\text{AA})}}$, $\mu_{\lambda_{\text{acetate}}}$, $\sigma_{\lambda_{\text{succinate}}^{(21\text{AA})}}$, $\sigma_{\lambda_{\text{acetate}}}$ | $\lambda_{\text{succinate}}^{(21\text{AA})}$ and $\lambda_{\text{acetate}}$ are the growth rates for succinate mixed with 21AA or acetate as the sole carbon source; $\mu_{\lambda_{\text{succinate}}^{(21\text{AA})}}$, $\mu_{\lambda_{\text{acetate}}}$ and $\sigma_{\lambda_{\text{succinate}}^{(21\text{AA})}}$, $\sigma_{\lambda_{\text{acetate}}}$ are the means and standard deviations of $\lambda_{\text{succinate}}^{(21\text{AA})}$ and $\lambda_{\text{acetate}}$. | See Appendix 9.2. |

*Appendix 1—table 3 Continued on next page*

*Appendix 1—table 3 Continued*

| Symbols | Illustrations/Definitions | Model variable/parameter settings for *E. coli*[*] |
|---|---|---|
| $\phi_{\mathrm{MT}}$, $\kappa_{\mathrm{MT}}$ | $\phi_{\mathrm{MT}}$ and $\kappa_{\mathrm{MT}}$ are the proteomic mass fraction of the enzymes and the effective substrate quality of related metabolites in the mitochondria for yeast and mammalian cells, respectively. | NA |
| $\mathrm{Pr}_f$ | The proportion of ATP generated from fermentation: $\mathrm{Pr}_f \equiv \dfrac{J_f^{(\mathrm{E})}}{J_f^{(\mathrm{E})}+J_r^{(\mathrm{E})}}$. | See *Equations S180, S189* and Appendix 10. |
| $\bar{\Delta}$ | The proteome efficiency difference between respiration and fermentation: $\bar{\Delta} \equiv 1/\varepsilon_r - 1/\varepsilon_f$. | See *Equations S181, S187* and Appendix 10. |
| $\mu_{\varepsilon_r}$, $\mu_{\varepsilon_f}$, $\mu_{1/\varepsilon_r}$, $\mu_{1/\varepsilon_f}$ | $\mu_{\varepsilon_r}$, $\mu_{\varepsilon_f}$, $\mu_{1/\varepsilon_r}$ and $\mu_{1/\varepsilon_f}$ are the mean values of $\varepsilon_r$, $\varepsilon_f$, $1/\varepsilon_r$ and $1/\varepsilon_f$, respectively. | See *Equations S182-S184* and Appendix 10. |
| $\sigma_{\varepsilon_r}$, $\sigma_{\varepsilon_f}$, $\sigma_{1/\varepsilon_r}$, $\sigma_{1/\varepsilon_f}$ | $\sigma_{\varepsilon_r}$, $\sigma_{\varepsilon_f}$, $\sigma_{1/\varepsilon_r}$, and $\sigma_{1/\varepsilon_f}$ are the standard deviations of $\varepsilon_r$, $\varepsilon_f$, $1/\varepsilon_r$ and $1/\varepsilon_f$, respectively. | See *Equations S182, S185* and Appendix 10. |
| $\chi_{\varepsilon_r}$, $\chi_{\varepsilon_f}$, $\chi_{1/\varepsilon_r}$, $\chi_{1/\varepsilon_f}$ | $\chi_{\varepsilon_r}$, $\chi_{\varepsilon_f}$, $\chi_{1/\varepsilon_r}$, and $\chi_{1/\varepsilon_f}$ are the coefficients of variation of $\varepsilon_r$, $\varepsilon_f$, $1/\varepsilon_r$ and $1/\varepsilon_f$, respectively. | See *Equations S185-S186* and Appendix 10. |
| $\mu_{\bar{\Delta}}$, $\sigma_{\bar{\Delta}}$ | $\mu_{\bar{\Delta}}$ and $\sigma_{\bar{\Delta}}$ are the mean and standard deviation of $\bar{\Delta}$, respectively. | See *Equations S187-S188* and Appendix 10. |
| $\langle\varepsilon_r\rangle$, $\langle\varepsilon_f\rangle$ | $\langle\varepsilon_r\rangle$ and $\langle\varepsilon_f\rangle$ are the population-averaged values of $\varepsilon_r$ and $\varepsilon_f$, respectively. | Measurable from experiments. See *Equations S183-S184* and Appendix 10. |

[*]Parameter settings for yeast and mammalian cells are specifically labeled as 'eukaryotic cells.'

[†]'NA' represents 'Not applicable.'

[‡]Extensive variables scale with the size of the cell population.

[§]Intensive variables are scale-invariant with respect to the cell population.

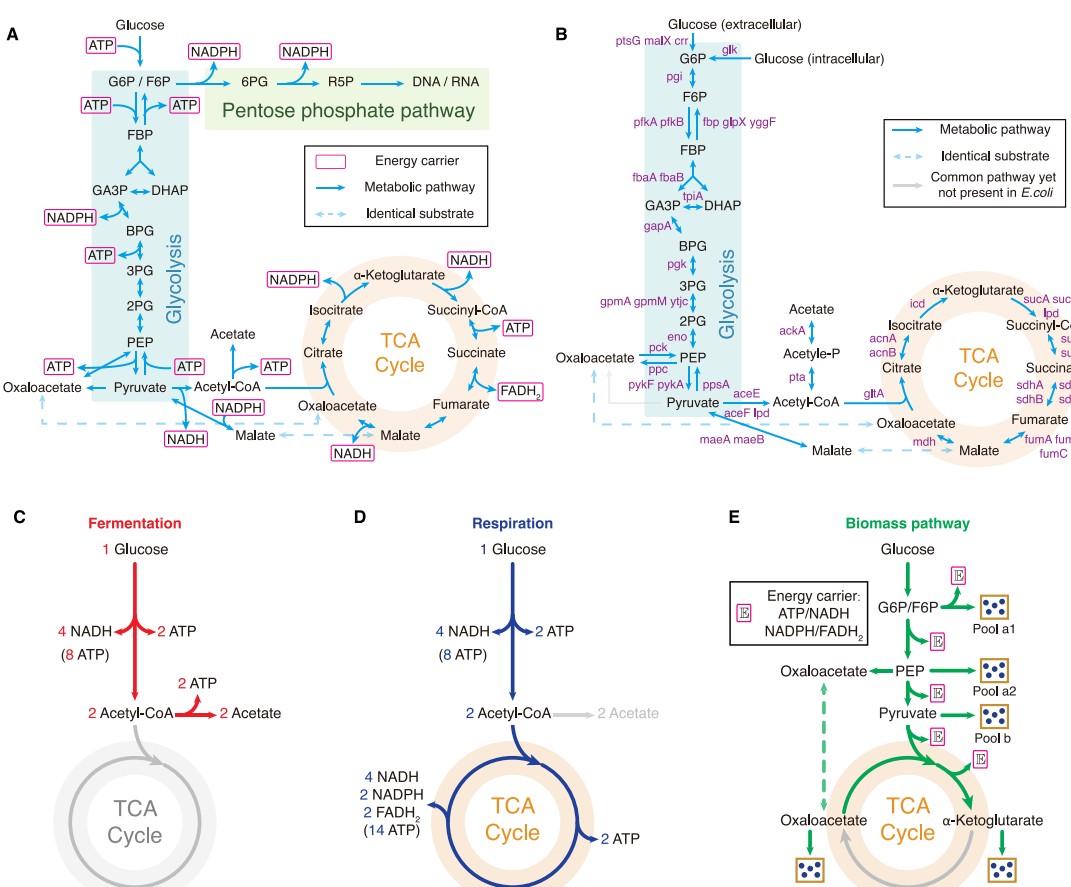

**Appendix 1—figure 1.** Central metabolic network and carbon utilization pathways of *E. coli*. (**A**) Energy biogenesis details in the central metabolic network. In *E. coli*, NADPH and NADH are interconvertible (*Sauer et al., 2004*), and all energy carriers can be converted to ATP through ADP phosphorylation. The conversion factors are: NADH = 2 ATP, NADPH = 2 ATP, FADH$_2$=1 ATP (*Neidhardt et al., 1990*). (**B**) Relevant genes encoding enzymes in the central metabolic network of *E. coli*. (**C–E**) Three independent fates of glucose metabolism in *E. coli*. (**C**) For energy biogenesis through fermentation, a molecule of glucose generates 12 ATPs. (**D**) For energy biogenesis via respiration, a molecule of glucose generates 26 ATPs. (**E**) For biomass synthesis, glucose is converted into precursors of biomass. Note that biomass synthesis is accompanied by ATP production (see Appendix 3.1).

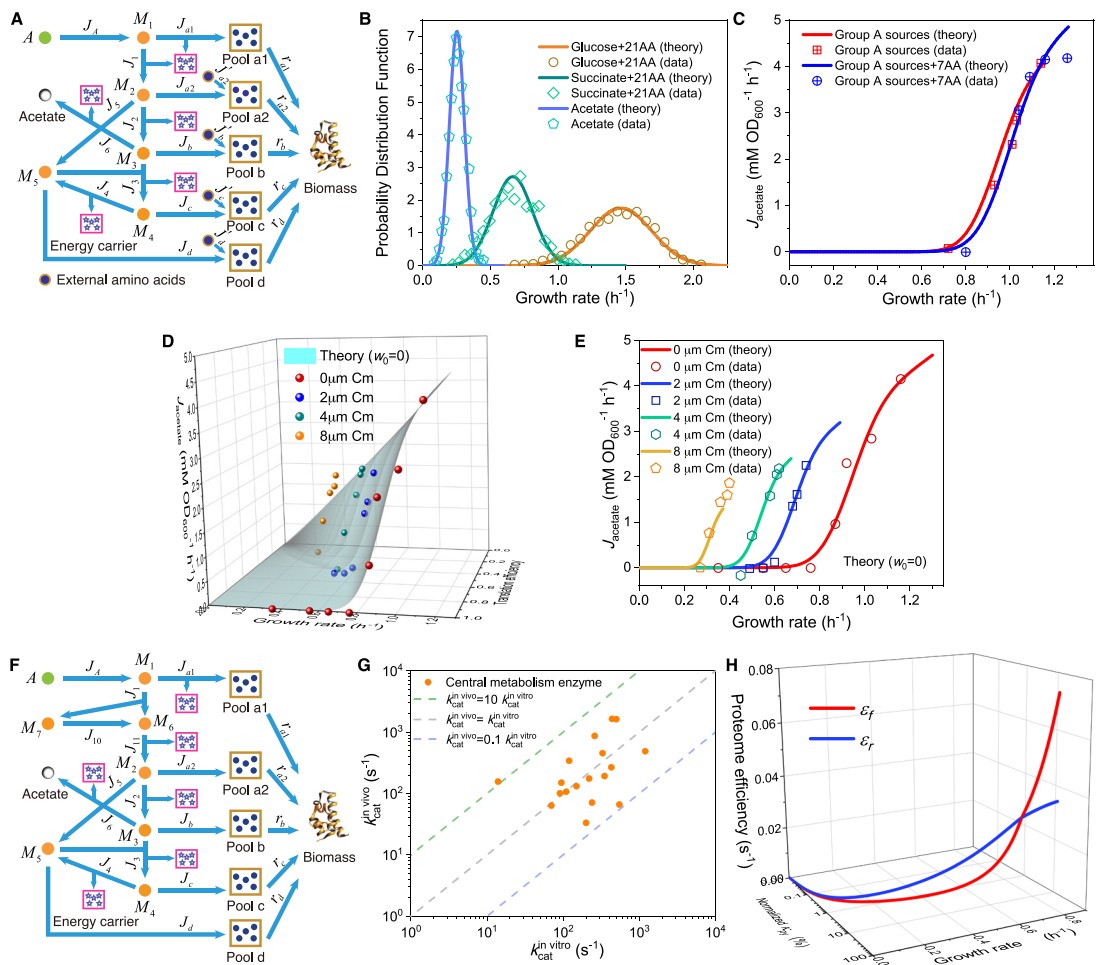

**Appendix 1—figure 2.** Model and results for experimental comparison of *E. coli*. (**A–C**) Model analysis for carbon utilization in mixtures with amino acids. (**A**) Coarse-grained model for the case of a Group A carbon source mixed with extracellular amino acids. (**B**) Model predictions (***Equations S157, S164-S165***) and single-cell reference experimental results (***Wallden et al., 2016***) showing growth rate distributions for *E. coli* in three culturing conditions. (**C**) Comparison of the growth rate-fermentation flux relation for *E. coli* in Group A carbon sources between minimal media and enriched media (those with 7AA). (**D–E**) Influence of translation inhibition on overflow metabolism in *E. coli*. (**D**) A 3D plot illustrating the relations among fermentation flux, growth rate, and translation efficiency (***Equations S79 and S160***). (**E**) Growth rate dependence of acetate excretion rate as $\kappa_A$ varies, with each fixed dose of Cm. Translation efficiency is tuned by the dose of Cm, and the maintenance energy coefficient is set to 0 (i.e. $w_0 = 0$). (**F**) Coarse-grained model for Group A carbon source utilization, which includes more details to compare with experiments. (**G**) Comparison of in vivo and in vitro catalytic rates for enzymes of *E. coli* within glycolysis and the TCA cycle (see ***Appendix 1—table 1*** for details). (**H**) The proteome efficiencies for energy biogenesis in the respiration and fermentation pathways vary with growth rate as functions of the substrate quality of pyruvate (***Equations S93 and S96***)

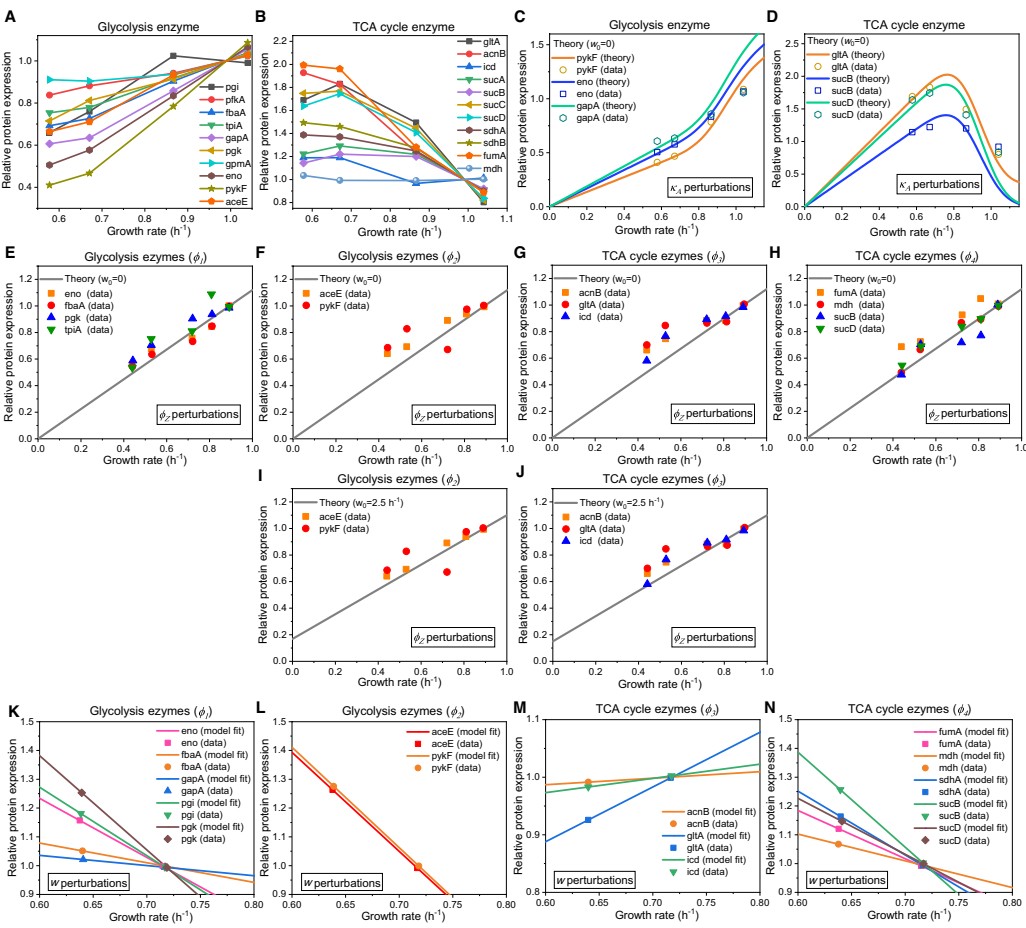

**Appendix 1—figure 3.** Relative protein expression of central metabolic enzymes in *E. coli* under various types of perturbations. (**A–D**) Relative protein expression under $\kappa_A$ perturbation. (**A**) Experimental data (*Hui et al., 2015*) for the catalytic enzymes at each step of glycolysis. (**B**) Experimental data (*Hui et al., 2015*) for the catalytic enzymes at each step of the TCA cycle. (**C**) Model predictions (*Equation S118*, with $w_0 = 0$) and experimental data (*Hui et al., 2015*) for representative glycolytic genes. (**D**) Model predictions (*Equation S118*, with $w_0 = 0$) and experimental data (*Hui et al., 2015*) for representative genes from the TCA cycle. (**E–J**) Relative protein expression under $\phi_Z$ perturbation. (**E, F, I**) Model predictions and experimental data (*Basan et al., 2015*) for representative glycolytic genes. (**G, H, J**) Model predictions and experimental data (*Basan et al., 2015*) for representative genes from the TCA cycle. (**E–H**) Results of $\phi_Z$ perturbation with $w_0 = 0$ (*Equation S120*). (**I–J**) Results of $\phi_Z$ perturbation with $w_0 = 2.5 \left( \mathrm{h}^{-1} \right)$ (*Equation S121*). (**K–N**) Relative protein expression upon energy dissipation. (**K–L**) Model fits (*Equations S127 and S123*) and experimental data (*Basan et al., 2015*) for representative glycolytic genes. (**M–N**) Model fits (*Equations S127 and S123*) and experimental data (*Basan et al., 2015*) for representative genes from the TCA cycle.

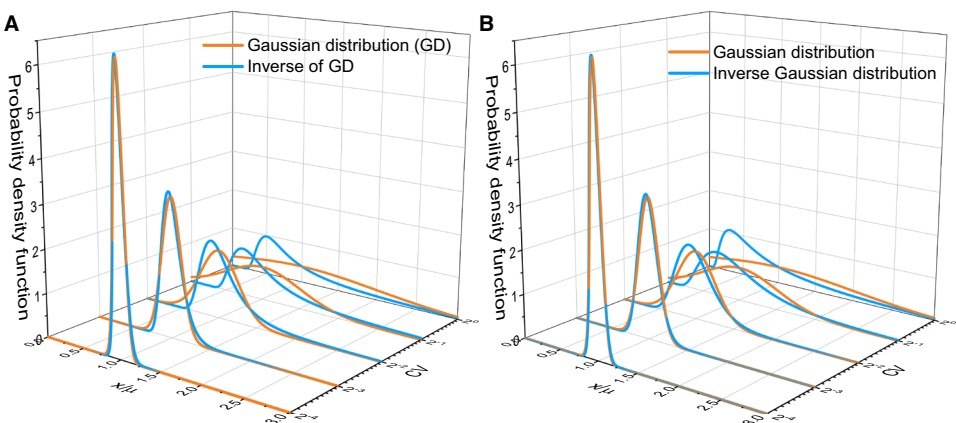

**Appendix 1—figure 4.** Asymptotic distributions of inverse Gaussian distribution and the inverse of Gaussian distribution. (**A**) Comparison between the inverse of Gaussian distribution and the corresponding Gaussian distribution for various values of the coefficient of variation (CV) (*Equations S140 and S145*). (**B**) Comparison between the inverse Gaussian distribution and the corresponding Gaussian distribution for various values of CV (*Equations S142 and S146*). Both the inverse Gaussian distribution and the inverse of Gaussian distribution converge to Gaussian distributions when CV is small.

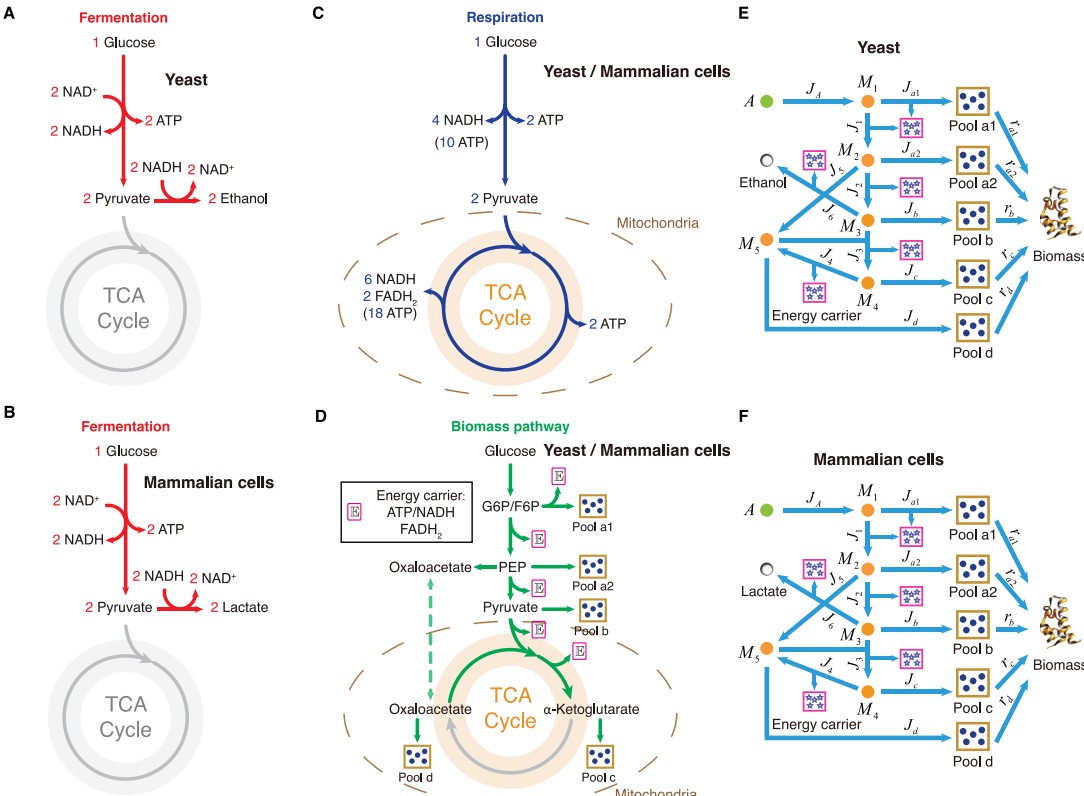

**Appendix 1—figure 5.** Carbon utilization in yeast and mammalian cells. (**A–D**) Three independent fates of glucose metabolism in yeast and mammalian cells. (**A–B**) For energy biogenesis through fermentation, one molecule of glucose generates 2 ATPs. (**C**) For energy biogenesis through respiration, one molecule of glucose generates 32 ATPs. (**D**) For biomass synthesis, glucose is converted into biomass precursors, with ATP produced as a byproduct. In yeast and mammalian cells, the energy stored in NADH and $FADH_2$ converts ADP into ATP in the mitochondria, with higher conversion factors than in *E. coli*: NADH = 2.5 ATP, $FADH_2$=1.5 ATP (*Nelson and Cox, 2008*). (**E**) Coarse-grained model for Group A carbon source utilization in yeast. (**F**) Coarse-grained model for Group A carbon source utilization in mammalian cells.

## Appendix 2

## Model framework

### 2.1 Proteome partition

Here, we adopt the proteome partition framework similar to that introduced by *Scott et al., 2010*. All proteins in a cell are classified into three classes: the fixed portion Q-class, the active ribosome-affiliated R-class, and the remaining catabolic/anabolic enzymes C-class. Each proteome class has a mass $M_i^{(P)}$ ($i = Q, R, C$) and mass fraction $\phi_i$, where $\phi_Q$ is a constant, and we define $\phi_{max} \equiv 1 - \phi_Q$. In the exponential growth phase, the ribosome allocation for protein synthesis of each class is $f_i$, with $f_Q + f_R + f_C = 1$.

To analyze cell growth optimization, we first consider the homogeneous case where all cells share identical biochemical parameters, simplifying the mass accumulation of the cell population into a 'big cell.' This simplification does not affect the value of growth rate $\lambda$. For bacteria, the protein turnover is negligible, so the mass accumulation of each class follows:

$$dM_i^{(P)}/dt = f_i \cdot k_T \cdot N_R \cdot m_{AA} \qquad (i = Q, R, C), \tag{S1}$$

where $m_{AA}$ stands for the average molecular weight of amino acids, $k_T$ is the translation rate, $N_R = M_{rp}^{(P)}/m_R$ is the number of ribosomes, $m_R$ is the protein mass of a single ribosome, and $M_{rp}^{(P)}$ is the total protein mass of ribosomes, with $M_R^{(P)}/M_{rp}^{(P)} = \varsigma \approx 1.67$ (*Neidhardt, 1996*; *Scott et al., 2010*). For a specific stable nutrient environment, $f_R$ and $k_T$ are temporal invariants. Thus,

$$M_i^{(P)}(t) = M_i^{(P)}(0) + f_i/f_R \cdot M_R^{(P)}(0) \cdot [\exp(\lambda \cdot t) - 1] \qquad (i = Q, R, C), \tag{S2}$$

where $\lambda = f_R \cdot k_T \cdot m_{AA}/(\varsigma \cdot m_R)$, and the total protein mass of the cell population $M_{protein} \equiv \sum_i^{Q,R,C} M_i^{(P)}$ follows:

$$M_{protein}(t) = M_{protein}(0) + M_R(0) \cdot [\exp(\lambda \cdot t) - 1]/f_R \tag{S3}$$

Over a long period in the exponential growth phase (i.e. $t \to +\infty$), we have $\phi_i = f_i$ ($i = Q, R, C$), and

$$\lambda = \phi_R \cdot \kappa_t, \tag{S4}$$

where $\kappa_t = k_T \cdot m_{AA}/(\varsigma \cdot m_R)$.

### 2.2 Precursor pools

Based on the entry points of the metabolic network, we classify the precursors of biomass components into five pools (*Figure 1A and B*): a1 (entry point: G6P/F6P), a2 (entry point: GA3P/3PG/PEP), b (entry point: pyruvate/acetyl-CoA), c (entry point: α-ketoglutarate), and d (entry point: oxaloacetate). These five pools draw approximately $r_{a1} = 24\%$, $r_{a2} = 24\%$, $r_b = 28\%$, $r_c = 12\%$, and $r_d = 12\%$ of the carbon flux (*Nelson and Cox, 2008*; *Wang et al., 2019*). There are overlapping components between Pools a1 and a2 due to the joint synthesis of some precursors. Therefore, we use Pool a to represent both Pools a1 and a2 in the descriptions.

### 2.3 Stoichiometric flux

We consider the following biochemical reaction between substrate $S_i$ and enzyme $E_i$:

$$E_i + S_i \underset{d_i}{\overset{a_i}{\rightleftharpoons}} E_i \cdot S_i \xrightarrow{k_i^{cat}} E_i + b_i \cdot S_{i+1} + c_i \cdot CO_2, \tag{S5}$$

where $a_i$, $d_i$ and $k_i^{cat}$ are the reaction parameters, $S_{i+1}$ is the product, $b_i$ and $c_i$ are the stoichiometric coefficients. For most of the reactions in the central metabolism, $b_i = 1$ and $c_i = 0$. The reaction rate follows Michaelis–Menten kinetics (*Nelson and Cox, 2008*):

$$v_i = k_i^{cat} \cdot [E_i] \cdot \frac{[S_i]}{[S_i] + K_i}, \tag{S6}$$

where $K_i \equiv \left(d_i + k_i^{\text{cat}}\right)/a_i$, $[E_i]$ and $[S_i]$ are the Michaelis constant, and the concentrations of enzyme $E_i$ and substrate $S_i$, respectively. For this reaction (**Equation S5**), $d\left[S_{i+1}\right]/dt = b_i \cdot v_i$ and $d\left[S_i\right]/dt = -v_i$. In the cell population (the 'big cell'), suppose that the cell volume is $V_{\text{cell}}$, then the stoichiometric flux of the reaction is:

$$J_i \equiv V_{\text{cell}} \cdot v_i. \tag{S7}$$

The copy number of enzyme $E_i$ is $N_{E_i} = V_{\text{cell}} \cdot [E_i]$ with a total weight of $M_{E_i} = N_{E_i} \cdot m_{E_i}$, where $m_{E_i}$ is the molecular weight of $E_i$. By defining the enzyme cost of an $E_i$ molecule as $n_{E_i} \equiv m_{E_i}/m_0$, where $m_0$ is a unit mass, then the cost of all $E_i$ molecules is $\Phi_i \equiv N_{E_i} \cdot n_{E_i}$ (**Wang et al., 2019**). By further defining $\xi_i \equiv \frac{k_i^{\text{cat}}}{n_{E_i}} \cdot \frac{[S_i]}{[S_i]+K_i}$, then:

$$J_i = \Phi_i \cdot \xi_i. \tag{S8}$$

The mass fraction of enzyme $E_i$ in the proteome is $\phi_i = M_{E_i}/M_{\text{protein}}$, and thus:

$$\phi_i = \Phi_i \cdot \frac{m_0}{M_{\text{protein}}}. \tag{S9}$$

## 2.4 Carbon flux and cell growth rate

To clarify the relation between the stoichiometric flux $J_i$ and growth rate $\lambda$, we consider the carbon flux in the biomass production. The carbon mass of the cell population (the 'big cell') is given by $M_{\text{carbon}} = M_{\text{protein}} \cdot r_{\text{carbon}}/r_{\text{protein}}$, where $r_{\text{carbon}}$ and $r_{\text{protein}}$ represent the mass fraction of carbon and protein within a cell. In the exponential growth phase, the carbon flux of the biomass production is given by:

$$J_{\text{BM}} = \frac{1}{m_{\text{carbon}}} \cdot \frac{dM_{\text{carbon}}}{dt} = \lambda \cdot \frac{M_{\text{carbon}}}{m_{\text{carbon}}}, \tag{S10}$$

where $m_{\text{carbon}}$ is the mass of a carbon atom. In fact, the carbon mass flux per stoichiometry varies depending on the entry point of the precursor pool. Taking Pool b as an example, there are three carbon atoms in a molecule of the entry point metabolite (i.e. pyruvate). Assuming that carbon atoms are conserved from pyruvate to Pool b, then the carbon flux of Pool b is given by $J_b^{\text{carbon}} = J_b \cdot N_{\text{py}}^{\text{carbon}}$, where $J_b$ is the stoichiometric flux from pyruvate to Pool b (**Figure 1A and B**) and $N_{\text{py}}^{\text{carbon}}$ stands for the carbon number of a pyruvate molecule. Combining with **Equation S10** and noting that $J_b^{\text{carbon}} = r_b \cdot J_{\text{BM}}$, we get $J_b \cdot N_{\text{py}}^{\text{carbon}} \cdot m_{\text{carbon}} = r_b \cdot \lambda \cdot M_{\text{carbon}}$. Similarly, for each precursor pool, we have:

$$J_i \cdot N_{\text{EP}_i}^{\text{carbon}} \cdot m_{\text{carbon}} = r_i \cdot \lambda \cdot M_{\text{carbon}} \qquad (i = a1, a2, b, c, d), \tag{S11}$$

where the subscript '$\text{EP}_i$' represents the entry point of Pool $i$, and $N_{\text{EP}_i}$ is the number of carbon atoms in a molecule of the entry-point metabolite.

For each substrate in intermediate steps of the metabolic network, we define $\kappa_i$ as the substrate quality:

$$\kappa_i \equiv \xi_i \cdot \frac{r_{\text{protein}}}{r_{\text{carbon}}} = \frac{r_{\text{protein}}}{r_{\text{carbon}}} \cdot \frac{k_i^{\text{cat}}}{n_{E_i}} \cdot \frac{[S_i]}{[S_i]+K_i}, \tag{S12}$$

and for each precursor pool, we define:

$$\eta_i \equiv r_i \cdot m_0 / (N_{\text{EP}_i}^{\text{carbon}} \cdot m_{\text{carbon}}) \quad (i = a1, a2, b, c, d). \tag{S13}$$

Combining **Equations S8, S9 and S11**, we have

$$\phi \cdot \kappa_i = \eta_i \cdot \lambda \qquad (i = a1, a2, b, c, d). \tag{S14}$$

Then, we define the normalized flux, which can be regarded as the flux per unit of biomass:

$$J_i^{(\text{N})} \equiv \phi_i \cdot \kappa_i, \tag{S15}$$

where the superscript '(N)' stands for normalized. Combined with *Equations S8, S9 and S12*, we have:

$$J_i^{(\mathrm{N})} \equiv J_i \cdot \frac{m_0}{M_{\mathrm{carbon}}}. \tag{S16}$$

Since $\sum_i^{a1,a2,b,c,d} r_i = 1$, by setting

$$m_0 = \left[ \sum_i r_i / N_{\mathrm{EP}_i}^{\mathrm{carbon}} \right]^{-1} \cdot m_{\mathrm{carbon}}, \tag{S17}$$

we then obtain:

$$\eta_i = \frac{r_i}{N_{\mathrm{EP}_i}^{\mathrm{carbon}}} \cdot \left[ \sum_j^{a1,a2,b,c,d} \frac{r_j}{N_{\mathrm{EP}_j}^{\mathrm{carbon}}} \right]^{-1} \qquad (i = a1, a2, b, c, d), \tag{S18}$$

and we have $\sum_i^{a1,a2,b,c,d} \eta_i = 1$, and

$$\sum_i^{a1,a2,b,c,d} \phi_i \cdot \kappa_i = \lambda. \tag{S19}$$

## 2.5 Intermediate nodes

In a metabolic network, the metabolites between the carbon source and precursor pools are referred to as intermediate nodes. As specified by *Wang et al., 2019*, to optimize cell growth rate, the substrate of each intermediate node is nearly saturated, and thus:

$$\kappa_i \approx \frac{r_{\mathrm{protein}}}{r_{\mathrm{carbon}}} \cdot \frac{k_i^{\mathrm{cat}}}{n_{E_i}} \tag{S20}$$

Real cases could be more complicated due to other forms of metabolic regulations. Recent quantitative studies (*Bennett et al., 2009*; *Park et al., 2016*) have shown that, at least in *E. coli*, for most of the substrate-enzyme pairs, $K_i$ is lower than the substrate concentration (i.e. $[S_i] > K_i$), which implies $\kappa_i \approx \frac{r_{\mathrm{protein}}}{r_{\mathrm{carbon}}} \cdot \frac{k_i^{\mathrm{cat}}}{n_{E_i}}$.

## Appendix 3

## Model and analysis

### 3.1 Coarse-grained model

In the coarse-grained model shown in *Figure 1B*, node $A$ represents an arbitrary carbon source of Group A (*Wang et al., 2019*), which joins at the upper part of glycolysis. Nodes M1, M2, M3, M4, and M5 stand for G6P, PEP, acetyl-CoA, α-ketoglutarate, and oxaloacetate, respectively. In the analysis of carbon supply into precursor pools, we lump sum G6P/F6P as M1, GA3P/3PG/PEP as M2, and pyruvate/acetyl-CoA as M3 for approximation. For the biochemical reactions, each follows *Equation S5* with $b_i = 1$ except for M1→2M2 and M3 +M5→M4. Basically, there are three independent possible fates for a Group A carbon source (e.g. glucose; see *Appendix 1—figure 1C-E*; *Chen and Nielsen, 2019*): energy biogenesis through fermentation; energy biogenesis via respiration (*Appendix 1—figure 1C and D*), or conversion into biomass components accompanied by energy biogenesis in the biomass pathway. Each fate involves a distinct fraction of the proteome, with no overlap between them (*Appendix 1—figure 1*).

By applying flux balance to the stoichiometric fluxes and combining with *Equation S8*, we have:

$$
\begin{cases}
\Phi_A \cdot \xi_A = \Phi_1 \cdot \xi_1 + \Phi_{a1} \cdot \xi_{a1}, \\
2\Phi_1 \cdot \xi_1 = \Phi_2 \cdot \xi_2 + \Phi_5 \cdot \xi_5 + \Phi_{a2} \cdot \xi_{a2}, \\
\Phi_2 \cdot \xi_2 = \Phi_3 \cdot \xi_3 + \Phi_6 \cdot \xi_6 + \Phi_b \cdot \xi_b, \\
\Phi_5 \cdot \xi_5 + \Phi_4 \cdot \xi_4 = \Phi_3 \cdot \xi_3 + \Phi_d \cdot \xi_d, \\
\Phi_3 \cdot \xi_3 = \Phi_4 \cdot \xi_4 + \Phi_c \cdot \xi_c.
\end{cases}
\tag{S21}
$$

Obviously, the stoichiometric fluxes of respiration $J_r$ and fermentation $J_f$ (*Appendix 1—figure 1C and D*) are:

$$
\begin{cases}
J_r \equiv J_4 = \Phi_4 \cdot \xi_4, \\
J_f \equiv J_6 = \Phi_6 \cdot \xi_6.
\end{cases}
\tag{S22}
$$

We further assume that the carbon atoms are conserved from each entry point metabolite to the precursor pool, and then,

$$
\Phi_i \cdot \xi_i \cdot N_{\text{EP}_i}^{\text{carbon}} = r_i \cdot J_{\text{BM}} \quad (i = a1, a2, b, c, d) .
\tag{S23}
$$

In terms of energy biogenesis for the relevant reactions, for convenience, we convert all the energy currencies into ATPs, namely, NADH→2ATP (*Neidhardt et al., 1990*), NADPH→2ATP (*Neidhardt et al., 1990*; *Sauer et al., 2004*), FADH$_2$→1ATP (*Neidhardt et al., 1990*). Then, we have

$$
\beta_1 \cdot \Phi_1 \cdot \xi_1 + \beta_2 \cdot \Phi_2 \cdot \xi_2 + \beta_3 \cdot \Phi_3 \cdot \xi_3 + \beta_4 \cdot \Phi_4 \cdot \xi_4 + \beta_6 \cdot \Phi_6 \cdot \xi_6 + \beta_{a1} \cdot \Phi_{a1} \cdot \xi_{a1} = J_{\text{E}},
\tag{S24}
$$

where $J_{\text{E}}$ represents the energy demand for cell proliferation, expressed as the stoichiometric energy flux in ATP. $\beta_i$ is the stoichiometric coefficient with $\beta_1 = 4$, $\beta_2 = 3$, $\beta_3 = 2$, $\beta_4 = 6$, $\beta_6 = 1$, and $\beta_{a1} = 4$ for *E. coli* (*Neidhardt et al., 1990*; *Sauer et al., 2004*). For bacteria, the energy demand is generally proportional to the carbon flux infused into biomass production, as the proportion of maintenance energy is roughly negligible (*Locasale and Cantley, 2010*). Thus,

$$
J_{\text{E}} = r_{\text{E}} \cdot J_{\text{BM}},
\tag{S25}
$$

where $r_{\text{E}}$ is the ratio and also a constant.

By applying the substitutions specified in *Equations S9, S12, S14-S18*, combined with *Equations S4, S10, S21-S25*, and the constraint of proteome resource allocation $\phi_{\text{R}} + \phi_{\text{C}} = \phi_{\text{max}}$, we have:

$$\begin{cases} \phi_A \cdot \kappa_A = \phi_1 \cdot \kappa_1 + \phi_{a1} \cdot \kappa_{a1}, \\ 2\phi_1 \cdot \kappa_1 = \phi_2 \cdot \kappa_2 + \phi_5 \cdot \kappa_5 + \phi_{a2} \cdot \kappa_{a2}, \\ \phi_2 \cdot \kappa_2 = \phi_3 \cdot \kappa_3 + \phi_6 \cdot \kappa_6 + \phi_b \cdot \kappa_b, \\ \phi_5 \cdot \kappa_5 + \phi_4 \cdot \kappa_4 = \phi_3 \cdot \kappa_3 + \phi_d \cdot \kappa_d, \\ \phi_3 \cdot \kappa_3 = \phi_4 \cdot \kappa_4 + \phi_c \cdot \kappa_c, \\ \phi_{a1} \cdot \kappa_{a1} = \eta_{a1} \cdot \lambda, \phi_{a2} \cdot \kappa_{a2} = \eta_{a2} \cdot \lambda, \phi_b \cdot \kappa_b = \eta_b \cdot \lambda, \phi_c \cdot \kappa_c = \eta_c \cdot \lambda, \phi_d \cdot \kappa_d = \eta_d \cdot \lambda, \\ \beta_1 \cdot \phi_1 \cdot \kappa_1 + \beta_2 \cdot \phi_2 \cdot \kappa_2 + \beta_3 \cdot \phi_3 \cdot \kappa_3 + \beta_4 \cdot \phi_4 \cdot \kappa_4 + \beta_6 \cdot \phi_6 \cdot \kappa_6 + \beta_{a1} \cdot \phi_{a1} \cdot \kappa_{a1} = J_E^{(N)}, \\ J_E^{(N)} = \eta_E \cdot \lambda, \lambda = \phi_R \cdot \kappa_t, J_r^{(N)} = \phi_4 \cdot \kappa_4, J_f^{(N)} = \phi_6 \cdot \kappa_6, \\ \phi_R + \phi_A + \phi_1 + \phi_2 + \phi_3 + \phi_4 + \phi_5 + \phi_6 + \phi_{a1} + \phi_{a2} + \phi_b + \phi_c + \phi_d = \phi_{max}, \end{cases} \tag{S26}$$

where $J_E^{(N)}$ and $\eta_E$ are defined as $J_E^{(N)} \equiv J_E \cdot \frac{m_0}{M_{carbon}}$ and $\eta_E \equiv r_E \cdot \left[ \sum_i r_i / N_{EP_i}^{carbon} \right]^{-1}$, respectively. Here, for each intermediate node, $\kappa_i$ follows **Equation S20**, which can be approximated as a constant. The substrate quality of the Group A carbon source $\kappa_A$ varies with the identity and concentration of the Group A carbon source:

$$\kappa_A \equiv \frac{r_{protein}}{r_{carbon}} \cdot \frac{k_A^{cat}}{m_{E_A}} \cdot \frac{[A]}{[A] + K_A} \cdot m_0, \tag{S27}$$

which is determined externally by the culture condition. From **Equation S26**, all $\phi_i$ and $\phi_R$ can be expressed in terms of $J_r^{(N)}$, $J_f^{(N)}$, and $\lambda$:

$$\begin{cases} \phi_A = \left[ J_r^{(N)} + J_f^{(N)} + (2\eta_{a1} + \eta_{a2} + \eta_b + 2\eta_c + \eta_d) \cdot \lambda \right] / (2 \cdot \kappa_A), \\ \phi_1 = \left[ J_r^{(N)} + J_f^{(N)} + (\eta_{a2} + \eta_b + 2\eta_c + \eta_d) \cdot \lambda \right] / (2 \cdot \kappa_1), \\ \phi_2 = \left[ J_r^{(N)} + J_f^{(N)} + (\eta_b + \eta_c) \cdot \lambda \right] / \kappa_2, \\ \phi_3 = \left( J_r^{(N)} + \eta_c \cdot \lambda \right) / \kappa_3, \phi_4 = J_r^{(N)} / \kappa_4, \\ \phi_5 = (\eta_c + \eta_d) \cdot \lambda / \kappa_5, \phi_6 = J_f^{(N)} / \kappa_6, \phi_R = \lambda / \kappa_t, \\ \phi_i = \eta_i \cdot \lambda / \kappa_i \quad (i = a1, a2, b, c, d). \end{cases} \tag{S28}$$

In **Equation S28**, for each $\phi_i$ or $\phi_R$, the $J_r^{(N)}$- and $J_f^{(N)}$-related proteome fraction terms belong to the fractions of the proteome dedicated to respiration (denoted as $\phi_r$) and fermentation (denoted as $\phi_f$), respectively. The $\lambda$-related proteome fraction terms belong to those involved in the biomass synthesis pathway (denoted as $\phi_{BM}$). Thus, $\phi_r = J_r^{(N)} \cdot \left[ 1 / (2 \cdot \kappa_A) + 1 / (2 \cdot \kappa_1) + 1/\kappa_2 + 1/\kappa_3 + 1/\kappa_4 \right]$,

$\phi_f = J_f^{(N)} \cdot \left[ 1 / (2 \cdot \kappa_A) + 1 / (2 \cdot \kappa_1) + 1/\kappa_2 + 1/\kappa_6 \right]$, and $\phi_{BM} = \lambda \cdot \left( \frac{1}{\kappa_t} + \frac{1 + \eta_{a1} + \eta_c}{2\kappa_A} + \frac{1 - \eta_{a1} + \eta_c}{2\kappa_1} + \frac{\eta_b + \eta_c}{\kappa_2} + \frac{\eta_c}{\kappa_3} + \frac{\eta_c + \eta_d}{\kappa_5} + \sum_i^{a1,a2,b,c,d} \frac{\eta_i}{\kappa_i} \right)$. By substituting **Equation S28** into **Equation S26**, we have:

$$\begin{cases} J_r^{(E)} + J_f^{(E)} = \varphi \cdot \lambda, \\ \dfrac{J_r^{(E)}}{\varepsilon_r} + \dfrac{J_f^{(E)}}{\varepsilon_f} = \phi_{max} - \psi \cdot \lambda. \end{cases} \tag{S29}$$

Here, $J_r^{(E)}$ and $J_f^{(E)}$ stand for the normalized energy fluxes of respiration and fermentation, with

$$\begin{cases} J_r^{(E)} = \beta_r^{(A)} \cdot J_r^{(N)} / 2, \\ J_f^{(E)} = \beta_f^{(A)} \cdot J_f^{(N)} / 2, \end{cases} \tag{S30}$$

where $\beta_r^{(A)} = \beta_1 + 2\left(\beta_2 + \beta_3 + \beta_4\right)$ and $\beta_f^{(A)} = \beta_1 + 2\left(\beta_2 + \beta_6\right)$, with $\beta_r^{(A)} = 26$ and $\beta_f^{(A)} = 12$ for *E. coli*. $\varepsilon_r$ and $\varepsilon_f$ represent the proteome efficiencies for energy biogenesis in the respiration and fermentation pathways (***Appendix 1—figure 1C-D***), defined as $\varepsilon_r \equiv J_r^{(E)}/\phi_r$ and $\varepsilon_f \equiv J_f^{(E)}/\phi_f$; that is, the normalized energy fluxes expressed in ATP generated per proteomic mass fraction dedicated to respiration and fermentation, respectively. Hence,

$$
\begin{cases}
\varepsilon_r & = \dfrac{\beta_r^{(A)}}{1/\kappa_A + 1/\kappa_1 + 2/\kappa_2 + 2/\kappa_3 + 2/\kappa_4}, \\[2mm]
\varepsilon_f & = \dfrac{\beta_f^{(A)}}{1/\kappa_A + 1/\kappa_1 + 2/\kappa_2 + 2/\kappa_6}.
\end{cases}
\tag{S31}
$$

$\psi^{-1}$ is the proteome efficiency for biomass generation in the biomass synthesis pathway (***Appendix 1—figure 1E***), defined as $\psi^{-1} \equiv \lambda/\phi_{\text{BM}} = \sum_i^{a1,a2,b,c,d} J_i^{(N)}/\phi_{\text{BM}}$ (see ***Equations S15 and S19***); that is, the normalized flux (which differs from the normalized energy flux used to define $\varepsilon_r$ and $\varepsilon_f$) generated per proteomic mass fraction dedicated to biomass synthesis. Hence

$$
\psi = \frac{1}{\kappa_t} + \frac{1 + \eta_{a1} + \eta_c}{2\kappa_A} + \frac{\eta_{a2} + \eta_b + 2\eta_c + \eta_d}{2\kappa_1} + \frac{\eta_b + \eta_c}{\kappa_2} + \frac{\eta_c}{\kappa_3} + \frac{\eta_c + \eta_d}{\kappa_5} + \sum_i^{a1,a2,b,c,d} \frac{\eta_i}{\kappa_i}.
\tag{S32}
$$

$\varphi$ is an energy demand coefficient (a constant), with

$$
\varphi \equiv \eta_E - \beta_1 \cdot \left(\eta_{a2} + \eta_b + 2\eta_c + \eta_d\right)/2 - \beta_2 \cdot \left(\eta_b + \eta_c\right) - \beta_3 \cdot \eta_c - \beta_{a1} \cdot \eta_{a1},
\tag{S33}
$$

and $\varphi \cdot \lambda$ stands for the normalized flux of energy demand other than the accompanying energy biogenesis from the biomass synthesis pathway.

## 3.2 The reason for overflow metabolism

Microbes optimize their growth rate to survive through the evolutionary process (***Vander Heiden et al., 2009***). The optimal growth principle also roughly holds for tumor cells, which proliferate while ignoring growth restriction signals and evading immune destruction by the host (***Vander Heiden et al., 2009***). First, we consider the optimal growth strategy for a single cell. The coarse-grained model for bacteria is summarized in ***Equation S26*** and further simplified in ***Equation S29***. Here, $\varepsilon_r$, $\varepsilon_f$ and $\psi$ are functions of $\kappa_A$ (see ***Equations S31, S32***), so we also denote them as $\varepsilon_r\left(\kappa_A\right)$, $\varepsilon_f\left(\kappa_A\right)$, $\psi\left(\kappa_A\right)$. Evidently, the fluxes of both respiration and fermentation take non-negative values, i.e., $J_r^{(E)}, J_f^{(E)} \geq 0$, and all the coefficients are positive: $\varepsilon_r\left(\kappa_A\right), \varepsilon_f\left(\kappa_A\right), \psi\left(\kappa_A\right), \varphi > 0$.

Thus, if $\varepsilon_r > \varepsilon_f$, then $\left(\psi + \varphi/\varepsilon_r\right) \cdot \lambda = \phi_{\max} - J_f^{(E)}\left(1/\varepsilon_f - 1/\varepsilon_r\right) \leq \phi_{\max}$. Obviously, the solution for optimal growth is:

$$
\begin{cases}
J_f^{(E)} = 0, \\
J_r^{(E)} = \varphi \cdot \lambda.
\end{cases}
\qquad \varepsilon_r > \varepsilon_f.
\tag{S34}
$$

Similarly, if $\varepsilon_f > \varepsilon_r$, then the optimal growth solution is:

$$
\begin{cases}
J_f^{(E)} = \varphi \cdot \lambda, \\
J_r^{(E)} = 0.
\end{cases}
\qquad \varepsilon_r < \varepsilon_f.
\tag{S35}
$$

In both cases, the growth rate $\lambda$ takes the maximum value for a given nutrient condition (i.e. given $\kappa_A$):

$$\lambda = \begin{cases} \lambda_r = \dfrac{\phi_{\max}}{\varphi/\varepsilon_r\left(\kappa_A\right) + \psi\left(\kappa_A\right)} & \varepsilon_r\left(\kappa_A\right) > \varepsilon_f\left(\kappa_A\right), \\[3mm] \lambda_f = \dfrac{\phi_{\max}}{\varphi/\varepsilon_f\left(\kappa_A\right) + \psi\left(\kappa_A\right)} & \varepsilon_r\left(\kappa_A\right) < \varepsilon_f\left(\kappa_A\right). \end{cases} \tag{S36}$$

So, why do microbes use the seemingly wasteful fermentation pathway when the growth rate is large under aerobic conditions? Prevalent explanations (**Basan et al., 2015**; **Chen and Nielsen, 2019**) suggest that it originates from that the proteome efficiency in fermentation is consistently higher than in respiration (i.e. $\varepsilon_f > \varepsilon_r$). If this is the case, why do microbes still use the normal respiration pathway when the growth rate is small? The answer lies in the fact that both $\varepsilon_r\left(\kappa_A\right)$ and $\varepsilon_f\left(\kappa_A\right)$ are not constants, but are dependent on nutrient conditions. In **Equation S31**, when $\kappa_A$ is small, consider the extreme case of $\kappa_A \to 0$, and then

$$\begin{cases} \varepsilon_r\left(\kappa_A \to 0\right) \approx \beta_r^{(A)} \cdot \kappa_A, \\[2mm] \varepsilon_f\left(\kappa_A \to 0\right) \approx \beta_f^{(A)} \cdot \kappa_A. \end{cases} \tag{S37}$$

Since $\beta_r^{(A)} \gg \beta_f^{(A)}$, clearly,

$$\varepsilon_r\left(\kappa_A \to 0\right) > \varepsilon_f\left(\kappa_A \to 0\right). \tag{S38}$$

Combined with **Equation S36**, thus cells would certainly use the respiration pathway when the growth rate is very small. Meanwhile, suppose that $\kappa_A^{\max}$ is the maximum value of $\kappa_A$ available across different Group A carbon sources, and if there exists a $\kappa_A$ (with $\kappa_A \leq \kappa_A^{\max}$) satisfying $\varepsilon_r\left(\kappa_A\right) < \varepsilon_f\left(\kappa_A\right)$, specifically,

$$\frac{\beta_r^{(A)} - \beta_f^{(A)}}{\kappa_A} < \beta_f^{(A)}\left(\frac{1}{\kappa_1} + \frac{2}{\kappa_2} + \frac{2}{\kappa_3} + \frac{2}{\kappa_4}\right) - \beta_r^{(A)} \cdot \left(\frac{1}{\kappa_1} + \frac{2}{\kappa_2} + \frac{2}{\kappa_6}\right), \tag{S39}$$

then $\Delta\left(\kappa_A\right) \equiv \varepsilon_f\left(\kappa_A\right)/\varepsilon_r\left(\kappa_A\right)$ is a monotonically increasing function of $\kappa_A$. Thus,

$$\varepsilon_r\left(\kappa_A^{\max}\right) < \varepsilon_f\left(\kappa_A^{\max}\right), \tag{S40}$$

and cells would use the fermentation pathway when the growth rate is large.

In practice, experimental studies using *E. coli* (**Basan et al., 2015**) have demonstrated that proteome efficiency in fermentation is higher than in respiration when the Group A carbon source is lactose at a saturated concentration, i.e., $\varepsilon_r\left(\kappa_{\text{lactose}}^{(\text{ST})}\right) < \varepsilon_f\left(\kappa_{\text{lactose}}^{(\text{ST})}\right)$. Here, $\kappa_{\text{lactose}}^{(\text{ST})}$ represents the substrate quality of lactose and the superscript '(ST)' signifies saturated concentration. In fact, *E. coli* grows much faster in G6p than lactose (**Basan et al., 2015**), thus, $\kappa_A^{\max} > \kappa_{\text{lactose}}^{(\text{ST})}$, and hence, **Equation S40** holds for *E. coli*. From a theoretical perspective, we can verify **Equation S39** and consequently **Equation S40** using **Equation S20**, combined with the in vivo/in vitro biochemical parameters obtained from experimental data (see **Appendix 1—table 1**; **Appendix 1—table 2**). For example, it is straightforward to confirm that $\varepsilon_r\left(\kappa_{\text{glucose}}^{(\text{ST})}\right) < \varepsilon_f\left(\kappa_{\text{glucose}}^{(\text{ST})}\right)$ using this method (see Appendix 9.2), further supporting the validity of **Equations S39-S40** (see also Appendix 10).

Now that **Equations S38-S40** are all valid, a critical value of $\kappa_A$, denoted as $\kappa_A^{(\text{C})}$, exists, satisfying $\Delta\left(\kappa_A^{(\text{C})}\right) = 1$. Thus,

$$\begin{cases} \varepsilon_f\left(\kappa_A\right) > \varepsilon_r\left(\kappa_A\right), & \kappa_A > \kappa_A^{(\text{C})}; \\[2mm] \varepsilon_f\left(\kappa_A\right) = \varepsilon_r\left(\kappa_A\right), & \kappa_A = \kappa_A^{(\text{C})}; \\[2mm] \varepsilon_f\left(\kappa_A\right) < \varepsilon_r\left(\kappa_A\right), & \kappa_A < \kappa_A^{(\text{C})}. \end{cases} \tag{S41}$$

Combined with **Equation S31**, we have:

$$\kappa_A^{(C)} = \frac{\beta_r^{(A)} - \beta_f^{(A)}}{\beta_f^{(A)} \left(1/\kappa_1 + 2/\kappa_2 + 2/\kappa_3 + 2/\kappa_4\right) - \beta_r^{(A)} \left(1/\kappa_1 + 2/\kappa_2 + 2/\kappa_6\right)}. \tag{S42}$$

By substituting **Equation S42** into **Equations S31, S32 and S36**, we obtain the expressions for $\varepsilon_r\left(\kappa_A^{(C)}\right)$, $\varepsilon_f\left(\kappa_A^{(C)}\right)$ and the critical growth rate at the transition point (i.e. $\lambda_C \equiv \lambda\left(\kappa_A^{(C)}\right)$):

$$\begin{cases} \varepsilon_r\left(\kappa_A^{(C)}\right) = \varepsilon_f\left(\kappa_A^{(C)}\right) = \dfrac{\beta_r^{(A)} - \beta_f^{(A)}}{2\left(1/\kappa_3 + 1/\kappa_4 - 1/\kappa_6\right)} = \dfrac{\beta_3 + \beta_4 - \beta_6}{1/\kappa_3 + 1/\kappa_4 - 1/\kappa_6}, \\ \lambda_C = \dfrac{\phi_{\max}}{\varphi/\varepsilon_{rlf}\left(\kappa_A^{(C)}\right) + \psi\left(\kappa_A^{(C)}\right)}, \end{cases} \tag{S43}$$

where $\varepsilon_{rlf}$ represents either $\varepsilon_r$ or $\varepsilon_f$. In **Figure 1E**, we show the dependencies of $\varepsilon_r\left(\kappa_A\right)$, $\varepsilon_f\left(\kappa_A\right)$, and $\lambda\left(\kappa_A\right)$ on $\kappa_A$ in a three-dimensional form, as $\kappa_A$ changes.

## 3.3 The relation between respiration/fermentation flux and growth rate

We proceed to study the relation between the respiration/fermentation flux and the cell growth rate. From **Equations S16 and S30**, we see that the stoichiometric fluxes $J_r$, $J_f$, the normalized fluxes $J_r^{(N)}$, $J_f^{(N)}$, and the normalized energy fluxes $J_r^{(E)}$, $J_f^{(E)}$ are all interconvertible. For convenience, we first analyze the relations between $J_r^{(E)}$, $J_f^{(E)}$, and $\lambda$ under growth rate optimization. In fact, all these terms are merely functions of $\kappa_A$ (see **Equations S34-S36**), which is determined by the nutrient condition (**Equation S27**).

In the homogeneous case, where all microbes share identical biochemical parameters, as $\lambda\left(\kappa_A\right)$ increases with $\kappa_A$, $J_f^{(E)}$ appear abruptly and $J_r^{(E)}$ vanish simultaneously as $\kappa_A$ exceeds $\kappa_A^{(C)}$ (**Figure 1E**; see also **Equations S34-S35, S41)**. Combining **Equations S34-S36 and S43**, we obtain:

$$\begin{cases} J_f^{(E)} = \varphi \cdot \lambda \cdot \theta\left(\lambda - \lambda_C\right), \\ J_r^{(E)} = \varphi \cdot \lambda \cdot \left[1 - \theta\left(\lambda - \lambda_C\right)\right]. \end{cases} \tag{S44}$$

where '$\theta$' stands for the Heaviside step function. Defining $\lambda_{\max} = \lambda\left(\kappa_A^{\max}\right)$, and then, $\left[0, \lambda_{\max}\right]$ is the relevant range of the x-axis. In fact, the digital responses in **Equation S44** are consistent with the numerical simulation results of **Molenaar et al., 2009**. However, these results are incompatible with the threshold-analog response in the standard picture of overflow metabolism (**Basan et al., 2015**; **Holms, 1996**).

In practice, the values of $k_i^{\mathrm{cat}}$ can be greatly influenced by the concentrations of potassium and phosphate (**García-Contreras et al., 2012**), which vary from cell to cell. Consequently, there is a distribution of values for $k_i^{\mathrm{cat}}$ among cell populations, commonly referred to as extrinsic noise (**Elowitz et al., 2002**). For convenience, we assume that each $k_i^{\mathrm{cat}}$ (and thus $\kappa_i$) follows a Gaussian distribution with a coefficient of variation (CV) of 25%. Therefore, the distributions of proteome efficiencies that determine the choice between respiration and fermentation, $\varepsilon_r$ and $\varepsilon_f$, and the critical growth rate for the transition, $\lambda_C$, can be approximated by Gaussian distributions for a cell population (see Appendix 8.1 for details). Specifically, $\lambda_C$ follows:

$$\lambda_C \sim \mathcal{N}\left(\mu_{\lambda_C}, \sigma_{\lambda_C}^2\right), \tag{S45}$$

where $\mu_{\lambda_C}$ and $\sigma_{\lambda_C}$ represent the mean and standard deviation of $\lambda_C$, with the CV $\sigma_{\lambda_C}/\mu_{\lambda_C}$ calculated to be 12% (see Appendix 9.2 for details). Note that $\lambda$ is $\kappa_A$ dependent, while $\lambda_C$ is independent of $\kappa_A$. Thus, given the growth rate of microbes in a culturing medium (e.g. in a chemostat), the normalized energy fluxes are:

$$\begin{cases} J_f^{(E)}(\lambda) = \dfrac{1}{2}\varphi \cdot \lambda \cdot \left[ \mathrm{erf}\left( \dfrac{\lambda - \mu_{\lambda_C}}{\sqrt{2}\sigma_{\lambda_C}} \right) + 1 \right], \\[4mm] J_r^{(E)}(\lambda) = \dfrac{1}{2}\varphi \cdot \lambda \cdot \left[ 1 - \mathrm{erf}\left( \dfrac{\lambda - \mu_{\lambda_C}}{\sqrt{2}\sigma_{\lambda_C}} \right) \right], \end{cases} \tag{S46}$$

where 'erf' represents the error function. In practice, given a culturing medium, there is also a probability distribution for the growth rate (*Appendix 1—figure 2B*; see also *Equation S157*). For approximation, in plotting the growth rate-respiration/fermentation flux relations, we use the deterministic (noise-free) value of the growth rate as a proxy. To compare with experiments, we essentially compare the normalized fluxes, $J_r^{(N)}$ and $J_f^{(N)}$ (see Appendix 9.1 for details). Combining *Equations S30 and S46*, we obtain:

$$\begin{cases} J_f^{(N)}(\lambda) = \dfrac{\varphi}{\beta_f^{(A)}} \cdot \lambda \cdot \left[ \mathrm{erf}\left( \dfrac{\lambda - \mu_{\lambda_C}}{\sqrt{2}\sigma_{\lambda_C}} \right) + 1 \right], \\[4mm] J_r^{(N)}(\lambda) = \dfrac{\varphi}{\beta_r^{(A)}} \cdot \lambda \cdot \left[ 1 - \mathrm{erf}\left( \dfrac{\lambda - \mu_{\lambda_C}}{\sqrt{2}\sigma_{\lambda_C}} \right) \right]. \end{cases} \tag{S47}$$

In *Figure 1C–D*, we see that *Equation S47* quantitatively illustrates the experimental data (*Basan et al., 2015*), where the model parameters were obtained using biochemical data for the catalytic enzymes (see *Appendix 1—table 1* for details).

## 3.4 Dependence of the model on optimization principles

In the derivation of the growth rate dependence of respiration/fermentation flux described above (*Equation S44* for the single-cell level and *Equation S47* for the population-averaged level), we applied the principles of optimal growth, incorporating both efficient protein allocation to enzymes and ribosomes (through ribosomal proteins). However, recent experimental studies show that the inactive portion of ribosomes (i.e. ribosomes not bound to mRNAs) may vary with culturing conditions (*Dai et al., 2017*; *Li et al., 2018*) and between individual cells within the same culture (*Pavlou et al., 2025*), despite an overall trend toward growth optimization. These findings (*Dai et al., 2017*; *Li et al., 2018*; *Pavlou et al., 2025*) suggest that ribosome allocation may be suboptimal under many culturing conditions, likely as a result of cells preparing for potential environmental changes (*Li et al., 2018*). Nevertheless, since our model's predictions regarding the binary choice between respiration and fermentation rely solely on comparing proteome efficiency between these two pathways, which involves only efficient protein allocation to enzymes, and because the active portion of ribosomes and the translation elongation rate can be approximated as constants within the growth rate range of interest for cells exhibiting overflow metabolism (*Dai et al., 2017*), our model remains applicable to suboptimal growth conditions. This can be achieved by incorporating suboptimal ribosome allocation factors, lowering the parameter $\kappa_t$ (which results in a larger $\psi$ through *Equation S32*), to account for these influences. For convenience, we present results for optimal growth below, while all model results can be extended to cases of suboptimal ribosome allocation.

Regarding the mechanism by which cells sense and choose between respiration and fermentation, although the standard picture of overflow metabolism (*Basan et al., 2015*; *Holms, 1996*) presents a growth rate dependence of fermentation flux, it is the proteome efficiency of respiration and fermentation, rather than the growth rate, that a cell should sense directly. Due to stochasticity in gene expression and metabolic reactions, the cell growth rate may fluctuate within a cell cycle (*Kiviet et al., 2014*; *Pavlou et al., 2025*), and suboptimal factors related to ribosome allocation (*Dai et al., 2017*; *Li et al., 2018*) would further complicate the scheme if cells were sensing via growth rate. Essentially, to expedite cell growth and survive under evolutionary pressure, cells should adopt the optimal strategy by directly sensing and comparing proteome efficiencies between respiration and fermentation, choosing the pathway with higher efficiency. This is analogous to how microbes choose between two types of carbon sources in a mixture for nutrient uptake (*Wang et al., 2019*). Mechanistically, the cyclic AMP (cAMP)-cAMP receptor protein (CRP) system plays an important role in sensing proteome efficiency and executing the optimal strategy between respiration and fermentation (*Basan et al., 2015*; *Towbin et al., 2017*; *Valgepea et al., 2010*; *Wehrens et al., 2023*). However, the roles of additional unidentified regulators are required to fully elucidate this mechanism (*Basan et al., 2015*; *Valgepea et al., 2010*).

## Appendix 4

## Model perturbations

### 4.1 Overexpression of useless proteins

Here, we consider the case of overexpression of the protein encoded by the *lacZ* gene (i.e. $\phi_Z$ perturbation) in *E. coli*. Effectively, this limits the proteome by altering $\phi_{\max}$:

$$\phi_{\max} \xrightarrow{\text{LacZ overexpression}} \phi_{\max} - \phi_Z, \tag{S48}$$

where $\phi_Z$ stands for the proteomic mass fraction of useless proteins, which is controllable in experiments. Then, the growth rate changes into a bivariate function of $\kappa_A$ and $\phi_Z$:

$$\lambda\left(\kappa_A, \phi_Z\right) = \begin{cases} \dfrac{\phi_{\max} - \phi_Z}{\varphi/\varepsilon_r\left(\kappa_A\right) + \psi\left(\kappa_A\right)} & \varepsilon_r\left(\kappa_A\right) > \varepsilon_f\left(\kappa_A\right), \\[4mm] \dfrac{\phi_{\max} - \phi_Z}{\varphi/\varepsilon_f\left(\kappa_A\right) + \psi\left(\kappa_A\right)} & \varepsilon_r\left(\kappa_A\right) < \varepsilon_f\left(\kappa_A\right), \end{cases} \tag{S49}$$

and thus,

$$\lambda\left(\kappa_A, \phi_Z\right) = \lambda\left(\kappa_A, 0\right)\left(1 - \phi_Z/\phi_{\max}\right). \tag{S50}$$

Obviously, $\kappa_A^{(C)}$ remains a constant (following **Equation S42**), while $\lambda_C\left(\phi_Z\right) \equiv \lambda\left(\kappa_A^{(C)}, \phi_Z\right)$ and $\lambda_{\max}\left(\phi_Z\right) \equiv \lambda\left(\kappa_A^{\max}, \phi_Z\right)$ become functions of $\phi_Z$:

$$\begin{cases} \lambda_C\left(\phi_Z\right) = \lambda_C\left(0\right)\left(1 - \phi_Z/\phi_{\max}\right), \\[2mm] \lambda_{\max}\left(\phi_Z\right) = \lambda_{\max}\left(0\right)\left(1 - \phi_Z/\phi_{\max}\right). \end{cases} \tag{S51}$$

In the homogeneous case, $J_f^{(E)}$ and $J_r^{(E)}$ follow:

$$\begin{cases} J_f^{(E)}\left(\kappa_A, \phi_Z\right) = \varphi \cdot \lambda\left(\kappa_A, \phi_Z\right) \cdot \theta\left(\lambda\left(\kappa_A, \phi_Z\right) - \lambda_C\left(\phi_Z\right)\right), \\[2mm] J_r^{(E)}\left(\kappa_A, \phi_Z\right) = \varphi \cdot \lambda\left(\kappa_A, \phi_Z\right) \cdot \left[1 - \theta\left(\lambda\left(\kappa_A, \phi_Z\right) - \lambda_C\left(\phi_Z\right)\right)\right]. \end{cases} \tag{S52}$$

Combined with **Equations S50-S51**, we have:

$$\begin{cases} J_f^{(E)}\left(\kappa_A, \phi_Z\right) = \varphi \cdot \lambda\left(\kappa_A, \phi_Z\right) \cdot \theta\left(\lambda\left(\kappa_A, 0\right) - \lambda_C\left(0\right)\right), \\[2mm] J_r^{(E)}\left(\kappa_A, \phi_Z\right) = \varphi \cdot \lambda\left(\kappa_A, \phi_Z\right) \cdot \left[1 - \theta\left(\lambda\left(\kappa_A, 0\right) - \lambda_C\left(0\right)\right)\right]. \end{cases} \tag{S53}$$

To compare with experiments, we assume that each $k_i^{\text{cat}}$ and $\kappa_i$ follow the extrinsic noise with a CV of 25% specified in Appendix 3.3, and we neglect the noise on $\phi_Z$ and $\phi_{\max}$. Combining **Equations S45 and S51**, $\lambda_C\left(\phi_Z\right)$ approximately follows a Gaussian distribution:

$$\lambda_C\left(\phi_Z\right) \sim \mathcal{N}\left(\mu_{\lambda_C}\left(\phi_Z\right), \sigma_{\lambda_C}\left(\phi_Z\right)^2\right), \tag{S54}$$

where $\mu_{\lambda_C}\left(\phi_Z\right)$ and $\sigma_{\lambda_C}\left(\phi_Z\right)$ represent the mean and standard deviation of $\lambda_C\left(\phi_Z\right)$, with

$$\begin{cases} \mu_{\lambda_C}\left(\phi_Z\right) = \mu_{\lambda_C}\left(0\right)\left(1 - \phi_Z/\phi_{\max}\right), \\[2mm] \sigma_{\lambda_C}\left(\phi_Z\right) = \sigma_{\lambda_C}\left(0\right)\left(1 - \phi_Z/\phi_{\max}\right). \end{cases} \tag{S55}$$

Here, $\mu_{\lambda_C}\left(0\right)$, $\sigma_{\lambda_C}\left(0\right)$, $\lambda_C\left(0\right)$, $\lambda_{\max}\left(0\right)$, and $\lambda\left(\kappa_A, 0\right)$ represent the parameters or variables free from $\phi_Z$ perturbation, just as those in Appendix 3.3. Since the noise on the multiplier term (i.e. $1 - \phi_Z/\phi_{\max}$) is negligible, the CV of $\lambda_C\left(\phi_Z\right)$ (i.e. $\sigma_{\lambda_C}\left(\phi_Z\right)/\mu_{\lambda_C}\left(\phi_Z\right)$) is unaffected by $\phi_Z$. By combining **Equations S46 and S48**, we obtain the relations between the normalized energy fluxes and growth rate:

$$\begin{cases} J_f^{(E)}\left(\lambda\left(\kappa_A,\phi_Z\right),\phi_Z\right) = \frac{1}{2}\varphi\cdot\lambda\left(\kappa_A,\phi_Z\right)\cdot\left[\mathrm{erf}\left(\frac{\lambda\left(\kappa_A,\phi_Z\right)-\mu_{\lambda_C}\left(\phi_Z\right)}{\sqrt{2}\sigma_{\lambda_C}\left(\phi_Z\right)}\right)+1\right], \\ J_r^{(E)}\left(\lambda\left(\kappa_A,\phi_Z\right),\phi_Z\right) = \frac{1}{2}\varphi\cdot\lambda\left(\kappa_A,\phi_Z\right)\cdot\left[1-\mathrm{erf}\left(\frac{\lambda\left(\kappa_A,\phi_Z\right)-\mu_{\lambda_C}\left(\phi_Z\right)}{\sqrt{2}\sigma_{\lambda_C}\left(\phi_Z\right)}\right)\right], \end{cases}$$

(S56)

where $\lambda\left(\kappa_A,\phi_Z\right)$, $\mu_{\lambda_C}\left(\phi_Z\right)$, and $\sigma_{\lambda_C}\left(\phi_Z\right)$ follow **Equations S50 and S55** accordingly. For a given value of $\phi_Z$, i.e., $\phi_Z$ is fixed, then, $\lambda\left(\kappa_A,\phi_Z\right)$ changes monotonically with $\kappa_A$. Combining **Equations S55-S56 and S30**, we obtain the relation between the normalized fluxes $J_r^{(N)}$, $J_f^{(N)}$, and the growth rate (where $\phi_Z$ is a parameter):

$$\begin{cases} J_f^{(N)}\left(\lambda,\phi_Z\right) = \frac{\varphi}{\beta_f^{(A)}}\cdot\lambda\cdot\left[\mathrm{erf}\left(\frac{\lambda-\mu_{\lambda_C}\left(0\right)\left(1-\phi_Z/\phi_{\max}\right)}{\sqrt{2}\sigma_{\lambda_C}\left(0\right)\left(1-\phi_Z/\phi_{\max}\right)}\right)+1\right], \\ J_r^{(N)}\left(\lambda,\phi_Z\right) = \frac{\varphi}{\beta_r^{(A)}}\cdot\lambda\cdot\left[1-\mathrm{erf}\left(\frac{\lambda-\mu_{\lambda_C}\left(0\right)\left(1-\phi_Z/\phi_{\max}\right)}{\sqrt{2}\sigma_{\lambda_C}\left(0\right)\left(1-\phi_Z/\phi_{\max}\right)}\right)\right]. \end{cases}$$

(S57)

In **Figure 2C**. we show that the model predictions (**Equation S57**) quantitatively agree with the experiments (**Basan et al., 2015**).

Meanwhile, we can also perturb the growth rate by tuning $\phi_Z$ in a stable culturing environment with fixed concentration of a Group A carbon source (i.e. given $[A]$). In fact, for this case there is a distribution of $\kappa_A$ values due to the extrinsic noise in $k_A^{\mathrm{cat}}$, yet this distribution is fixed. For convenience of description, we still referred to it as fixed $\kappa_A$. Then, combining **Equations S30, S50, S55 and S56**, we get:

$$\begin{cases} J_f^{(N)}\left(\lambda,\phi_Z\right) = \frac{\varphi}{\beta_f^{(A)}}\cdot\left[\mathrm{erf}\left(\frac{\lambda\left(\kappa_A,0\right)-\mu_{\lambda_C}\left(0\right)}{\sqrt{2}\sigma_{\lambda_C}\left(0\right)}\right)+1\right]\cdot\lambda, \\ J_r^{(N)}\left(\lambda,\phi_Z\right) = \frac{\varphi}{\beta_r^{(A)}}\cdot\left[1-\mathrm{erf}\left(\frac{\lambda\left(\kappa_A,0\right)-\mu_{\lambda_C}\left(0\right)}{\sqrt{2}\sigma_{\lambda_C}\left(0\right)}\right)\right]\cdot\lambda. \end{cases}$$

(S58)

Here, $\lambda\left(\kappa_A,0\right)$ remains unaltered as $\kappa_A$ is fixed. Therefore, in this case, $J_f^{(N)}$ and $J_r^{(N)}$ are proportional to $\lambda$, where the slopes are both functions of $\kappa_A$. More specifically, the slope of $J_f^{(N)}$ is a monotonically increasing function of $\kappa_A$, while that of $J_r^{(N)}$ is a monotonically decreasing function of $\kappa_A$. In **Figure 2B**, we see that the model predictions (**Equation S58**) agree quantitatively with the experiments (**Basan et al., 2015**).

In fact, the growth rate can be altered by tuning $\phi_Z$ and $\kappa_A$ simultaneously. Then, the relations among the energy fluxes, growth rate, and $\phi_Z$ still follow **Equation S57** (where $\phi_Z$ is a variable). In a 3-D representation, these relations correspond to a surface. In **Figure 2A**, we show that the model predictions (**Equation S57**) match well with the experimental data (**Basan et al., 2015**).

## 4.2 Energy dissipation

In practice, energy dissipation disrupts the proportional relationship between energy demand and biomass production. Thus, **Equation S25** becomes:

$$J_E = r_E\cdot J_{BM} + w\cdot\frac{M_{\mathrm{carbon}}}{m_0},$$

(S59)

where $w$ represents the dissipation coefficient. In fact, maintenance energy contributes to energy dissipation, and we define the maintenance energy coefficient as $w_0$. In bacteria, the impact of maintenance energy is roughly negligible, yet in tumor cells, it plays a much more significant role (**Locasale and Cantley, 2010**).

The introduction of energy dissipation leads to a modification of **Equation S26**: combining **Equation S59** and **Equation S16**, we have:

$$J_E^{(N)} = \eta_E\cdot\lambda + w.$$

(S60)

Then, *Equation S29* changes to:

$$\begin{cases} J_r^{(E)} + J_f^{(E)} = \varphi \cdot \lambda + w, \\ \dfrac{J_r^{(E)}}{\varepsilon_r} + \dfrac{J_f^{(E)}}{\varepsilon_f} = \phi_{\max} - \psi \cdot \lambda. \end{cases} \tag{S61}$$

Consequently, if $\varepsilon_r > \varepsilon_f$, the optimal growth strategy for the cell is:

$$\begin{cases} J_f^{(E)} = 0, \\ J_r^{(E)} = \varphi \cdot \lambda + w, \end{cases} \quad \varepsilon_r > \varepsilon_f, \tag{S62}$$

and if $\varepsilon_f > \varepsilon_r$, the optimal growth strategy is:

$$\begin{cases} J_f^{(E)} = \varphi \cdot \lambda + w, \\ J_r^{(E)} = 0. \end{cases} \quad \varepsilon_r < \varepsilon_f. \tag{S63}$$

Then, the growth rate becomes a bivariate function of both $\kappa_A$ and $w$:

$$\lambda\left(\kappa_A, w\right) = \begin{cases} \dfrac{\phi_{\max} - w/\varepsilon_r\left(\kappa_A\right)}{\varphi/\varepsilon_r\left(\kappa_A\right) + \psi\left(\kappa_A\right)} & \varepsilon_r\left(\kappa_A\right) > \varepsilon_f\left(\kappa_A\right), \\ \dfrac{\phi_{\max} - w/\varepsilon_f\left(\kappa_A\right)}{\varphi/\varepsilon_f\left(\kappa_A\right) + \psi\left(\kappa_A\right)} & \varepsilon_r\left(\kappa_A\right) < \varepsilon_f\left(\kappa_A\right). \end{cases} \tag{S64}$$

Clearly, $\kappa_A^{(C)}$ is still a constant, while $\lambda_C\left(w\right) \equiv \lambda\left(\kappa_A^{(C)}, w\right)$ and $\lambda_{\max}\left(w\right) \equiv \lambda\left(\kappa_A^{\max}, w\right)$ become functions of $w$:

$$\begin{cases} \lambda_C\left(w\right) = \lambda_C\left(0\right)\left\{1 - w/\left[\varepsilon_{r/f}\left(\kappa_A^{(C)}\right)\phi_{\max}\right]\right\}, \\ \lambda_{\max}\left(w\right) = \lambda_{\max}\left(0\right)\left\{1 - w/\left[\varepsilon_f\left(\kappa_A^{\max}\right)\phi_{\max}\right]\right\}. \end{cases} \tag{S65}$$

For a cell population, in the homogeneous case, $J_f^{(E)}$ and $J_r^{(E)}$ follow:

$$\begin{cases} J_f^{(E)}\left(\kappa_A, w\right) = \left[\varphi \cdot \lambda\left(\kappa_A, w\right) + w\right] \cdot \theta\left(\lambda\left(\kappa_A, w\right) - \lambda_C\left(w\right)\right), \\ J_r^{(E)}\left(\kappa_A, w\right) = \left[\varphi \cdot \lambda\left(\kappa_A, w\right) + w\right] \cdot \left[1 - \theta\left(\lambda\left(\kappa_A, w\right) - \lambda_C\left(w\right)\right)\right]. \end{cases} \tag{S66}$$

To compare with experiments, we assume the same extent of extrinsic noise in $k_i^{\text{cat}}$ (and thus $\kappa_i$) as that specified in Appendix 3.3. Combining *Equations S45 and S65*, $\lambda_C\left(w\right)$ approximately follows a Gaussian distribution:

$$\lambda_C\left(w\right) \sim \mathcal{N}\left(\mu_{\lambda_C}\left(w\right), \sigma_{\lambda_C}\left(w\right)^2\right), \tag{S67}$$

where $\mu_{\lambda_C}\left(w\right)$ and $\sigma_{\lambda_C}\left(w\right)$ represent the mean and standard deviation of $\lambda_C\left(w\right)$, and

$$\begin{cases} \mu_{\lambda_C}\left(w\right) = \mu_{\lambda_C}\left(0\right)\left\{1 - w/\left[\varepsilon_{r/f}\left(\kappa_A^{(C)}\right)\phi_{\max}\right]\right\}, \\ \sigma_{\lambda_C}\left(w\right) \approx \sigma_{\lambda_C}\left(0\right)\left\{1 - w/\left[\varepsilon_{r/f}\left(\kappa_A^{(C)}\right)\phi_{\max}\right]\right\}. \end{cases} \tag{S68}$$

Here, $\mu_{\lambda_C}\left(0\right)$, $\sigma_{\lambda_C}\left(0\right)$, $\lambda_C\left(0\right)$, $\lambda_{\max}\left(0\right)$, and $\lambda\left(\kappa_A, 0\right)$ represent parameters or variables unaffected by energy dissipation. In fact, there is a distribution of values for $\varepsilon_{r/f}\left(\kappa_A^{(C)}\right)$. For approximation, we use the deterministic value of $\varepsilon_{r/f}\left(\kappa_A^{(C)}\right)$ in *Equation S68*, and then the CV of $\lambda_C\left(w\right)$ remains largely unperturbed by $w$. Combining *Equations S46, S66 and S67*, we have:

$$
\begin{cases}
J_f^{(E)}\left(\lambda\left(\kappa_A, w\right), w\right) = \dfrac{1}{2}\left(\varphi \cdot \lambda\left(\kappa_A, w\right) + w\right) \cdot \left[\mathrm{erf}\left(\dfrac{\lambda\left(\kappa_A, w\right) - \mu_{\lambda_C}(w)}{\sqrt{2}\sigma_{\lambda_C}(w)}\right) + 1\right], \\[3mm]
J_r^{(E)}\left(\lambda\left(\kappa_A, w\right), w\right) = \dfrac{1}{2}\left(\varphi \cdot \lambda\left(\kappa_A, w\right) + w\right) \cdot \left[1 - \mathrm{erf}\left(\dfrac{\lambda\left(\kappa_A, w\right) - \mu_{\lambda_C}(w)}{\sqrt{2}\sigma_{\lambda_C}(w)}\right)\right].
\end{cases}
\tag{S69}
$$

Since the dissipation coefficient $w$ is tunable in experiments, for a given value of $w$, $\lambda\left(\kappa_A, w\right)$ changes monotonically with $\kappa_A$. Combining **Equations S68-S69 and S30**, we have (here $w$ is a parameter):

$$
\begin{cases}
J_f^{(N)}(\lambda, w) = \dfrac{\varphi \cdot \lambda + w}{\beta_f^{(A)}} \cdot \left[\mathrm{erf}\left(\dfrac{\lambda - \mu_{\lambda_C}(0)\left\{1 - w/\left[\varepsilon_{rlf}\left(\kappa_A^{(C)}\right)\phi_{\max}\right]\right\}}{\sqrt{2}\sigma_{\lambda_C}(0)\left\{1 - w/\left[\varepsilon_{rlf}\left(\kappa_A^{(C)}\right)\phi_{\max}\right]\right\}}\right) + 1\right], \\[5mm]
J_r^{(N)}(\lambda, w) = \dfrac{\varphi \cdot \lambda + w}{\beta_r^{(A)}} \cdot \left[1 - \mathrm{erf}\left(\dfrac{\lambda - \mu_{\lambda_C}(0)\left\{1 - w/\left[\varepsilon_{rlf}\left(\kappa_A^{(C)}\right)\phi_{\max}\right]\right\}}{\sqrt{2}\sigma_{\lambda_C}(0)\left\{1 - w/\left[\varepsilon_{rlf}\left(\kappa_A^{(C)}\right)\phi_{\max}\right]\right\}}\right)\right].
\end{cases}
\tag{S70}
$$

The comparison between model predictions (**Equation S70**) and experimental results (**Basan et al., 2015**) is shown in **Figure 3B**, which shows quantitative agreement. Meanwhile, the growth rate can also be perturbed by changing $\kappa_A$ and $w$ simultaneously. The relations among the energy fluxes, growth rate and $w$ follow **Equation S70** (here $w$ is a variable). In a 3D representation, these relations form a surface. As shown in **Figure 3A**, the model predictions (**Equation S70**) agree quantitatively with the experimental results (**Basan et al., 2015**).

## 4.3 Translation inhibition

In *E. coli*, the translation rate can be modified by adding different concentrations of translation inhibitors, e.g., chloramphenicol (Cm). The net effect of this perturbation is represented as:

$$
\kappa_t \xrightarrow{\text{Translation inhibition}} \kappa_t/\left(\iota + 1\right),
\tag{S71}
$$

where $\iota$ stands for the inhibition coefficient with $\iota > 0$, and $\left(1 + \iota\right)^{-1}$ represents the translation efficiency. Thus, **Equation S32** changes to:

$$
\psi\left(\kappa_A, \iota\right) = \frac{\iota + 1}{\kappa_t} + \frac{1 + \eta_{a1} + \eta_c}{2\kappa_A} + \frac{\eta_{a2} + \eta_b + 2\eta_c + \eta_d}{2\kappa_1} + \frac{\eta_b + \eta_c}{\kappa_2} + \frac{\eta_c}{\kappa_3} + \frac{\eta_c + \eta_d}{\kappa_5} + \sum_i^{a1,a2,b,c,d} \frac{\eta_i}{\kappa_i}
\tag{S72}
$$

First, we consider the case where maintenance energy is neglected, i.e., $w_0 = 0$. In this case, the growth rate takes the following form:

$$
\lambda\left(\kappa_A, \iota\right) = \begin{cases}
\dfrac{\phi_{\max}}{\varphi/\varepsilon_r\left(\kappa_A\right) + \psi\left(\kappa_A, \iota\right)} & \varepsilon_r\left(\kappa_A\right) > \varepsilon_f\left(\kappa_A\right), \\[4mm]
\dfrac{\phi_{\max}}{\varphi/\varepsilon_f\left(\kappa_A\right) + \psi\left(\kappa_A, \iota\right)} & \varepsilon_r\left(\kappa_A\right) < \varepsilon_f\left(\kappa_A\right),
\end{cases}
\tag{S73}
$$

where $\lambda\left(\kappa_A, 0\right)$ and $\psi\left(\kappa_A, 0\right)$ represent the terms unaffected by translation inhibition. Thus, $\lambda_C\left(\iota\right) \equiv \lambda\left(\kappa_A^{(C)}, \iota\right)$ and $\lambda_{\max}\left(\iota\right) \equiv \lambda\left(\kappa_A^{\max}, \iota\right)$ become functions of $\iota$:

$$
\begin{cases}
\lambda_C\left(\iota\right) = \lambda_C(0)\,\dfrac{\varphi/\varepsilon_{rlf}\left(\kappa_A^{(C)}\right) + \psi\left(\kappa_A^{(C)}, 0\right)}{\varphi/\varepsilon_{rlf}\left(\kappa_A^{(C)}\right) + \psi\left(\kappa_A^{(C)}, \iota\right)}, \\[5mm]
\lambda_{\max}\left(\iota\right) = \lambda_{\max}(0)\,\dfrac{\varphi/\varepsilon_f\left(\kappa_A^{\max}\right) + \psi\left(\kappa_A^{\max}, 0\right)}{\varphi/\varepsilon_f\left(\kappa_A^{\max}\right) + \psi\left(\kappa_A^{\max}, \iota\right)}.
\end{cases}
\tag{S74}
$$

In the homogeneous case, $J_f^{(E)}$ and $J_r^{(E)}$ follow:

$$\begin{cases} J_f^{(\mathrm{E})}\left(\kappa_A, \iota\right) = \varphi \cdot \lambda\left(\kappa_A, \iota\right) \cdot \theta\left(\lambda\left(\kappa_A, \iota\right) - \lambda_{\mathrm{C}}\left(\iota\right)\right), \\ J_r^{(\mathrm{E})}\left(\kappa_A, \iota\right) = \varphi \cdot \lambda\left(\kappa_A, \iota\right) \cdot \left[1 - \theta\left(\lambda\left(\kappa_A, \iota\right) - \lambda_{\mathrm{C}}\left(\iota\right)\right)\right]. \end{cases} \tag{S75}$$

To compare with experiments, we assume that extrinsic noise exists in $k_i^{\mathrm{cat}}$ and $\kappa_i$ as specified in Appendix 3.3. Combining **Equations S45 and S74**, $\lambda_{\mathrm{C}}\left(\iota\right)$ can be approximated by a Gaussian distribution:

$$\lambda_{\mathrm{C}}\left(\iota\right) \sim \mathcal{N}\left(\mu_{\lambda_{\mathrm{C}}}\left(\iota\right), \sigma_{\lambda_{\mathrm{C}}}\left(\iota\right)^2\right), \tag{S76}$$

where $\mu_{\lambda_{\mathrm{C}}}\left(\iota\right)$ and $\sigma_{\lambda_{\mathrm{C}}}\left(\iota\right)$ represent the mean and standard deviation of $\lambda_{\mathrm{C}}\left(\iota\right)$, with

$$\begin{cases} \mu_{\lambda_{\mathrm{C}}}\left(\iota\right) = \mu_{\lambda_{\mathrm{C}}}\left(0\right) \dfrac{\varphi/\varepsilon_{rlf}\left(\kappa_A^{(\mathrm{C})}\right) + \psi\left(\kappa_A^{(\mathrm{C})}, 0\right)}{\varphi/\varepsilon_{rlf}\left(\kappa_A^{(\mathrm{C})}\right) + \psi\left(\kappa_A^{(\mathrm{C})}, \iota\right)}, \\[3ex] \sigma_{\lambda_{\mathrm{C}}}\left(\iota\right) \approx \sigma_{\lambda_{\mathrm{C}}}\left(0\right) \dfrac{\varphi/\varepsilon_{rlf}\left(\kappa_A^{(\mathrm{C})}\right) + \psi\left(\kappa_A^{(\mathrm{C})}, 0\right)}{\varphi/\varepsilon_{rlf}\left(\kappa_A^{(\mathrm{C})}\right) + \psi\left(\kappa_A^{(\mathrm{C})}, \iota\right)}. \end{cases} \tag{S77}$$

Here, $\mu_{\lambda_{\mathrm{C}}}\left(0\right)$, $\sigma_{\lambda_{\mathrm{C}}}\left(0\right)$, $\psi\left(\kappa_A^{(\mathrm{C})}, 0\right)$, $\lambda_{\mathrm{C}}\left(0\right)$ and $\lambda_{\max}\left(0\right)$ stand for the terms unaffected by translation inhibition. Essentially, there are distributions of values for $\varepsilon_{rlf}\left(\kappa_A^{(\mathrm{C})}\right)$, $\psi\left(\kappa_A^{(\mathrm{C})}, 0\right)$ and $\psi\left(\kappa_A^{(\mathrm{C})}, \iota\right)$. For approximation, we use the deterministic values of these terms in **Equation S77**, and then the CV of $\lambda_{\mathrm{C}}\left(\iota\right)$ can be approximated by $\lambda_{\mathrm{C}}\left(0\right)$. Combining **Equations S46, S75 and S76**, we have:

$$\begin{cases} J_f^{(\mathrm{E})}\left(\lambda\left(\kappa_A, \iota\right), \iota\right) = \dfrac{1}{2}\varphi \cdot \lambda\left(\kappa_A, \iota\right) \cdot \left[\mathrm{erf}\left(\dfrac{\lambda\left(\kappa_A, \iota\right) - \mu_{\lambda_{\mathrm{C}}}\left(\iota\right)}{\sqrt{2}\sigma_{\lambda_{\mathrm{C}}}\left(\iota\right)}\right) + 1\right], \\[3ex] J_r^{(\mathrm{E})}\left(\lambda\left(\kappa_A, \iota\right), \iota\right) = \dfrac{1}{2}\varphi \cdot \lambda\left(\kappa_A, \iota\right) \cdot \left[1 - \mathrm{erf}\left(\dfrac{\lambda\left(\kappa_A, \iota\right) - \mu_{\lambda_{\mathrm{C}}}\left(\iota\right)}{\sqrt{2}\sigma_{\lambda_{\mathrm{C}}}\left(\iota\right)}\right)\right]. \end{cases} \tag{S78}$$

In the experiments, the inhibition coefficient $\iota$ is controllable by adjusting the concentration of the translation inhibitor. For a given value of $\iota$, $\lambda\left(\kappa_A, \iota\right)$ changes monotonically with $\kappa_A$. Combining **Equations S30 and S78**, we have (here $\iota$ is a parameter):

$$\begin{cases} J_f^{(\mathrm{N})}\left(\lambda, \iota\right) = \dfrac{\varphi \cdot \lambda}{\beta_f^{(A)}} \cdot \left[\mathrm{erf}\left(\dfrac{\lambda - \mu_{\lambda_{\mathrm{C}}}\left(\iota\right)}{\sqrt{2}\sigma_{\lambda_{\mathrm{C}}}\left(\iota\right)}\right) + 1\right], \\[3ex] J_r^{(\mathrm{N})}\left(\lambda, \iota\right) = \dfrac{\varphi \cdot \lambda}{\beta_r^{(A)}} \cdot \left[1 - \mathrm{erf}\left(\dfrac{\lambda - \mu_{\lambda_{\mathrm{C}}}\left(\iota\right)}{\sqrt{2}\sigma_{\lambda_{\mathrm{C}}}\left(\iota\right)}\right)\right], \end{cases} \tag{S79}$$

where $\mu_{\lambda_{\mathrm{C}}}\left(\iota\right)$ and $\sigma_{\lambda_{\mathrm{C}}}\left(\iota\right)$ follow **Equation S77**. The growth rate can also be perturbed by altering both $\kappa_A$ and $\iota$ simultaneously. In this case, the relations among the energy fluxes, growth rate and $\iota$ still follow **Equation S79** (here $\iota$ is a variable). The comparison between **Equation S79** and experimental data (**Basan et al., 2015**) is shown in **Appendix 1—figure 2D (3-D) and E(2-D)**. Overall, there is good consistency; however, there remains a noticeable discrepancy when $\iota$ is large (i.e. at high concentration of the translation inhibitor). This led us to consider the maintenance energy through the coefficient $w_0$, which is small but may account for this discrepancy. Then, $\lambda\left(\kappa_A, \iota\right)$ changes into:

$$\lambda\left(\kappa_A, \iota\right) = \begin{cases} \dfrac{\phi_{\max} - w_0/\varepsilon_r\left(\kappa_A\right)}{\varphi/\varepsilon_r\left(\kappa_A\right) + \psi\left(\kappa_A, \iota\right)} & \varepsilon_r\left(\kappa_A\right) > \varepsilon_f\left(\kappa_A\right), \\[3ex] \dfrac{\phi_{\max} - w_0/\varepsilon_f\left(\kappa_A\right)}{\varphi/\varepsilon_f\left(\kappa_A\right) + \psi\left(\kappa_A, \iota\right)} & \varepsilon_r\left(\kappa_A\right) < \varepsilon_f\left(\kappa_A\right), \end{cases} \tag{S80}$$

while $\lambda_C(\iota) \equiv \lambda\left(\kappa_A^{(C)}, \iota\right)$ and $\lambda_{\max}(\iota) \equiv \lambda\left(\kappa_A^{\max}, \iota\right)$ still follow **Equation S74**, though the forms of $\lambda_C(0)$ and $\lambda_{\max}(0)$ change to:

$$\begin{cases} \lambda_C(0) = \dfrac{\phi_{\max} - w_0/\varepsilon_{rlf}\left(\kappa_A^{(C)}\right)}{\varphi/\varepsilon_{rlf}\left(\kappa_A^{(C)}\right) + \psi\left(\kappa_A^{(C)}, 0\right)}, \\[3mm] \lambda_{\max}(0) = \dfrac{\phi_{\max} - w_0/\varepsilon_f\left(\kappa_A^{\max}\right)}{\varphi/\varepsilon_f\left(\kappa_A^{\max}\right) + \psi\left(\kappa_A^{\max}, 0\right)}. \end{cases} \tag{S81}$$

In the homogeneous case, $J_f^{(E)}$ and $J_r^{(E)}$ follow:

$$\begin{cases} J_f^{(E)}(\kappa_A, \iota) = [\varphi \cdot \lambda(\kappa_A, \iota) + w_0] \cdot \theta\left(\lambda(\kappa_A, \iota) - \lambda_C(\iota)\right), \\[2mm] J_r^{(E)}(\kappa_A, \iota) = [\varphi \cdot \lambda(\kappa_A, \iota) + w_0] \cdot \left[1 - \theta\left(\lambda(\kappa_A, \iota) - \lambda_C(\iota)\right)\right]. \end{cases} \tag{S82}$$

To compare with experiments, we assume that the extrinsic noise follows the specification in Appendix 3.3. Combining **Equations S45, S74 and S81**, $\lambda_C(\iota)$ approximately follows a Gaussian distribution:

$$\lambda_C(\iota) \sim \mathcal{N}\left(\mu_{\lambda_C}(\iota), \sigma_{\lambda_C}(\iota)^2\right) \tag{S83}$$

Here $\mu_{\lambda_C}(\iota)$ and $\sigma_{\lambda_C}(\iota)$ still follow **Equation S77**, while $\mu_{\lambda_C}(0)$ and $\sigma_{\lambda_C}(0)$ change accordingly with $\lambda_C(0)$ (see **Equation S81**). For approximation, we use the deterministic values of the relevant terms in **Equation S77**, and then the CV of $\lambda_C(\iota)$ is roughly the same as $\lambda_C(0)$. Combining **Equations S46, S82 and S83**, we have:

$$\begin{cases} J_f^{(E)}(\lambda(\kappa_A, \iota), \iota) = \dfrac{1}{2}(\varphi \cdot \lambda(\kappa_A, \iota) + w_0) \cdot \left[\mathrm{erf}\left(\dfrac{\lambda(\kappa_A, \iota) - \mu_{\lambda_C}(\iota)}{\sqrt{2}\sigma_{\lambda_C}(\iota)}\right) + 1\right], \\[4mm] J_r^{(E)}(\lambda(\kappa_A, \iota), \iota) = \dfrac{1}{2}(\varphi \cdot \lambda(\kappa_A, \iota) + w_0) \cdot \left[1 - \mathrm{erf}\left(\dfrac{\lambda(\kappa_A, \iota) - \mu_{\lambda_C}(\iota)}{\sqrt{2}\sigma_{\lambda_C}(\iota)}\right)\right]. \end{cases} \tag{S84}$$

Thus, for a given $\iota$, $\lambda(\kappa_A, \iota)$ changes monotonically with $\kappa_A$. Combining **Equations S30 and S84**, we have (here $\iota$ is a parameter):

$$\begin{cases} J_f^{(N)}(\lambda, \iota) = \dfrac{\varphi \cdot \lambda + w_0}{\beta_f^{(A)}} \cdot \left[\mathrm{erf}\left(\dfrac{\lambda - \mu_{\lambda_C}(\iota)}{\sqrt{2}\sigma_{\lambda_C}(\iota)}\right) + 1\right]. \\[4mm] J_r^{(N)}(\lambda, \iota) = \dfrac{\varphi \cdot \lambda + w_0}{\beta_r^{(A)}} \cdot \left[1 - \mathrm{erf}\left(\dfrac{\lambda - \mu_{\lambda_C}(\iota)}{\sqrt{2}\sigma_{\lambda_C}(\iota)}\right)\right]. \end{cases} \tag{S85}$$

The growth rate and fluxes can also be perturbed by altering both $\kappa_A$ and $\iota$ simultaneously. The relations among the energy fluxes, growth rate, and $\iota$ would still follow **Equation S85**, except that $\iota$ is now regarded as a variable. Assuming a small amount of maintenance energy by assigning $w_0 = 2.5\ (h^{-1})$, we find that the experimental results (**Basan et al., 2015**) agree quantitatively well with the model predictions (**Figure 3C and D**).

# Appendix 5

## Overflow metabolism in substrates other than Group A carbon sources

Due to the topology of the metabolic network, for cells using Group A carbon sources, the behavior of overflow metabolism follows *Equation 5* (or *Equation S47*) upon $\kappa_A$ perturbation (i.e. varying the type or concentration of a Group A carbon source). This has been demonstrated clearly in the above analysis and agrees quantitatively with experiments. However, further analysis is required for cells using substrates other than Group A sources due to the topological differences in carbon utilization (*Wang et al., 2019*). In principle, substrates entering from glycolysis or the points before acetyl-CoA are potentially involved in overflow metabolism, while those joining from the TCA cycle are not relevant to this behavior. Still, mixed carbon sources are likely to induce a different profile of overflow metabolism, as long as there is a carbon source derived from glycolysis.

### 5.1 Pyruvate

The coarse-grained model for pyruvate utilization is shown in *Figure 3E*. Here, nodes $M_1$, $M_2$, $M_3$, $M_4$, $M_5$ follow the descriptions in Appendix 3.1. Each biochemical reaction follows *Equation S5* with $b_i = 1$ except that $2M_2 \rightarrow M_1$ and $M_3 + M_5 \rightarrow M_4$. By applying flux balance to the stoichiometric fluxes, combining with *Equation S8*, we have:

$$
\begin{cases}
\Phi_{py} \cdot \xi_{py} = \Phi_7 \cdot \xi_7 + \Phi_8 \cdot \xi_8, \\
\Phi_7 \cdot \xi_7 = 2\Phi_9 \cdot \xi_9 + \Phi_5 \cdot \xi_5 + \Phi_{a2} \cdot \xi_{a2}, \\
\Phi_9 \cdot \xi_9 = \Phi_{a1} \cdot \xi_{a1}, \\
\Phi_8 \cdot \xi_8 = \Phi_3 \cdot \xi_3 + \Phi_6 \cdot \xi_6 + \Phi_b \cdot \xi_b, \\
\Phi_5 \cdot \xi_5 + \Phi_4 \cdot \xi_4 = \Phi_3 \cdot \xi_3 + \Phi_d \cdot \xi_d, \\
\Phi_3 \cdot \xi_3 = \Phi_4 \cdot \xi_4 + \Phi_c \cdot \xi_c.
\end{cases}
\tag{S86}
$$

For energy biogenesis, we convert all the energy currencies into ATPs, and then,

$$
\beta_8 \cdot \Phi_8 \cdot \xi_8 + \beta_3 \cdot \Phi_3 \cdot \xi_3 + \beta_4 \cdot \Phi_4 \cdot \xi_4 + \beta_6 \cdot \Phi_6 \cdot \xi_6 + \beta_{a1} \cdot \Phi_{a1} \cdot \xi_{a1} - \beta_7 \cdot \Phi_7 \cdot \xi_7 - \beta_9 \cdot \Phi_9 \cdot \xi_9 = J_E
\tag{S87}
$$

where $\beta_7 = 1$, $\beta_8 = 2$, $\beta_3 = 2$, $\beta_4 = 6$, $\beta_6 = 1$, $\beta_9 = 6$, $\beta_{a1} = 4$ for *E. coli* (*Neidhardt et al., 1990*; *Sauer et al., 2004*), and $J_E$ follows *Equation S25*. By applying the substitutions specified in *Equations S9, S12, S14-S18*, combined with *Equations S4, S10, S22, S23, S25, S86-S87*, and the constraint of proteome resource allocation, we have:

$$
\begin{cases}
\phi_{py} \cdot \kappa_{py} = \phi_7 \cdot \kappa_7 + \phi_8 \cdot \kappa_8, \\
\phi_7 \cdot \kappa_7 = 2\phi_9 \cdot \kappa_9 + \phi_5 \cdot \kappa_5 + \phi_{a2} \cdot \kappa_{a2}, \\
\phi_9 \cdot \kappa_9 = \phi_{a1} \cdot \kappa_{a1} \\
\phi_8 \cdot \kappa_8 = \phi_3 \cdot \kappa_3 + \phi_6 \cdot \kappa_6 + \phi_b \cdot \kappa_b \\
\phi_3 \cdot \kappa_3 = \phi_4 \cdot \kappa_4 + \phi_c \cdot \kappa_c \\
\phi_5 \cdot \kappa_5 + \phi_4 \cdot \kappa_4 = \phi_3 \cdot \kappa_3 + \phi_d \cdot \kappa_d \\
\phi_{a1} \cdot \kappa_{a1} = \eta_{a1} \cdot \lambda, \phi_{a2} \cdot \kappa_{a2} = \eta_{a2} \cdot \lambda, \phi_b \cdot \kappa_b = \eta_b \cdot \lambda, \phi_c \cdot \kappa_c = \eta_c \cdot \lambda, \phi_d \cdot \kappa_d = \eta_d \cdot \lambda, \\
\beta_8 \cdot \phi_8 \cdot \kappa_8 + \beta_3 \cdot \phi_3 \cdot \kappa_3 + \beta_4 \cdot \phi_4 \cdot \kappa_4 + \beta_6 \cdot \phi_6 \cdot \kappa_6 + \beta_{a1} \cdot \phi_{a1} \cdot \kappa_{a1} \\
-\beta_7 \cdot \phi_7 \cdot \kappa_7 - \beta_9 \cdot \phi_9 \cdot \kappa_9 = J_E^{(N)}, \\
J_E^{(N)} = \eta_E \cdot \lambda, \lambda = \phi_R \cdot \kappa_t, J_r^{(N)} = \phi_4 \cdot \kappa_4, J_f^{(N)} = \phi_6 \cdot \kappa_6, \\
\phi_R + \phi_{py} + \phi_3 + \phi_4 + \phi_5 + \phi_6 + \phi_7 + \phi_8 + \phi_9 + \phi_{a1} + \phi_{a2} + \phi_b + \phi_c + \phi_d = \phi_{max},
\end{cases}
\tag{S88}
$$

where $\eta_E = r_E \cdot \left[\sum_i r_i / N_{EP_i}^{carbon}\right]^{-1}$. $\kappa_i$ is approximately a constant which follows *Equation S20* for each of the intermediate node. The substrate quality of $\kappa_{py}$ varies with the external concentration of pyruvate ([py]),

$$\kappa_{py} \equiv \frac{r_{protein}}{r_{carbon}} \cdot \frac{k_{py}^{cat}}{m_{E_{py}}} \cdot \frac{[py]}{[py] + K_{py}} \cdot m_0. \tag{S89}$$

From *Equation S88*, all $\phi_i$ can be expressed by $J_r^{(N)}$, $J_f^{(N)}$, and $\lambda$:

$$\begin{cases} \phi_{py} = \left[\left(2\eta_{a1} + \eta_{a2} + \eta_b + 2\eta_c + \eta_d\right)\lambda + J_r^{(N)} + J_f^{(N)}\right]/\kappa_{py}, \\[2mm] \phi_7 = \left(2\eta_{a1} + \eta_{a2} + \eta_c + \eta_d\right)\lambda/\kappa_7, \phi_9 = \eta_{a1} \cdot \lambda/\kappa_9 \\[2mm] \phi_8 = \left[J_r^{(N)} + J_f^{(N)} + \left(\eta_b + \eta_c\right)\lambda\right]/\kappa_8 \\[2mm] \phi_3 = \left(J_r^{(N)} + \eta_c \cdot \lambda\right)/\kappa_3, \phi_4 = J_r^{(N)}/\kappa_4, \\[2mm] \phi_5 = \left(\eta_c + \eta_d\right)\lambda/\kappa_5, \phi_6 = J_f^{(N)}/\kappa_6, \\[2mm] \phi_i = \eta_i \cdot \lambda/\kappa_i \quad (i = a1, a2, b, c, d). \end{cases} \tag{S90}$$

By substituting *Equation S90* into *Equation S88*, we have:

$$\begin{cases} J_r^{(E,py)} + J_f^{(E,py)} = \varphi_{py} \cdot \lambda, \\[2mm] \dfrac{J_r^{(E,py)}}{\varepsilon_r^{(py)}} + \dfrac{J_f^{(E,py)}}{\varepsilon_f^{(py)}} = \phi_{max} - \psi_{py} \cdot \lambda. \end{cases} \tag{S91}$$

Here, $J_r^{(E,py)}$ and $J_f^{(E,py)}$ stand for the normalized energy fluxes of respiration and fermentation, respectively, with

$$\begin{cases} J_r^{(E,py)} = \beta_r^{(py)} \cdot J_r^{(N)}, \\[2mm] J_f^{(E,py)} = \beta_f^{(py)} \cdot J_f^{(N)}. \end{cases} \tag{S92}$$

where $\beta_r^{(py)} = \beta_3 + \beta_4 + \beta_8$ and $\beta_f^{(py)} = \beta_6 + \beta_8$, with $\beta_r^{(py)} = 10$ and $\beta_f^{(py)} = 3$ for *E. coli*. The coefficients $\varepsilon_r^{(py)}$ and $\varepsilon_f^{(py)}$ represent the proteome efficiencies for energy biogenesis using pyruvate in respiration and fermentation pathways, respectively, with

$$\begin{cases} \varepsilon_r^{(py)} = \dfrac{\beta_r^{(py)}}{1/\kappa_{py} + 1/\kappa_8 + 1/\kappa_3 + 1/\kappa_4}, \\[3mm] \varepsilon_f^{(py)} = \dfrac{\beta_f^{(py)}}{1/\kappa_{py} + 1/\kappa_8 + 1/\kappa_6}. \end{cases} \tag{S93}$$

$\psi_{py}^{-1}$ is the proteome efficiency for biomass generation using pyruvate in the biomass synthesis pathway, with

$$\psi_{py} = \frac{1}{\kappa_t} + \frac{1 + \eta_{a1} + \eta_c}{\kappa_{py}} + \frac{1 - \eta_b + \eta_{a1}}{\kappa_7} + \frac{\eta_b + \eta_c}{\kappa_8} + \frac{\eta_{a1}}{\kappa_9} + \frac{\eta_c}{\kappa_3} + \frac{\eta_c + \eta_d}{\kappa_5} + \sum_i^{a1,a2,b,c,d} \frac{\eta_i}{\kappa_i} \tag{S94}$$

$\varphi_{py}$ is an energy demand coefficient (a constant), with

$$\varphi_{py} \equiv \eta_E + \beta_7 \cdot \left(1 - \eta_b + \eta_{a1}\right) + \beta_9 \cdot \eta_{a1} - \beta_8 \cdot \left(\eta_c + \eta_b\right) - \beta_3 \cdot \eta_c - \beta_{a1} \cdot \eta_{a1}, \tag{S95}$$

Evidently, *Equation S91* is identical in form with *Equation S29*. The growth rate changes into $\kappa_{\mathrm{py}}$ dependent:

$$
\lambda\left(\kappa_{\mathrm{py}}\right) =
\begin{cases}
\dfrac{\phi_{\max}}{\varphi_{\mathrm{py}}/\varepsilon_r^{(\mathrm{py})}\left(\kappa_{\mathrm{py}}\right) + \psi_{\mathrm{py}}\left(\kappa_{\mathrm{py}}\right)} & \varepsilon_r^{(\mathrm{py})}\left(\kappa_{\mathrm{py}}\right) > \varepsilon_f^{(\mathrm{py})}\left(\kappa_{\mathrm{py}}\right), \\[2em]
\dfrac{\phi_{\max}}{\varphi_{\mathrm{py}}/\varepsilon_f^{(\mathrm{py})}\left(\kappa_{\mathrm{py}}\right) + \psi_{\mathrm{py}}\left(\kappa_{\mathrm{py}}\right)} & \varepsilon_r^{(\mathrm{py})}\left(\kappa_{\mathrm{py}}\right) < \varepsilon_f^{(\mathrm{py})}\left(\kappa_{\mathrm{py}}\right).
\end{cases}
\tag{S96}
$$

When $\kappa_{\mathrm{py}}$ is very small, combined with *Equation S93*, then,

$$
\begin{cases}
\varepsilon_r^{(\mathrm{py})}\left(\kappa_{\mathrm{py}} \to 0\right) \approx \beta_r^{(\mathrm{py})} \cdot \kappa_{\mathrm{py}}, \\[0.5em]
\varepsilon_f^{(\mathrm{py})}\left(\kappa_{\mathrm{py}} \to 0\right) \approx \beta_f^{(\mathrm{py})} \cdot \kappa_{\mathrm{py}}.
\end{cases}
\tag{S97}
$$

Obviously, $\beta_r^{(\mathrm{py})} \gg \beta_f^{(\mathrm{py})}$, and hence

$$
\varepsilon_r^{(\mathrm{py})}\left(\kappa_{\mathrm{py}} \to 0\right) > \varepsilon_f^{(\mathrm{py})}\left(\kappa_{\mathrm{py}} \to 0\right).
\tag{S98}
$$

As long as

$$
\frac{\beta_r^{(\mathrm{py})} - \beta_f^{(\mathrm{py})}}{\kappa_{\mathrm{py}}^{(\mathrm{ST})}} < \beta_f^{(\mathrm{py})}\left(\frac{1}{\kappa_8} + \frac{1}{\kappa_3} + \frac{1}{\kappa_4}\right) - \beta_r^{(\mathrm{py})} \cdot \left(\frac{1}{\kappa_8} + \frac{1}{\kappa_6}\right),
\tag{S99}
$$

where the superscript '(ST)' stands for the saturated concentration, then,

$$
\varepsilon_r^{(\mathrm{py})}\left(\kappa_{\mathrm{py}}^{(\mathrm{ST})}\right) < \varepsilon_f^{(\mathrm{py})}\left(\kappa_{\mathrm{py}}^{(\mathrm{ST})}\right),
\tag{S100}
$$

and there exists a critical value of $\kappa_{\mathrm{py}}$, denoted as $\kappa_{\mathrm{py}}^{(\mathrm{C})}$, with

$$
\begin{cases}
\varepsilon_r^{(\mathrm{py})}\left(\kappa_{\mathrm{py}}^{(\mathrm{C})}\right) = \varepsilon_f^{(\mathrm{py})}\left(\kappa_{\mathrm{py}}^{(\mathrm{C})}\right) = \dfrac{\beta_r^{(\mathrm{py})} - \beta_f^{(\mathrm{py})}}{1/\kappa_3 + 1/\kappa_4 - 1/\kappa_6} = \dfrac{\beta_3 + \beta_4 - \beta_6}{1/\kappa_3 + 1/\kappa_4 - 1/\kappa_6}, \\[1.5em]
\lambda_{\mathrm{C}}^{(\mathrm{py})} \equiv \lambda\left(\kappa_{\mathrm{py}}^{(\mathrm{C})}\right) = \dfrac{\phi_{\max}}{\varphi_{\mathrm{py}}/\varepsilon_{r/f}^{(\mathrm{py})}\left(\kappa_{\mathrm{py}}^{(\mathrm{C})}\right) + \psi_{\mathrm{py}}\left(\kappa_{\mathrm{py}}^{(\mathrm{C})}\right)}.
\end{cases}
\tag{S101}
$$

Here, $\lambda_{\mathrm{C}}^{(\mathrm{py})}$ is the growth rate at the transition point, and $\varepsilon_{r/f}^{(\mathrm{py})}$ stands for either $\varepsilon_r^{(\mathrm{py})}$ or $\varepsilon_f^{(\mathrm{py})}$. In *Appendix 1—figure 2H*, we show the dependencies of $\varepsilon_r^{(\mathrm{py})}\left(\kappa_{\mathrm{py}}\right)$, $\varepsilon_f^{(\mathrm{py})}\left(\kappa_{\mathrm{py}}\right)$ and $\lambda\left(\kappa_{\mathrm{py}}\right)$ on $\kappa_{\mathrm{py}}$ in a 3-dimensional form. In the homogeneous case, $J_f^{(\mathrm{E,py})}$ and $J_r^{(\mathrm{E,py})}$ follow:

$$
\begin{cases}
J_f^{(\mathrm{E,py})} = \varphi_{\mathrm{py}} \cdot \lambda \cdot \theta\left(\lambda - \lambda_{\mathrm{C}}^{(\mathrm{py})}\right), \\[0.5em]
J_r^{(\mathrm{E,py})} = \varphi_{\mathrm{py}} \cdot \lambda \cdot \left[1 - \theta\left(\lambda - \lambda_{\mathrm{C}}^{(\mathrm{py})}\right)\right].
\end{cases}
\tag{S102}
$$

Defining $\lambda_{\max}^{(\mathrm{py})} = \lambda\left(\kappa_{\mathrm{py}}^{(\mathrm{ST})}\right)$, and then, $\left[0, \lambda_{\max}^{(\mathrm{py})}\right]$ is the relevant range of the x axis. To compare with experiments, we assume the same extent of extrinsic noise in $k_i^{\mathrm{cat}}$ as specified in Appendix 3.3. Then, $\lambda_{\mathrm{C}}^{(\mathrm{py})}$ approximately follows a Gaussian distribution:

$$
\lambda_{\mathrm{C}}^{(\mathrm{py})} \sim \mathcal{N}\left(\mu_{\lambda_{\mathrm{C}}^{(\mathrm{py})}}, \sigma_{\lambda_{\mathrm{C}}^{(\mathrm{py})}}^2\right),
\tag{S103}
$$

where $\mu_{\lambda_C^{(py)}}$ and $\sigma_{\lambda_C^{(py)}}$ stand for the mean and standard deviation of $\lambda_C^{(py)}$. Then, the relations between the normalized energy fluxes and growth rate are:

$$
\begin{cases}
J_f^{(E,py)}(\lambda) = \dfrac{1}{2}\varphi_{py}\cdot\lambda\cdot\left[\mathrm{erf}\left(\dfrac{\lambda-\mu_{\lambda_C^{(py)}}}{\sqrt{2}\sigma_{\lambda_C^{(py)}}}\right)+1\right], \\[3ex]
J_r^{(E,py)}(\lambda) = \dfrac{1}{2}\varphi_{py}\cdot\lambda\cdot\left[1-\mathrm{erf}\left(\dfrac{\lambda-\mu_{\lambda_C^{(py)}}}{\sqrt{2}\sigma_{\lambda_C^{(py)}}}\right)\right].
\end{cases}
\tag{S104}
$$

Combined with **Equation S92**, we have:

$$
\begin{cases}
J_f^{(N)}(\lambda) = \dfrac{\varphi_{py}}{2\beta_f^{(py)}}\cdot\lambda\cdot\left[\mathrm{erf}\left(\dfrac{\lambda-\mu_{\lambda_C^{(py)}}}{\sqrt{2}\sigma_{\lambda_C^{(py)}}}\right)+1\right], \\[3ex]
J_r^{(N)}(\lambda) = \dfrac{\varphi_{py}}{2\beta_r^{(py)}}\cdot\lambda\cdot\left[1-\mathrm{erf}\left(\dfrac{\lambda-\mu_{\lambda_C^{(py)}}}{\sqrt{2}\sigma_{\lambda_C^{(py)}}}\right)\right].
\end{cases}
\tag{S105}
$$

In **Figure 3F**, we show that the model predictions (**Equation S105**) align quantitatively with the experimental results (**Holms, 1996**).

## 5.2 Mixture of a Group A carbon source with extracellular amino acids

In the case of a Group A carbon source mixed with amino acids, the coarse-grained model is shown in **Appendix 1—figure 2A**. This model can be used to analyze mixtures with one or multiple types of extracellular amino acids. Here, **Equations S21, S22, S24 and S25** still apply, but **Equation S23** changes to (the case of $i = a1$ remains the same as **Equation S23**):

$$
\Phi_i\cdot\xi_i\cdot N_{EP_i}^{carbon} + \Phi_i'\cdot\xi_i'\cdot N_{P_i}^{carbon} = r_i\cdot J_{BM} \quad (i = a2, b, c, d).
\tag{S106}
$$

Here, $N_{P_i}^{carbon}$ represents the number of carbon atoms in a molecule of Pool $i$. For simplicity, we assume:

$$
N_{P_i}^{carbon} \approx N_{EP_i}^{carbon}.
\tag{S107}
$$

In the case where all 21 types of amino acids are present and each is at saturated concentration (denoted as '21AA'), we have:

$$
\begin{cases}
\phi_A\cdot\kappa_A = \phi_1\cdot\kappa_1 + \phi_{a1}\cdot\kappa_{a1}, \\
2\phi_1\cdot\kappa_1 = \phi_2\cdot\kappa_2 + \phi_5\cdot\kappa_5 + \phi_{a2}\cdot\kappa_{a2}, \\
\phi_2\cdot\kappa_2 = \phi_3\cdot\kappa_3 + \phi_6\cdot\kappa_6 + \phi_b\cdot\kappa_b, \\
\phi_5\cdot\kappa_5 + \phi_4\cdot\kappa_4 = \phi_3\cdot\kappa_3 + \phi_d\cdot\kappa_d, \\
\phi_3\cdot\kappa_3 = \phi_4\cdot\kappa_4 + \phi_c\cdot\kappa_c, \\
\phi_{a1}\cdot\kappa_{a1} = \eta_{a1}\cdot\lambda, \phi_{a2}\cdot\kappa_{a2} + \phi_{a2}^{(21AA)}\cdot\kappa_{a2}^{(21AA)} = \eta_{a2}\cdot\lambda, \phi_b\cdot\kappa_b + \phi_b^{(21AA)}\cdot\kappa_b^{(21AA)} = \eta_b\cdot\lambda, \\
\phi_c\cdot\kappa_c + \phi_c^{(21AA)}\cdot\kappa_c^{(21AA)} = \eta_c\cdot\lambda, \phi_d\cdot\kappa_d + \phi_d^{(21AA)}\cdot\kappa_d^{(21AA)} = \eta_d\cdot\lambda, \\
\beta_1\cdot\phi_1\cdot\kappa_1 + \beta_2\cdot\phi_2\cdot\kappa_2 + \beta_3\cdot\phi_3\cdot\kappa_3 + \beta_4\cdot\phi_4\cdot\kappa_4 + \beta_6\cdot\phi_6\cdot\kappa_6 + \beta_{a1}\cdot\phi_{a1}\cdot\kappa_{a1} = J_E^{(N)}, \\
J_E^{(N)} = \eta_E\cdot\lambda, \lambda = \phi_R\cdot\kappa_t, J_r^{(N)} = \phi_4\cdot\kappa_4, J_f^{(N)} = \phi_6\cdot\kappa_6, \\
\phi_R + \phi_A + \displaystyle\sum_i^6 \phi_i + \sum_j^{a1,a2,b,c,d} \phi_j + \phi_{a2}^{(21AA)} + \phi_b^{(21AA)} + \phi_c^{(21AA)} + \phi_d^{(21AA)} = \phi_{max},
\end{cases}
\tag{S108}
$$

where $\phi_i$ and $\kappa_i$ are defined following **Equations S9 and S12**. Since the cell growth rate significantly increases with the mixture of amino acids, we deduce that Pools a2-d are supplied by amino acids in growth optimization, with

$$\phi_i = 0 \quad (i = a2, b, c, d). \tag{S109}$$

Amino acids should be more efficient in the supply of biomass synthesis than the Group A carbon source for Pools a2-d, i.e.,

$$\begin{cases} 1/\kappa_{a2}^{(21AA)} < 1/\kappa_{a2} + 1/\left(2\kappa_1\right) + 1/\left(2\kappa_A\right), \\ 1/\kappa_b^{(21AA)} < 1/\kappa_b + 1/\kappa_2 + 1/\left(2\kappa_1\right) + 1/\left(2\kappa_A\right), \\ 1/\kappa_c^{(21AA)} < 1/\kappa_c + 1/\kappa_5 + 1/\kappa_3 + 1/\kappa_2 + 1/\kappa_1 + 1/\kappa_A, \\ 1/\kappa_d^{(21AA)} < 1/\kappa_d + 1/\kappa_5 + 1/\left(2\kappa_1\right) + 1/\left(2\kappa_A\right). \end{cases} \tag{S110}$$

In practice, the requirement for proteome efficiency when using amino acids is even higher, since the biomass synthesis pathway is accompanied by energy biogenesis for Group A carbon sources, but not for amino acids. Combining *Equations S108 and S109*, we have:

$$\begin{cases} J_r^{(E)} + J_f^{(E)} = \varphi_{21AA} \cdot \lambda, \\ \dfrac{J_r^{(E)}}{\varepsilon_r} + \dfrac{J_f^{(E)}}{\varepsilon_f} = \phi_{\max} - \psi_{21AA} \cdot \lambda, \end{cases} \tag{S111}$$

where $J_r^{(E)}$, $J_f^{(E)}$ follow *Equation S30*, while $\varepsilon_r$ and $\varepsilon_f$ follow *Equation S31*. $\psi_{21AA}^{-1}$ is the proteome efficiency for biomass generation in the biomass synthesis pathway under this nutrient condition, with

$$\psi_{21AA} = \frac{1}{\kappa_t} + \frac{\eta_{a1}}{\kappa_A} + \frac{\eta_{a1}}{\kappa_{a1}} + \frac{\eta_{a2}}{\kappa_{a2}^{(21AA)}} + \frac{\eta_b}{\kappa_b^{(21AA)}} + \frac{\eta_c}{\kappa_c^{(21AA)}} + \frac{\eta_d}{\kappa_d^{(21AA)}} \tag{S112}$$

$\varphi_{21AA}$ is an energy demand coefficient, with

$$\varphi_{21AA} \equiv \eta_E - \beta_{a1} \cdot \eta_{a1} \tag{S113}$$

Combining *Equations S111 and S31*, the formula for the growth rate is:

$$\lambda\left(\kappa_A\right) = \begin{cases} \lambda_r^{(21AA)} = \dfrac{\phi_{\max}}{\varphi_{21AA}/\varepsilon_r\left(\kappa_A\right) + \psi_{21AA}\left(\kappa_A\right)} & \varepsilon_r\left(\kappa_A\right) > \varepsilon_f\left(\kappa_A\right), \\[3mm] \lambda_f^{(21AA)} = \dfrac{\phi_{\max}}{\varphi_{21AA}/\varepsilon_f\left(\kappa_A\right) + \psi_{21AA}\left(\kappa_A\right)} & \varepsilon_r\left(\kappa_A\right) < \varepsilon_f\left(\kappa_A\right). \end{cases} \tag{S114}$$

In fact, *Equations S37-S42* still apply. $\varepsilon_{r/f}\left(\kappa_A^{(C)}\right)$ satisfies *Equation S43*, while $\lambda_C^{(21AA)} \equiv \lambda\left(\kappa_A^{(C)}\right)$ and $\lambda_{\max}^{(21AA)} \equiv \lambda\left(\kappa_A^{\max}\right)$ are:

$$\begin{cases} \lambda_C^{(21AA)} = \dfrac{\phi_{\max}}{\dfrac{\varphi_{21AA}}{\varepsilon_{r/f}\left(\kappa_A^{(C)}\right)} + \psi_{21AA}\left(\kappa_A^{(C)}\right)}, \\[5mm] \lambda_{\max}^{(21AA)} = \dfrac{\phi_{\max}}{\dfrac{\varphi_{21AA}}{\varepsilon_f\left(\kappa_A^{\max}\right)} + \psi_{21AA}\left(\kappa_A^{\max}\right)}. \end{cases} \tag{S115}$$

When extrinsic noise is taken into account, $\lambda_C^{(21AA)}$ approximately follows a Gaussian distribution:

$$\lambda_C^{(21AA)} \sim \mathcal{N}\left(\mu_{\lambda_C^{(21AA)}}, \sigma_{\lambda_C^{(21AA)}}^2\right), \tag{S116}$$

and the normalized fluxes $J_r^{(N)}$, $J_f^{(N)}$ change to:

$$\begin{cases} J_f^{(N)}(\lambda) = \dfrac{\varphi_{21AA}}{\beta_f^{(A)}} \cdot \lambda \cdot \left[ \mathrm{erf}\left( \dfrac{\lambda - \mu_{\lambda_C^{(21AA)}}}{\sqrt{2}\sigma_{\lambda_C^{(21AA)}}} \right) + 1 \right], \\[3mm] J_r^{(N)}(\lambda) = \dfrac{\varphi_{21AA}}{\beta_r^{(A)}} \cdot \lambda \cdot \left[ 1 - \mathrm{erf}\left( \dfrac{\lambda - \mu_{\lambda_C^{(21AA)}}}{\sqrt{2}\sigma_{\lambda_C^{(21AA)}}} \right) \right]. \end{cases} \tag{S117}$$

The above analysis can be extended to cases where a Group A carbon source is mixed with arbitrary combinations of amino acids. *Equations S111, S114-S117* would remain in a similar form, while *Equations S112-S113* would change depending on the combinations of amino acid. In *Appendix 1—figure 2B and C*, we compare model predictions (see also Appendix 8.2 and *Equation S157*) with experimental data (*Basan et al., 2015*; *Wallden et al., 2016*) from mixtures of 21 or 7 types of amino acids along with a Group A carbon source, demonstrating quantitative agreement. Additionally, the increase in the critical threshold of growth rate for the growth rate-dependent fermentation flux in mixtures with extracellular amino acids (i.e. $\lambda_C^{(21AA)}, \lambda_C^{(7AA)} > \lambda_C$; see *Appendix 1—figure 2C*) has also been observed in other experimental findings (*Peebo et al., 2015*).

## Appendix 6

### Enzyme allocation upon perturbations

#### 6.1 Carbon limitation within Group A carbon sources

In *Equation S28*, we present the model predictions for the dependencies of enzyme proteomic mass fractions on growth rate and energy fluxes. To compare with experiments, we assume the same extent of extrinsic noise in $k_i^{\text{cat}}$ as specified in Appendix 3.3. Relative protein expression data for enzymes within glycolysis and the TCA cycle are available from existing studies and are comparable to the $\phi_1$-$\phi_4$ enzymes of our model (*Figure 1B*). Upon $\kappa_A$ perturbation, $\kappa_A$ is a variable while $w_0$ is fixed (see Appendix 2.5). Combining *Equations S28 and S47* (with $w_0 = 0$), we obtain:

$$\begin{cases} \phi_1 = \dfrac{\lambda}{\kappa_1} \left\{ \dfrac{\varphi \cdot \left( \beta_r^{(A)} - \beta_f^{(A)} \right)}{2\beta_r^{(A)} \cdot \beta_f^{(A)}} \cdot \left[ \text{erf}\left( \dfrac{\lambda - \mu_{\lambda_C}}{\sqrt{2}\sigma_{\lambda_C}} \right) + 1 \right] + \dfrac{\varphi}{\beta_r^{(A)}} + \dfrac{\eta_{a2} + \eta_b + 2\eta_c + \eta_d}{2} \right\}, \\[3mm] \phi_2 = \dfrac{\lambda}{\kappa_2} \left\{ \dfrac{\varphi \cdot \left( \beta_r^{(A)} - \beta_f^{(A)} \right)}{\beta_r^{(A)} \cdot \beta_f^{(A)}} \cdot \left[ \text{erf}\left( \dfrac{\lambda - \mu_{\lambda_C}}{\sqrt{2}\sigma_{\lambda_C}} \right) + 1 \right] + \dfrac{2\varphi}{\beta_r^{(A)}} + \eta_b + \eta_c \right\}, \\[3mm] \phi_3 = \dfrac{\lambda}{\kappa_3} \left\{ \dfrac{\varphi}{\beta_r^{(A)}} \cdot \left[ 1 - \text{erf}\left( \dfrac{\lambda - \mu_{\lambda_C}}{\sqrt{2}\sigma_{\lambda_C}} \right) \right] + \eta_c \right\}, \\[3mm] \phi_4 = \dfrac{\lambda}{\kappa_4} \cdot \dfrac{\varphi}{\beta_r^{(A)}} \cdot \left[ 1 - \text{erf}\left( \dfrac{\lambda - \mu_{\lambda_C}}{\sqrt{2}\sigma_{\lambda_C}} \right) \right]. \end{cases} \tag{S118}$$

In *Appendix 1—figure 3C and D*, we show the comparisons between model predictions (*Equation S118*, $w_0 = 0$) and experimental data (*Hui et al., 2015*), which are consistent overall. We then consider the influence of maintenance energy as specified in Appendix 4.2. Here, we continue to choose $w_0 = 2.5 \left( h^{-1} \right)$ as previously adopted in Appendix 4.3. Thus, *Equation S28* still holds. Combined with *Equation S85* under the condition that $\iota = 0$, we have:

$$\begin{cases} \phi_1 = \dfrac{1}{2 \cdot \kappa_1} \left\{ \begin{array}{l} \dfrac{\varphi \cdot \lambda + w_0}{\beta_f^{(A)}} \cdot \left[ \text{erf}\left( \dfrac{\lambda - \mu_{\lambda_C}}{\sqrt{2}\sigma_{\lambda_C}} \right) + 1 \right] + \dfrac{\varphi \cdot \lambda + w_0}{\beta_r^{(A)}} \cdot \left[ 1 - \text{erf}\left( \dfrac{\lambda - \mu_{\lambda_C}}{\sqrt{2}\sigma_{\lambda_C}} \right) \right] \\[3mm] + (\eta_{a2} + \eta_b + 2\eta_c + \eta_d)\, \lambda \end{array} \right\}, \\[6mm] \phi_2 = \dfrac{1}{\kappa_2} \left\{ \dfrac{\varphi \cdot \lambda + w_0}{\beta_f^{(A)}} \cdot \left[ \text{erf}\left( \dfrac{\lambda - \mu_{\lambda_C}}{\sqrt{2}\sigma_{\lambda_C}} \right) + 1 \right] + \dfrac{\varphi \cdot \lambda + w_0}{\beta_r^{(A)}} \cdot \left[ 1 - \text{erf}\left( \dfrac{\lambda - \mu_{\lambda_C}}{\sqrt{2}\sigma_{\lambda_C}} \right) \right] + (\eta_b + \eta_c)\, \lambda \right\} \\[3mm] \phi_3 = \dfrac{1}{\kappa_3} \left\{ \dfrac{\varphi \cdot \lambda + w_0}{\beta_r^{(A)}} \cdot \left[ 1 - \text{erf}\left( \dfrac{\lambda - \mu_{\lambda_C}}{\sqrt{2}\sigma_{\lambda_C}} \right) \right] + \eta_c \cdot \lambda \right\}, \\[3mm] \phi_4 = \dfrac{1}{\kappa_4} \cdot \dfrac{\varphi \cdot \lambda + w_0}{\beta_r^{(A)}} \cdot \left[ 1 - \text{erf}\left( \dfrac{\lambda - \mu_{\lambda_C}}{\sqrt{2}\sigma_{\lambda_C}} \right) \right]. \end{cases}$$

$$\tag{S119}$$

In *Figure 4A–B*, we show that the model predictions (*Equation S119*, $w_0 = 2.5\,(h^{-1})$) generally agree with the experiments (*Hui et al., 2015*). However, there are different basal expressions of these enzymes, likely due to living demands other than cell proliferation, such as preparation for starvation (*Mori et al., 2017*) or changes in the type of the nutrient (*Basan et al., 2020*; *Kussell and Leibler, 2005*).

#### 6.2 Overexpression of useless proteins

In the case of $\phi_Z$ perturbation under each nutrient condition with fixed $\kappa_A$ (see Appendix 4.1), we consider the same extent of extrinsic noise in $k_i^{\text{cat}}$ as specified in Appendix 3.3. The relation between enzyme allocation and growth rate can be obtained by combining *Equations S28 and S58* (with $w_0 = 0$):

$$
\begin{cases}
\phi_1 = \dfrac{\lambda}{2 \cdot \kappa_1} \left\{ \begin{array}{l} (\eta_{a2} + \eta_b + 2\eta_c + \eta_d) + \dfrac{\varphi}{\beta_r^{(A)}} \cdot \left[ 1 - \mathrm{erf}\left( \dfrac{\lambda(\kappa_A, 0) - \mu_{\lambda_C}(0)}{\sqrt{2}\sigma_{\lambda_C}(0)} \right) \right] \\[3mm] + \dfrac{\varphi}{\beta_f^{(A)}} \cdot \left[ \mathrm{erf}\left( \dfrac{\lambda(\kappa_A, 0) - \mu_{\lambda_C}(0)}{\sqrt{2}\sigma_{\lambda_C}(0)} \right) + 1 \right] \end{array} \right\}, \\[12mm]
\phi_2 = \dfrac{\lambda}{\kappa_2} \left\{ \begin{array}{l} (\eta_b + \eta_c) + \dfrac{\varphi}{\beta_r^{(A)}} \cdot \left[ 1 - \mathrm{erf}\left( \dfrac{\lambda(\kappa_A, 0) - \mu_{\lambda_C}(0)}{\sqrt{2}\sigma_{\lambda_C}(0)} \right) \right] \\[3mm] + \dfrac{\varphi}{\beta_f^{(A)}} \cdot \left[ \mathrm{erf}\left( \dfrac{\lambda(\kappa_A, 0) - \mu_{\lambda_C}(0)}{\sqrt{2}\sigma_{\lambda_C}(0)} \right) + 1 \right] \end{array} \right\}, \\[12mm]
\phi_3 = \dfrac{\lambda}{\kappa_3} \left\{ \dfrac{\varphi}{\beta_r^{(A)}} \cdot \left[ 1 - \mathrm{erf}\left( \dfrac{\lambda(\kappa_A, 0) - \mu_{\lambda_C}(0)}{\sqrt{2}\sigma_{\lambda_C}(0)} \right) \right] + \eta_c \right\}, \\[8mm]
\phi_4 = \dfrac{\lambda}{\kappa_4} \left\{ \dfrac{\varphi}{\beta_r^{(A)}} \cdot \left[ 1 - \mathrm{erf}\left( \dfrac{\lambda(\kappa_A, 0) - \mu_{\lambda_C}(0)}{\sqrt{2}\sigma_{\lambda_C}(0)} \right) \right] \right\}.
\end{cases}
\tag{S120}
$$

Here $\lambda(\kappa_A, 0)$ is the growth rate for $\phi_Z = 0$, and thus it is a parameter rather than a variable. The growth rate is defined as $\lambda(\kappa_A, \phi_Z)$, which follows *Equation S50*. Thus, $\phi_i$ is proportional to the growth rate $\lambda$. In *Appendix 1—figure 3E and F*, we observe that the model predictions (*Equation S120*) generally agree with the experiments (*Basan et al., 2015*). Next, we consider the influence of maintenance energy with $w_0 = 2.5 \ \left( h^{-1} \right)$. Combining *Equations S28, S58 and S85* (with $\iota = 0$), we get:

$$
\begin{cases}
\phi_1 = \dfrac{w_0}{2\kappa_1} \left\{ \dfrac{1}{\beta_f^{(A)}} \cdot \left[ \mathrm{erf}\left( \dfrac{\lambda(\kappa_A, 0) - \mu_{\lambda_C}(0)}{\sqrt{2}\sigma_{\lambda_C}(0)} \right) + 1 \right] + \dfrac{1}{\beta_r^{(A)}} \cdot \left[ 1 - \mathrm{erf}\left( \dfrac{\lambda(\kappa_A, 0) - \mu_{\lambda_C}(0)}{\sqrt{2}\sigma_{\lambda_C}(0)} \right) \right] \right\} \\[5mm]
+ \dfrac{\lambda}{2\kappa_1} \left\{ \begin{array}{l} \dfrac{\varphi}{\beta_f^{(A)}} \cdot \left[ \mathrm{erf}\left( \dfrac{\lambda(\kappa_A, 0) - \mu_{\lambda_C}(0)}{\sqrt{2}\sigma_{\lambda_C}(0)} \right) + 1 \right] + \dfrac{\varphi}{\beta_r^{(A)}} \cdot \left[ 1 - \mathrm{erf}\left( \dfrac{\lambda(\kappa_A, 0) - \mu_{\lambda_C}(0)}{\sqrt{2}\sigma_{\lambda_C}(0)} \right) \right] \\[3mm] + (\eta_{a2} + \eta_b + 2\eta_c + \eta_d) \end{array} \right\}, \\[12mm]
\phi_2 = \dfrac{w_0}{\kappa_2} \left\{ \dfrac{1}{\beta_f^{(A)}} \cdot \left[ \mathrm{erf}\left( \dfrac{\lambda(\kappa_A, 0) - \mu_{\lambda_C}(0)}{\sqrt{2}\sigma_{\lambda_C}(0)} \right) + 1 \right] + \dfrac{1}{\beta_r^{(A)}} \cdot \left[ 1 - \mathrm{erf}\left( \dfrac{\lambda(\kappa_A, 0) - \mu_{\lambda_C}(0)}{\sqrt{2}\sigma_{\lambda_C}(0)} \right) \right] \right\} \\[5mm]
+ \dfrac{\lambda}{\kappa_2} \left\{ \begin{array}{l} \dfrac{\varphi}{\beta_f^{(A)}} \cdot \left[ \mathrm{erf}\left( \dfrac{\lambda(\kappa_A, 0) - \mu_{\lambda_C}(0)}{\sqrt{2}\sigma_{\lambda_C}(0)} \right) + 1 \right] + \dfrac{\varphi}{\beta_r^{(A)}} \cdot \left[ 1 - \mathrm{erf}\left( \dfrac{\lambda(\kappa_A, 0) - \mu_{\lambda_C}(0)}{\sqrt{2}\sigma_{\lambda_C}(0)} \right) \right] \\[3mm] + (\eta_b + \eta_c) \end{array} \right\}, \\[12mm]
\phi_3 = \dfrac{\lambda}{\kappa_3} \left\{ \dfrac{\varphi}{\beta_r^{(A)}} \cdot \left[ 1 - \mathrm{erf}\left( \dfrac{\lambda(\kappa_A, 0) - \mu_{\lambda_C}(0)}{\sqrt{2}\sigma_{\lambda_C}(0)} \right) \right] + \eta_c \right\} \\[5mm]
+ \dfrac{w_0}{\kappa_3} \cdot \dfrac{1}{\beta_r^{(A)}} \cdot \left[ 1 - \mathrm{erf}\left( \dfrac{\lambda(\kappa_A, 0) - \mu_{\lambda_C}(0)}{\sqrt{2}\sigma_{\lambda_C}(0)} \right) \right], \\[8mm]
\phi_4 = \dfrac{\lambda}{\kappa_4} \cdot \dfrac{\varphi}{\beta_r^{(A)}} \cdot \left[ 1 - \mathrm{erf}\left( \dfrac{\lambda(\kappa_A, 0) - \mu_{\lambda_C}(0)}{\sqrt{2}\sigma_{\lambda_C}(0)} \right) \right] + \dfrac{w_0}{\kappa_4} \cdot \dfrac{1}{\beta_r^{(A)}} \cdot \left[ 1 - \mathrm{erf}\left( \dfrac{\lambda(\kappa_A, 0) - \mu_{\lambda_C}(0)}{\sqrt{2}\sigma_{\lambda_C}(0)} \right) \right].
\end{cases}
\tag{S121}
$$

Here, the growth rate is defined as $\lambda(\kappa_A, \phi_Z)$, and $\lambda(\kappa_A, 0)$ is a parameter rather than a variable. Thus, $\phi_i$ is a linear function of the growth rate $\lambda$, with a positive slope and a positive y-intercept. In *Figure 4C–D* and *Appendix 1—figure 3I-J*, we show that the model predictions (*Equation S121*) agree quantitively with the experimental data (*Basan et al., 2015*).

## 6.3 Energy dissipation

In the case of energy dissipation under each nutrient condition, $w$ is perturbed while $\kappa_A$ is fixed. The relation between protein allocation and growth rate can be obtained by combining *Equations*

*S28* and *S70*. However, since $w$ is explicitly present in *Equation S70*, it is necessary to reduce this variable to obtain the growth rate dependence of enzyme allocation. From *Equation S64*, we have:

$$\lambda\left(\kappa_A, w\right) = \lambda\left(\kappa_A, 0\right)\left\{1 - \frac{w}{\phi_{\max}} \cdot \left[\frac{1}{\varepsilon_r\left(\kappa_A\right)} - \theta\left(\varepsilon_f\left(\kappa_A\right) - \varepsilon_r\left(\kappa_A\right)\right) \cdot \left(\frac{1}{\varepsilon_r\left(\kappa_A\right)} - \frac{1}{\varepsilon_f\left(\kappa_A\right)}\right)\right]\right\}.$$

(S122)

Here, $\lambda\left(\kappa_A, 0\right) \equiv \lambda\left(\kappa_A, w = 0\right)$ (satisfying *Equation S64*) is a parameter rather than a variable. '$\theta$' stands for the Heaviside step function. Thus, we have:

$$w\left(\lambda\right) = \frac{\phi_{\max} \cdot \left[1 - \lambda/\lambda\left(\kappa_A, 0\right)\right]}{\left[1/\varepsilon_r\left(\kappa_A\right) - \theta\left(\varepsilon_f\left(\kappa_A\right) - \varepsilon_r\left(\kappa_A\right)\right) \cdot \left(1/\varepsilon_r\left(\kappa_A\right) - 1/\varepsilon_f\left(\kappa_A\right)\right)\right]},$$

(S123)

where the energy dissipation coefficient $w$ is regarded as a function of the growth rate.

Combining *Equations S28, S70* and *S123*, we get:

$$\begin{cases}
\phi_1 = \frac{1}{2\kappa_1}\left\{\mathrm{erf}\left(\frac{\lambda - \mu_{\lambda_C}(0)\left[1 - \frac{w(\lambda)}{\varepsilon_{r/f}\left(\kappa_A^{(C)}\right)\phi_{\max}}\right]}{\sqrt{2}\sigma_{\lambda_C}(0)\left[1 - \frac{w(\lambda)}{\varepsilon_{r/f}\left(\kappa_A^{(C)}\right)\phi_{\max}}\right]}\right) \cdot \left[\frac{\varphi \cdot \lambda + w(\lambda)}{\beta_f^{(A)}} - \frac{\varphi \cdot \lambda + w(\lambda)}{\beta_r^{(A)}}\right] \right. \\
\left. \qquad + \left[\varphi \cdot \lambda + w(\lambda)\right]\left(\frac{1}{\beta_f^{(A)}} - \frac{1}{\beta_r^{(A)}} + \frac{2}{\beta_r^{(A)}}\right) + \left(\eta_{a2} + \eta_b + 2\eta_c + \eta_d\right) \cdot \lambda\right\}, \\[2em]
\phi_2 = \frac{1}{\kappa_2}\left\{\mathrm{erf}\left(\frac{\lambda - \mu_{\lambda_C}(0)\left[1 - \frac{w(\lambda)}{\varepsilon_{r/f}\left(\kappa_A^{(C)}\right)\phi_{\max}}\right]}{\sqrt{2}\sigma_{\lambda_C}(0)\left[1 - \frac{w(\lambda)}{\varepsilon_{r/f}\left(\kappa_A^{(C)}\right)\phi_{\max}}\right]}\right) \cdot \left[\frac{\varphi \cdot \lambda + w(\lambda)}{\beta_f^{(A)}} - \frac{\varphi \cdot \lambda + w(\lambda)}{\beta_r^{(A)}}\right] \right. \\
\left. \qquad + \left[\varphi \cdot \lambda + w(\lambda)\right]\left(\frac{1}{\beta_f^{(A)}} - \frac{1}{\beta_r^{(A)}} + \frac{2}{\beta_r^{(A)}}\right) + \left(\eta_b + \eta_c\right) \cdot \lambda\right\}, \\[2em]
\phi_3 = \frac{1}{\kappa_3}\left\{\frac{\left[\varphi \cdot \lambda + w(\lambda)\right]}{\beta_r^{(A)}} \cdot \left[1 - \mathrm{erf}\left(\frac{\lambda - \mu_{\lambda_C}(0)\left[1 - \frac{w(\lambda)}{\varepsilon_{r/f}\left(\kappa_A^{(C)}\right)\phi_{\max}}\right]}{\sqrt{2}\sigma_{\lambda_C}(0)\left[1 - \frac{w(\lambda)}{\varepsilon_{r/f}\left(\kappa_A^{(C)}\right)\phi_{\max}}\right]}\right)\right] + \eta_c \cdot \lambda\right\}, \\[2em]
\phi_4 = \frac{1}{\kappa_4} \cdot \frac{\left[\varphi \cdot \lambda + w(\lambda)\right]}{\beta_r^{(A)}} \cdot \left[1 - \mathrm{erf}\left(\frac{\lambda - \mu_{\lambda_C}(0)\left[1 - \frac{w(\lambda)}{\varepsilon_{r/f}\left(\kappa_A^{(C)}\right)\phi_{\max}}\right]}{\sqrt{2}\sigma_{\lambda_C}(0)\left[1 - \frac{w(\lambda)}{\varepsilon_{r/f}\left(\kappa_A^{(C)}\right)\phi_{\max}}\right]}\right)\right],
\end{cases}$$

(S124)

where $w\left(\lambda\right)$ follows *Equation S123*. When $\kappa_A$ lies in the vicinity of $\kappa_A^{(C)}$ or $w$ is small so that

$$\left(1 - \frac{w}{\varepsilon_{rlf}\left(\kappa_A\right) \cdot \phi_{\max}}\right) \Big/ \left(1 - \frac{w}{\varepsilon_{rlf}\left(\kappa_A^{(C)}\right) \cdot \phi_{\max}}\right) \approx 1 \tag{S125}$$

then we have:

$$\begin{cases} J_f^{(N)}\left(\lambda, w\right) = \dfrac{\varphi \cdot \lambda + w}{\beta_f{}^{(A)}} \cdot \left[\mathrm{erf}\left(\dfrac{\lambda\left(\kappa_A, 0\right) - \mu_{\lambda_C}\left(0\right)}{\sqrt{2}\sigma_{\lambda_C}\left(0\right)}\right) + 1\right], \\[3mm] J_r^{(N)}\left(\lambda, w\right) = \dfrac{\varphi \cdot \lambda + w}{\beta_r^{(A)}} \cdot \left[1 - \mathrm{erf}\left(\dfrac{\lambda\left(\kappa_A, 0\right) - \mu_{\lambda_C}\left(0\right)}{\sqrt{2}\sigma_{\lambda_C}\left(0\right)}\right)\right], \end{cases} \tag{S126}$$

and thus:

$$\begin{cases} \phi_1 = \dfrac{1}{2\kappa_1}\left\{ \begin{array}{l} \left[\varphi \cdot \lambda + w\left(\lambda\right)\right]\left(\dfrac{1}{\beta_f{}^{(A)}} - \dfrac{1}{\beta_r^{(A)}}\right) \cdot \left[\mathrm{erf}\left(\dfrac{\lambda\left(\kappa_A, 0\right) - \mu_{\lambda_C}\left(0\right)}{\sqrt{2}\sigma_{\lambda_C}\left(0\right)}\right) + 1\right] \\[3mm] + \dfrac{2}{\beta_r^{(A)}}\left[\varphi \cdot \lambda + w\left(\lambda\right)\right] + \left(\eta_{a2} + \eta_b + 2\eta_c + \eta_d\right) \cdot \lambda \end{array} \right\}, \\[8mm] \phi_2 = \dfrac{1}{\kappa_2}\left\{ \begin{array}{l} \left[\varphi \cdot \lambda + w\left(\lambda\right)\right]\left(\dfrac{1}{\beta_f{}^{(A)}} - \dfrac{1}{\beta_r^{(A)}}\right) \cdot \left[\mathrm{erf}\left(\dfrac{\lambda\left(\kappa_A, 0\right) - \mu_{\lambda_C}\left(0\right)}{\sqrt{2}\sigma_{\lambda_C}\left(0\right)}\right) + 1\right] \\[3mm] + \dfrac{2}{\beta_r^{(A)}}\left[\varphi \cdot \lambda + w\left(\lambda\right)\right] + \left(\eta_b + \eta_c\right) \cdot \lambda \end{array} \right\}, \\[8mm] \phi_3 = \dfrac{1}{\kappa_3}\left\{ \dfrac{\left[\varphi \cdot \lambda + w\left(\lambda\right)\right]}{\beta_r^{(A)}} \cdot \left[1 - \mathrm{erf}\left(\dfrac{\lambda\left(\kappa_A, 0\right) - \mu_{\lambda_C}\left(0\right)}{\sqrt{2}\sigma_{\lambda_C}\left(0\right)}\right)\right] + \eta_c \cdot \lambda \right\}, \\[5mm] \phi_4 = \dfrac{1}{\kappa_4} \cdot \dfrac{\left[\varphi \cdot \lambda + w\left(\lambda\right)\right]}{\beta_r^{(A)}} \cdot \left[1 - \mathrm{erf}\left(\dfrac{\lambda\left(\kappa_A, 0\right) - \mu_{\lambda_C}\left(0\right)}{\sqrt{2}\sigma_{\lambda_C}\left(0\right)}\right)\right], \end{cases} \tag{s127}$$

Note that in *Equation S123*, $w$ is a linear function of $\lambda$ with a negative slope. Thus $\phi_i$ exhibits a linear relation with $\lambda$ when *Equation S125* is satisfied (see *Equation S127*). In fact, the slope of $\phi_4$ is certainly negative (combining *Equations S64, S123 and S127*), while the sign of the slope for other $\phi_i$ depends on parameters. For a given nutrient, the enzymes corresponding to the same $\phi_i$ should exhibit the same slope sign. Another restriction is that if the slope sign of $\phi_1$ is negative, then the slope sign of $\phi_2$ is surely negative. In *Appendix 1—figure 3K-N*, we show that our model results agree well with the experimental data (*Basan et al., 2015*; *Equation S127*).

## Appendix 7

## Other aspects of the model

### 7.1 A coarse-grained model with more details

To compare with experiments, we consider a coarse-grained model with more details, as shown in *Appendix 1—figure 2F*. Here, nodes $M_6$, $M_7$ represent GA3P and DHAP, respectively. Other nodes follow the descriptions specified in Appendix 3.1. Each biochemical reaction follows *Equation S5* with $b_i = 1$ except that $M_1 \rightarrow M_6 + M_7$ and $M_3 + M_5 \rightarrow M_4$. By applying flux balance to the stoichiometric fluxes, combined with *Equation S8*, we obtain:

$$
\begin{cases}
\Phi_A \cdot \xi_A = \Phi_1 \cdot \xi_1 + \Phi_{a1} \cdot \xi_{a1}, \\
\Phi_{11} \cdot \xi_{11} = \Phi_{10} \cdot \xi_{10} + \Phi_1 \cdot \xi_1, \Phi_{10} \cdot \xi_{10} = \Phi_1 \cdot \xi_1, \\
\Phi_{11} \cdot \xi_{11} = \Phi_2 \cdot \xi_2 + \Phi_5 \cdot \xi_5 + \Phi_{a2} \cdot \xi_{a2}, \\
\Phi_2 \cdot \xi_2 = \Phi_3 \cdot \xi_3 + \Phi_6 \cdot \xi_6 + \Phi_b \cdot \xi_b, \\
\Phi_5 \cdot \xi_5 + \Phi_4 \cdot \xi_4 = \Phi_3 \cdot \xi_3 + \Phi_d \cdot \xi_d, \\
\Phi_3 \cdot \xi_3 = \Phi_4 \cdot \xi_4 + \Phi_c \cdot \xi_c.
\end{cases}
\tag{S128}
$$

While *Equations S22-S25* still hold. By applying the substitutions specified in *Equations S9, S12, S14-S18*, combined with *Equations S4, S10, S22-S25, S128*, and the constraint of proteome resource allocation, we get:

$$
\begin{cases}
\phi_A \cdot \kappa_A = \phi_1 \cdot \kappa_1 + \phi_{a1} \cdot \kappa_{a1}, \\
\phi_{11} \cdot \kappa_{11} = \phi_{10} \cdot \kappa_{10} + \phi_1 \cdot \kappa_1, \phi_{10} \cdot \kappa_{10} = \phi_1 \cdot \kappa_1, \\
\phi_{11} \cdot \kappa_{11} = \phi_2 \cdot \kappa_2 + \phi_5 \cdot \kappa_5 + \phi_{a2} \cdot \kappa_{a2}, \\
\phi_2 \cdot \kappa_2 = \phi_3 \cdot \kappa_3 + \phi_6 \cdot \kappa_6 + \phi_b \cdot \kappa_b, \\
\phi_5 \cdot \kappa_5 + \phi_4 \cdot \kappa_4 = \phi_3 \cdot \kappa_3 + \phi_d \cdot \kappa_d, \\
\phi_3 \cdot \kappa_3 = \phi_4 \cdot \kappa_4 + \phi_c \cdot \kappa_c, \\
\phi_{a1} \cdot \kappa_{a1} = \eta_{a1} \cdot \lambda, \phi_{a2} \cdot \kappa_{a2} = \eta_{a2} \cdot \lambda, \phi_b \cdot \kappa_b = \eta_b \cdot \lambda, \phi_c \cdot \kappa_c = \eta_c \cdot \lambda, \phi_d \cdot \kappa_d = \eta_d \cdot \lambda, \\
\beta_1 \cdot \phi_1 \cdot \kappa_1 + \beta_2 \cdot \phi_2 \cdot \kappa_2 + \beta_3 \cdot \phi_3 \cdot \kappa_3 + \beta_4 \cdot \phi_4 \cdot \kappa_4 + \beta_6 \cdot \phi_6 \cdot \kappa_6 + \beta_{a1} \cdot \phi_{a1} \cdot \kappa_{a1} = J_E^{(N)}, \\
J_E^{(N)} = \eta_E \cdot \lambda, \lambda = \phi_R \cdot \kappa_t, J_r^{(N)} = \phi_4 \cdot \kappa_4, J_f^{(N)} = \phi_6 \cdot \kappa_6, \\
\phi_R + \phi_A + \phi_1 + \phi_2 + \phi_3 + \phi_4 + \phi_5 + \phi_6 + \phi_7 + \phi_8 + \phi_{a1} + \phi_{a2} + \phi_b + \phi_c + \phi_d = \phi_{max}.
\end{cases}
\tag{S129}
$$

Then, *Equation S28* still holds, while $\phi_{10}$ and $\phi_{11}$ are:

$$
\begin{cases}
\phi_{10} = \left[ J_r^{(N)} + J_f^{(N)} + \left( \eta_{a2} + \eta_b + 2\eta_c + \eta_d \right) \lambda \right] / 2 \cdot \kappa_{10} \left( 2 \cdot \kappa_{10} \right), \\
\phi_{11} = \left[ J_r^{(N)} + J_f^{(N)} + \left( \eta_{a2} + \eta_b + 2\eta_c + \eta_d \right) \lambda \right] / \kappa_{11}.
\end{cases}
\tag{S130}
$$

By substituting *Equations S28 and S130* into *Equation S129*, we get:

$$
\begin{cases}
J_r^{(E)} + J_f^{(E)} = \varphi \cdot \lambda, \\
\dfrac{J_r^{(E)}}{\varepsilon_r^{(dt)}} + \dfrac{J_f^{(E)}}{\varepsilon_f^{(dt)}} = \phi_{max} - \psi_{dt} \cdot \lambda,
\end{cases}
\tag{S131}
$$

where 'dt' stands for details. *Equations S30 and S33* still hold. $\varepsilon_r^{(dt)}$ and $\varepsilon_f^{(dt)}$ represent the proteome efficiencies for energy biogenesis in the respiration and fermentation pathways, respectively, with

$$
\begin{cases}
\varepsilon_r^{(\mathrm{dt})} = \dfrac{\beta_r^{(A)}}{1/\kappa_A + 1/\kappa_1 + 1/\kappa_{10} + 2/\kappa_{11} + 2/\kappa_2 + 2/\kappa_3 + 2/\kappa_4}, \\[4mm]
\varepsilon_f^{(\mathrm{dt})} = \dfrac{\beta_f^{(A)}}{1/\kappa_A + 1/\kappa_1 + 1/\kappa_{10} + 2/\kappa_{11} + 2/\kappa_2 + 2/\kappa_6}.
\end{cases}
\tag{S132}
$$

$\psi_{\mathrm{dt}}^{-1}$ is the proteome efficiency for biomass generation in the biomass synthesis pathway, with

$$
\psi_{\mathrm{dt}} = \frac{1}{\kappa_t} + \frac{1 + \eta_{a1} + \eta_c}{2\kappa_A} + \left(\eta_{a2} + \eta_b + 2\eta_c + \eta_d\right)\left(\frac{1}{2\kappa_1} + \frac{1}{2\kappa_{10}} + \frac{1}{\kappa_{11}}\right)
$$
$$
+ \frac{\eta_b + \eta_c}{\kappa_2} + \frac{\eta_c}{\kappa_3} + \frac{\eta_c + \eta_d}{\kappa_5} + \sum_i^{a1,a2,b,c,d} \frac{\eta_i}{\kappa_i}.
\tag{S133}
$$

## 7.2 Estimation of the in vivo enzyme catalytic rates

We use the method introduced by **Davidi et al., 2016**, combined with proteome experimental data (**Basan et al., 2015**; **Appendix 1—table 2**), to estimate the in vivo enzyme catalytic rates. Combining **Equations S28 and S130**, we have:

$$
\begin{cases}
\kappa_1 = \left[J_r^{(N)} + J_f^{(N)} + \left(\eta_{a2} + \eta_b + 2\eta_c + \eta_d\right)\lambda\right]/\left(2 \cdot \phi_1\right), \\[3mm]
\kappa_2 = \left[J_r^{(N)} + J_f^{(N)} + \left(\eta_b + \eta_c\right)\lambda\right]/\phi_2, \\[3mm]
\kappa_3 = \left(J_r^{(N)} + \eta_c \cdot \lambda\right)/\phi_3, \kappa_4 = J_r^{(N)}/\phi_4, \\[3mm]
\kappa_5 = \left(\eta_c + \eta_d\right)\lambda/\phi_5, \kappa_6 = J_f^{(N)}/\phi_6 \\[3mm]
\kappa_{10} = \left[J_r^{(N)} + J_f^{(N)} + \left(\eta_{a2} + \eta_b + 2\eta_c + \eta_d\right)\lambda\right]/\left(2 \cdot \phi_{10}\right), \\[3mm]
\kappa_{11} = \left[J_r^{(N)} + J_f^{(N)} + \left(\eta_{a2} + \eta_b + 2\eta_c + \eta_d\right)\lambda\right]/\phi_{11}.
\end{cases}
\tag{S134}
$$

Here, $J_r^{(N)}$, $J_f^{(N)}$, $\lambda$ and $\phi_i$ ($i$ = 1-6,10-11) are measurable from experiments (see Appendix 9.1 and **Appendix 1—table 2**). Thus, we can obtain the in vivo values of $\kappa_i$ from **Equation S134**. Combined with **Equations S17 and S20**, we have

$$
k_i^{\mathrm{cat}} = \frac{r_{\mathrm{carbon}}}{r_{\mathrm{protein}}} \cdot \frac{m_{E_i}}{m_{\mathrm{carbon}}} \cdot \kappa_i \cdot \left[\sum_i r_i / N_{\mathrm{EP}_i}^{\mathrm{carbon}}\right]
\tag{S135}
$$

**Equation S135** is the in vivo result for the enzyme catalytic rate. In **Appendix 1—figure 2G**, we show a comparison between in vivo and in vitro results for $k_{\mathrm{cat}}$ values of enzymes within glycolysis and the TCA cycle, which are roughly consistent. In the applications, we prioritized the use of in vivo results for enzyme catalytic rates, and use in vitro data as a substitute when there were gaps.

## 7.3 Comparison with existing models that illustrate experimental results

For the coarse-grained model described in Appendix 3, the normalized stoichiometric influx of a Group A carbon source is given by:

$$
J_{\mathrm{in}}^{(N)} \equiv J_A^{(N)} = \phi_A \cdot \kappa_A.
\tag{S136}
$$

Combined with the first equation in **Equation S28** and **Equation S30**, we obtain:

$$
J_{\mathrm{in}}^{(N)} - \vartheta \cdot \lambda = \frac{J_r^{(E)}}{\beta_r^{(A)}} + \frac{J_f^{(E)}}{\beta_f^{(A)}},
\tag{S137}
$$

where $\vartheta = \eta_{a1} + \eta_c + (\eta_{a2} + \eta_b + \eta_d)/2$. Evidently, $\beta_r^{(A)}$, $\beta_f^{(A)}$ and $\vartheta$ are constant parameters. In this subsection, we highlight the major differences between our model presented in Appendix 3 and existing models that illustrate the growth rate dependence of fermentation flux in the standard picture of overflow metabolism (**Basan et al., 2015**; **Holms, 1996**; **Meyer et al., 1984**; **Nanchen et al., 2006**).

Based on the modeling principles rather than the detailed mechanisms, there are two major classes of existing models that can illustrate experimental results. Both classes of models regard the proteome efficiencies $\varepsilon_r$ and $\varepsilon_f$ as constants, with $\varepsilon_f > \varepsilon_r$ if used, or follow functionally equivalent propositions. However, in our model, $\varepsilon_r$ and $\varepsilon_f$ are both functions of $\kappa_A$, which vary significantly upon nutrient perturbation, with $\varepsilon_r(\kappa_A \to 0) > \varepsilon_f(\kappa_A \to 0)$ and $\varepsilon_r(\kappa_A^{\max}) < \varepsilon_f(\kappa_A^{\max})$ (see **Equations S38, S40-S41**). Furthermore, there are significant differences in the modeling and optimization principles, as listed below.

The first class of models (**Chen and Nielsen, 2019**; **Majewski and Domach, 1990**; **Shlomi et al., 2011**; **Varma and Palsson, 1994**; **Vazquez et al., 2010**; **Vazquez and Oltvai, 2016**; **Zhuang et al., 2011**) optimize the ratio of biomass outflow to carbon influx $\lambda / J_{\mathrm{in}}^{(\mathrm{N})}$, either to optimize the growth rate for a given carbon influx or to minimize the carbon influx for a given growth rate. Since respiration is far more efficient than fermentation in terms of energy biogenesis per unit carbon, to optimize the ratio $\lambda / J_{\mathrm{in}}^{(\mathrm{N})}$, cells would preferentially use respiration when the carbon influx is small. As carbon influx increases above a certain threshold, factors such as proteome allocation direct cells toward fermentation in a threshold-linear response, since they consider $\varepsilon_f > \varepsilon_r$. Our model is significantly different from this class of models in the optimization principle, as we purely optimize the cell growth rate for a given nutrient condition, without imposing a special constraint on the carbon influx.

The second class of models, represented by **Basan et al., 2015**, also adopt the optimization of $\lambda / J_{\mathrm{in}}^{(\mathrm{N})}$ in the interpretation of their model results. However, the growth rate dependence of fermentation flux was derived prior to the application of growth rate optimization (although it can be derived by optimizing $\lambda / J_{\mathrm{in}}^{(\mathrm{N})}$). In fact, **Equations S29 and S137** in our model are very similar in form to those in **Basan et al., 2015**, yet there are critical differences, which we list below. In **Equation S29**, by regarding $J_r^{(\mathrm{E})}$ and $J_f^{(\mathrm{E})}$ as the two variables in a system of linear equations, we obtain the following expressions:

$$
\begin{cases}
J_r^{(\mathrm{E})} = \dfrac{\phi_{\max} - \left(\psi + \dfrac{\varphi}{\varepsilon_f}\right) \cdot \lambda}{\dfrac{1}{\varepsilon_r} - \dfrac{1}{\varepsilon_f}}, \\[2em]
J_f^{(\mathrm{E})} = \dfrac{\left(\psi + \dfrac{\varphi}{\varepsilon_r}\right) \cdot \lambda - \phi_{\max}}{\dfrac{1}{\varepsilon_r} - \dfrac{1}{\varepsilon_f}}.
\end{cases}
\tag{S138}
$$

In **Basan et al., 2015**, **Equation S138** is considered to be the relation between $J_{rlf}^{(\mathrm{E})}$ and $\lambda$ upon nutrient (and thus $J_{\mathrm{in}}^{(\mathrm{N})}$) perturbation, while $\varepsilon_r$ and $\varepsilon_f$ are regarded as constants throughout the perturbation. By contrast, in our model, **Equation S138** serves as a constraint under a given nutrient condition with fixed $\kappa_A$, and is not relevant to nutrient perturbation. For wild-type strains, if $\varepsilon_r(\kappa_A) > \varepsilon_f(\kappa_A)$ (or vice versa), then the solution for optimal growth is $J_r^{(\mathrm{E})}(\kappa_A) = \varphi \cdot \lambda(\kappa_A)$ and $J_f^{(\mathrm{E})}(\kappa_A) = 0$, with $\lambda(\kappa_A) = \dfrac{\varepsilon_r(\kappa_A) \cdot \phi_{\max}}{\varphi + \varepsilon_r(\kappa_A) \cdot \psi(\kappa_A)}$. This solution, which satisfies **Equation S138**, corresponds to a point rather than a line in the relation between growth rate $\lambda$ and normalized energy flux $J_{rlf}^{(\mathrm{E})}$ upon $\kappa_A$ perturbation.

## Appendix 8

## Probability density functions of variables and parameters

### 8.1 Probability density function of $\kappa_i$

Enzyme catalysis is crucial for the survival of living organisms, as it significantly accelerates biochemical reactions by reducing the energy barrier between the substrate and product (**Nelson and Cox, 2008**). However, the maximal turnover rate of enzymes, $k_{cat}$, varies notably between in vivo and in vitro measurements (**Davidi et al., 2016**). Recent studies suggest that differences in the aquatic medium are the primary cause of this variation (**Davidi et al., 2016**; **García-Contreras et al., 2012**). In particular, potassium and phosphate concentrations have a significant influence on $k_{cat}$ (**García-Contreras et al., 2012**), and these concentrations exhibit some degree of variation among cell populations under intracellular conditions (**García-Contreras et al., 2012**). For simplicity, we assume that the turnover rate of each enzyme $E_i$, $k_i^{cat}$, follows a Gaussian distribution $\mathcal{N}\left(\mu_{k_i^{cat}}, \sigma_{k_i^{cat}}^2\right)$ with $k_i^{cat} > 0$ among cells (representing extrinsic noise [**Elowitz et al., 2002**], denoted as $\chi_{ext}$). The probability density function of $k_i^{cat}$ is then given by:

$$k_i^{cat} \sim \mathcal{N}'\left(x; \mu_{k_i^{cat}}, \sigma_{k_i^{cat}}^2\right) = \begin{cases} l\dfrac{1}{\sigma_{k_i^{cat}}\sqrt{2\pi}} e^{-\frac{1}{2}\left(\frac{x - \mu_{k_i^{cat}}}{\sigma_{k_i^{cat}}}\right)^2}, & x \geq 0. \\ \\ 0, & x < 0. \end{cases} \tag{S139}$$

When the CV of the $k_i^{cat}$ distribution (i.e. $\sigma_{k_i^{cat}}/\mu_{k_i^{cat}}$) is less than 1/3 , $\mathcal{N}'\left(x; \mu_{k_i^{cat}}, \sigma_{k_i^{cat}}^2\right)$ is almost identical to $\mathcal{N}\left(\mu_{k_i^{cat}}, \sigma_{k_i^{cat}}^2\right)$. In this case, $1/k_i^{cat}$ follows the positive inverse of Gaussian (IOG) distribution, and the probability density function is:

$$\text{IOG}\left(x; \mu_{1/k_i^{cat}}, \zeta_{1/k_i^{cat}}\right) = \begin{cases} \sqrt{\dfrac{\zeta_{1/k_i^{cat}}}{2\pi x^4}}\exp\left(-\dfrac{1}{2}\dfrac{\zeta_{1/k_i^{cat}}\left(x - \mu_{1/k_i^{cat}}\right)^2}{x^2\mu_{1/k_i^{cat}}^2}\right), & x \geq 0, \\ \\ 0, & x < 0, \end{cases} \tag{S140}$$

where $\zeta_{1/k_i^{cat}} = 1/\sigma_{k_i^{cat}}^2$ is the shape parameter, and $\mu_{1/k_i^{cat}} = 1/\mu_{k_i^{cat}}$ is the mean.

Meanwhile, due to the stochastic nature of biochemical reactions, we apply Gillespie's chemical Langevin equation (**Gillespie, 2000**) to account for intrinsic noise (**Elowitz et al., 2002**) (denoted as $\chi_{int}$). For cell size regulation of *E. coli* within a cell cycle, the cell mass at the initiation of DNA replication per chromosome origin remains constant (**Donachie, 1968**). Thus, the time required for enzyme $E_i$ to complete a catalytic job (with a timescale of $1/k_i^{cat}$) can be approximated as the first passage time of a stochastic process, with

$$\begin{cases} X_i(t = 0) = 0, \\ dX_i/dt = \alpha_i + \sqrt{\alpha_i}\Gamma_i(t), \\ T_\Theta = \inf\{t > 0 | X_i(t) = \Theta\}. \end{cases} \tag{S141}$$

Here $\alpha_i \equiv k_i^{cat} \cdot \Theta$, where $\Theta$ is proportional to the cell volume, and $\Gamma_i(t)$ represents independent, temporally uncorrelated Gaussian white noise. Then, for a given value of $k_i^{cat}$, the first passage time $T_\Theta$ follows an Inverse Gaussian (IG) distribution (**Folks and Chhikara, 1978**):

$$\text{IG}\left(x; \mu'_{1/k_i^{cat}}, \zeta'_{1/k_i^{cat}}\right) = \begin{cases} \sqrt{\dfrac{\zeta'_{1/k_i^{cat}}}{2\pi x^3}}\exp\left(-\dfrac{1}{2}\dfrac{\zeta'_{1/k_i^{cat}}\left(x - \mu'_{1/k_i^{cat}}\right)^2}{x\mu'^2_{1/k_i^{cat}}}\right), & x \geq 0, \\ \\ 0, & x < 0, \end{cases} \tag{S142}$$

where $\zeta'_{1/k_i^{cat}} = \Theta/k_i^{cat}$ is the shape parameter, and $\mu'_{1/k_i^{cat}} = 1/k_i^{cat}$ represents the mean. The variance of this distribution is $\sigma'^2_{1/k_i^{cat}} \equiv \mu'^3_{1/k_i^{cat}}/\zeta'_{1/k_i^{cat}} = 1/\left[\Theta \cdot \left(k_i^{cat}\right)^2\right]$. Thus, we can obtain the CV:

$$\sigma'_{1/k_i^{\text{cat}}} / \mu'_{1/k_i^{\text{cat}}} = \Theta^{-\frac{1}{2}}, \tag{S143}$$

which is inversely proportional to the square root of cell volume. Evidently, the intrinsic and extrinsic noise make orthogonal contributions to the total noise (*Elowitz et al., 2002*) (denoted as $\chi_{\text{tot}}$):

$$\chi_{\text{tot}}^2 = \chi_{\text{int}}^2 + \chi_{\text{ext}}^2. \tag{S144}$$

In fact, when the CV is small (i.e. CV <<1), both the IOG and IG distributions converge into Gaussian distributions (*Appendix 1—figure 4*). In the back-of-the-envelope calculations, we approximate $x$ in all denominator terms of IOG $(x; \mu, \zeta)$ and IG $(x; \mu, \zeta)$ as µ (since CV <<1). Then, both the IOG and IG distributions can be approximated as follows:

$$\text{IOG}\left(x; \mu_{1/k_i^{\text{cat}}}, \zeta_{1/k_i^{\text{cat}}}\right) \xrightarrow{\text{CV} \ll 1} \mathcal{N}\left(\mu_{1/k_i^{\text{cat}}}, \sigma_{1/k_i^{\text{cat}}}^2\right), \tag{S145}$$

with a variance of $\sigma_{1/k_i^{\text{cat}}}^2 = \mu_{1/k_i^{\text{cat}}}^4 / \zeta_{1/k_i^{\text{cat}}}$, and

$$\text{IG}\left(x; \mu'_{1/k_i^{\text{cat}}}, \zeta'_{1/k_i^{\text{cat}}}\right) \xrightarrow{\text{CV} \ll 1} \mathcal{N}\left(\mu'_{1/k_i^{\text{cat}}}, \sigma'^2_{1/k_i^{\text{cat}}}\right), \tag{S146}$$

with a variance of $\sigma'^2_{1/k_i^{\text{cat}}} = \mu'^3_{1/k_i^{\text{cat}}} / \zeta'_{1/k_i^{\text{cat}}}$. Rigorously, we show below that IG $(x; \mu, \zeta)$ shrinks to be $\mathcal{N}\left(\mu, \mu^3/\zeta\right)$ when the CV is small. For the IG distribution, the characteristic function of the variable $x$ is given by *Folks and Chhikara, 1978*; *Kampen, 1992*:

$$G\left(k\right) = \int_{-\infty}^{\infty} e^{ikx} \cdot \text{IG}\left(x; \mu, \zeta\right) dx = \exp\left\{\frac{\zeta}{\mu}\left[1 - \sqrt{1 - \frac{2i\mu^2 k}{\zeta}}\right]\right\}, \tag{S147}$$

and therefore,

$$\text{IG}\left(x; \mu, \zeta\right) = \frac{1}{2\pi} \int_{-\infty}^{\infty} e^{-ikx} \cdot G\left(k\right) dk. \tag{S148}$$

When the variance $\sigma^2 \equiv \mu^3/\zeta$ is very small, we essentially require $2\mu^2 k/\zeta = 2\sigma^2 k/\mu \ll 1$, and then $\sqrt{1 - \frac{2i\mu^2 k}{\zeta}} \approx 1 - \frac{\mu^2}{\zeta}ki + \frac{\mu^4}{2\zeta^2}k^2$. Thus,

$$\begin{cases} G(k) \approx \exp\left(\mu k i - \frac{\mu^3}{2\zeta}k^2\right), \\ \text{IG}(x; \mu, \zeta) \approx \sqrt{\frac{\zeta}{2\pi\mu^3}} \exp\left(-\frac{\zeta(x-\mu)^2}{2\mu^3}\right) = \mathcal{N}\left(\mu, \frac{\mu^3}{\zeta}\right). \end{cases} \tag{S149}$$

This leads to:

$$\lim_{\sigma \to 0} \text{IG}\left(x; \mu, \zeta\right) = \mathcal{N}\left(\mu, \mu^3/\zeta\right). \tag{S150}$$

In fact, intrinsic noise does affect the short-term measurement of enzyme catalytic rate and growth rate at the single-cell level. However, its contribution in the long term is averaged out and thus becomes negligible. For simplicity, we approximate $\chi_{\text{tot}} \approx \chi_{\text{ext}}$. Combined with *Equations S145-S146*, it is straightforward to verify that $1/k_i^{\text{cat}}$ shares roughly the same CV as $k_i^{\text{cat}}$:

$$\sigma_{1/k_i^{\text{cat}}} / \mu_{1/k_i^{\text{cat}}} = \sigma_{k_i^{\text{cat}}} / \mu_{k_i^{\text{cat}}}. \tag{S151}$$

For convenience, in the model analysis, we approximate both IOG and IG distributions as Gaussian distributions. Then, all $1/k_i^{\text{cat}}$ are independent, normally distributed random variables following Gaussian distributions:

$$1/k_i^{\text{cat}} \sim \mathcal{N}\left(\mu_{1/k_i^{\text{cat}}}, \sigma_{1/k_i^{\text{cat}}}^2\right). \tag{S152}$$

Using the properties of Gaussian distributions, for a series of constant real numbers $\gamma_i$, the summation of $\gamma_i/k_i^{\text{cat}}$, which we define as $\Xi \equiv \sum_{i=1}^{n} \gamma_i/k_i^{\text{cat}}$, follows a Gaussian distribution (**Kampen, 1992**):

$$\Xi \sim \mathcal{N}\left(\mu_\Xi, \sigma_\Xi^2\right), \tag{S153}$$

with $\mu_\Xi = \sum_{i=1}^{n} \gamma_i \mu_{1/k_i^{\text{cat}}}$ and $\sigma_\Xi^2 = \sum_{i=1}^{n} \left(\gamma_i \sigma_{1/k_i^{\text{cat}}}\right)^2$. The relation between $\kappa_i$ and $k_i^{\text{cat}}$ is shown in **Equation S12**. To optimize cell growth rate, each $\kappa_i$ of the intermediate nodes satisfies **Equation S20**, while $\kappa_A$ satisfies **Equation S27**. Thus, for a given nutrient condition ([A] is fixed), all the ratios $k_i^{\text{cat}}/\kappa_i$ are constants. Combined with **Equations S139, S145-S146, and S152**, the distributions of all $\kappa_i$ and $1/\kappa_i$ can be approximated as Gaussian distributions:

$$\begin{cases} \kappa_i \sim \mathcal{N}\left(\mu_{\kappa_i}, \sigma_{\kappa_i}^2\right), \\ \dfrac{1}{\kappa_i} \sim \mathcal{N}\left(\mu_{1/\kappa_i}, \sigma_{1/\kappa_i}^2\right), \end{cases} \tag{S154}$$

where $\mu_{\kappa_i}$ and $\mu_{1/\kappa_i}$ are the means of $\kappa_i$ and $1/\kappa_i$, and $\sigma_{\kappa_i}$ and $\sigma_{1/\kappa_i}$ are their standard deviations. Using the properties of Gaussian distributions, combined with **Equation S31, S32, S36, S42-S43, S145-S146 and S153**, $\varepsilon_r$, $\varepsilon_f$, $\psi$, $\lambda_r$, $\lambda_f$, $\kappa_A^{(C)}$ and $\lambda_C$ also roughly follow Gaussian distributions.

## 8.2 Probability density function of the growth rate $\lambda$

From Appendix 8.1, we note that $\lambda_r$ and $\lambda_f$ (see **Equation S36**) roughly follow Gaussian distributions, with

$$\begin{cases} \lambda_r \sim \mathcal{N}\left(\mu_{\lambda_r}, \sigma_{\lambda_r}^2\right), \\ \lambda_f \sim \mathcal{N}\left(\mu_{\lambda_f}, \sigma_{\lambda_f}^2\right), \end{cases} \tag{S155}$$

where $\mu_{\lambda_{r/f}}$ and $\sigma_{\lambda_{r/f}}$ represent the mean and standard deviation, respectively. We further assume that the correlation between $\lambda_r$ and $\lambda_f$ is $\rho_{rf}$. From **Equation S36**, we see that the growth rate $\lambda$ takes the maximum of $\lambda_r$ and $\lambda_f$, i.e.,

$$\lambda = \max\left(\lambda_r, \lambda_f\right). \tag{S156}$$

Then, the cumulative distribution function of $\lambda$ is $P\left(\lambda \leq x\right) = \int_{-\infty}^{x} \int_{-\infty}^{x} f\left(x_1, x_2\right) dx_1 dx_2$, where

$$\begin{aligned} f\left(x_1, x_2\right) = &\frac{\left(1 - \rho_{rf}^2\right)^{-\frac{1}{2}}}{2\pi \sigma_{\lambda_r} \sigma_{\lambda_f}} \exp\left(-\frac{1}{2\left(1 - \rho_{rf}^2\right)}\left[\left(\frac{x_1 - \mu_{\lambda_r}}{\sigma_{\lambda_r}}\right)^2\right.\right. \\ &\left.\left. - 2\rho_{rf}\left(\frac{x_1 - \mu_{\lambda_r}}{\sigma_{\lambda_r}}\right)\left(\frac{x_2 - \mu_{\lambda_f}}{\sigma_{\lambda_f}}\right) + \left(\frac{x_2 - \mu_{\lambda_f}}{\sigma_{\lambda_f}}\right)^2\right]\right). \end{aligned}$$

Thus, the probability density function of the growth rate $\lambda$ is given by:

$$\begin{aligned} f_\lambda\left(x\right) = &\frac{1}{2\sqrt{2\pi}\sigma_{\lambda_r}} e^{-\frac{1}{2}\left(\frac{x - \mu_{\lambda_r}}{\sigma_{\lambda_r}}\right)^2}\left[\text{erf}\left(\frac{\left(x - \mu_{\lambda_f}\right)\sigma_{\lambda_r} - \rho_{rf}\sigma_{\lambda_f}\left(x - \mu_{\lambda_r}\right)}{\sigma_{\lambda_r}\sigma_{\lambda_f}\sqrt{2\left(1 - \rho_{rf}^2\right)}}\right) + 1\right] \\ &+ \frac{1}{2\sqrt{2\pi}\sigma_{\lambda_f}} e^{-\frac{1}{2}\left(\frac{x - \mu_{\lambda_f}}{\sigma_{\lambda_f}}\right)^2}\left[\text{erf}\left(\frac{\left(x - \mu_{\lambda_r}\right)\sigma_{\lambda_f} - \rho_{rf}\sigma_{\lambda_r}\left(x - \mu_{\lambda_f}\right)}{\sigma_{\lambda_r}\sigma_{\lambda_f}\sqrt{2\left(1 - \rho_{rf}^2\right)}}\right) + 1\right]. \end{aligned} \tag{S157}$$

In *Appendix 1—figure 2B*, we show that *Equation S157* quantitatively matches the experimental data for *E. coli* under the relevant conditions.

## Appendix 9

### Model comparison with experiments on *E. coli*

#### 9.1 Flux comparison with experiments on *E. coli*

In Appendix 7.2, we see that the values of $J_f^{(N)}$ and $J_r^{(N)}$ are required to calculate the in vivo enzyme catalytic rates of the intermediate nodes. Here, we use $J_{\text{acetate}}$ and $J_{\text{CO}_2,r}$ to represent the stoichiometric fluxes of acetate from the fermentation pathway and $CO_2$ from the respiration pathway, respectively. Combined with the stoichiometric coefficients of both pathways, we have:

$$\begin{cases} J_{\text{acetate}} = J_f, \\ J_{\text{CO}_2,r} = 3 \cdot J_r. \end{cases} \tag{S158}$$

By further combining with *Equations S16-S17*, we get:

$$\begin{cases} J_f^{(N)} = J_{\text{acetate}} \cdot \dfrac{m_{\text{carbon}}}{M_{\text{carbon}}} \cdot \left[ \sum_i r_i / N_{\text{EP}_i}^{\text{carbon}} \right]^{-1}, \\ J_r^{(N)} = \dfrac{1}{3} \cdot J_{\text{CO}_2,r} \cdot \dfrac{m_{\text{carbon}}}{M_{\text{carbon}}} \cdot \left[ \sum_i r_i / N_{\text{EP}_i}^{\text{carbon}} \right]^{-1}. \end{cases} \tag{S159}$$

In fact, the values of $J_{\text{acetate}}$ and $J_{\text{CO}_2,r}$ scale with the mass of the 'big cell,' which increases over time. In experiments, the measurable fluxes are typically expressed in the unit of mM/OD$_{600}$/h (*Basan et al., 2015*). Thus, we define $J_{\text{acetate}}^{(M)}$ and $J_{\text{CO}_2,r}^{(M)}$ as the fluxes of $J_{\text{acetate}}$ and $J_{\text{CO}_2,r}$ (per biomass) in the unit of mM/OD$_{600}$/h, respectively. The superscript '(M)' represents the measurable flux in this unit. For *E. coli*, we use the following biochemical data collected from published literature: 1 OD$_{600}$ roughly corresponds to $6 \times 10^8$ cells/mL (*Stevenson et al., 2016*), the average mass of a cell is 1 pg (*Milo and Phillips, 2015*), the biomass percentage of the cell weight is 30% (*Neidhardt et al., 1990*), the molar mass of carbon is 12 g (*Nelson and Cox, 2008*), $r_{\text{carbon}} = 0.48$ (*Neidhardt et al., 1990*) and $r_{\text{protein}} = 0.55$ (*Neidhardt et al., 1990*). Combined with the values of $r_i$ (see Appendix 2.2) and $N_{\text{EP}_i}^{\text{carbon}}$, where $N_{\text{EP}_{a1}}^{\text{carbon}} = 6$, $N_{\text{EP}_{a2}}^{\text{carbon}} = 3$, $N_{\text{EP}_b}^{\text{carbon}} = 3$, $N_{\text{EP}_c}^{\text{carbon}} = 5$, and $N_{\text{EP}_d}^{\text{carbon}} = 4$ (*Nelson and Cox, 2008*), we have:

$$\begin{cases} J_f^{(N)} \approx J_{\text{acetate}}^{(M)}/2, \\ J_r^{(N)} \approx J_{\text{CO}_2,r}^{(M)}/6. \end{cases} \tag{S160}$$

From *Equation S18*, we obtain the values of $\eta_i$ for each precursor pool: $\eta_{a1} = 0.15$, $\eta_{a2} = 0.30$, $\eta_b = 0.35$, $\eta_c = 0.09$, and $\eta_d = 0.11$. Still, the value of $\eta_E$ is required to compare the growth rate dependence of fermentation/respiration fluxes between model results and experiments, which we will specify in Appendix 9.2.

#### 9.2 Model parameter settings using experimental data of *E. coli*

We have collected biochemical data for *E. coli*, as shown in *Appendix 1—table 1* and *Appendix 1—table 2*, to set the model parameters. This includes the molecular weight (MW) and in vitro $k_{\text{cat}}$ values of the catalytic enzymes, as well as the proteome and flux data used to calculate the in vivo turnover numbers. To reduce measurement noise, we take the average rather than the maximum value of in vivo $k_{\text{cat}}$ from calculations using data from four cultures (see *Appendix 1—table 2*). Here, we prioritize the use of in vivo $k_{\text{cat}}$ wherever applicable unless there is a gap in the in vivo data (see *Appendix 1—table 1*).

Note that our models are coarse grained. For example, the flux $J_3$ shown in *Figure 1B* actually corresponds to three different reactions in the metabolic network (see *Figure 1A* and *Appendix 1—table 1*), which we label as $J_3^{(i)}$ ($i$=1, 2, 3). For each $J_3^{(i)}$, there are corresponding variables/parameters of $\Phi_3^{(i)}$, $\xi_3^{(i)}$, $\phi_3^{(i)}$, $\kappa_3^{(i)}$ satisfying *Equations S8, S9 and S12*, Evidently, $J_3^{(i)} = J_3$ ($i$=1, 2, 3), and it is straightforward to derive the following relation between $\kappa_3^{(i)}$ and $\kappa_3$:

$$1/\kappa_3 = \sum_{i=1}^{3} 1/\kappa_3^{(i)}. \tag{S161}$$

In fact, *Equation S161* can be generalized to determine the values of other $\kappa_i$ in the coarse-grained models combined with the biochemical data. For the coarse-grained model of Group A carbon source utilization shown in *Figure 1B*, we have the values for parameters $\kappa_i$ ($i$=1, …, 6), and then $\varepsilon_{r/f}\left(\kappa_A^{(C)}\right) = 122\left(\text{h}^{-1}\right)$. Evidently, $\varepsilon_r\left(\kappa_{\text{glucose}}^{(\text{ST})}\right) < \varepsilon_f\left(\kappa_{\text{glucose}}^{(\text{ST})}\right)$, $\varepsilon_r\left(\kappa_{\text{lactose}}^{(\text{ST})}\right) < \varepsilon_f\left(\kappa_{\text{lactose}}^{(\text{ST})}\right)$, and thus $\varepsilon_r\left(\kappa_A^{\max}\right) < \varepsilon_f\left(\kappa_A^{\max}\right)$. For pyruvate, we have $\varepsilon_{r/f}^{(\text{py})}\left(\kappa_{\text{py}}^{(C)}\right) = \varepsilon_{r/f}\left(\kappa_A^{(C)}\right) = 122\left(\text{h}^{-1}\right)$ (see *Equations S43 and S101*), and it is easy to check that $\varepsilon_r\left(\kappa_{\text{py}}^{(\text{ST})}\right) < \varepsilon_f\left(\kappa_{\text{py}}^{(\text{ST})}\right)$.

For the remaining model parameters, note that we have classified the inactive ribosomal-affiliated proteins into the Q-class, and then $\phi_{\max} = 48\%$ (*Scott et al., 2010*). The value of $\kappa_t$ is obtainable from experiments: the translation speed is 20.1aa/s (*Scott et al., 2010*), with 7336 amino acids per ribosome (*Neidhardt, 1996*) and $\varsigma \approx 1.67$ (*Neidhardt, 1996*; *Scott et al., 2010*) (see Appendix 2.1), hence $\kappa_t = 1/610\left(\text{s}^{-1}\right)$. However, there are insufficient data to determine the values of $\kappa_i$ ($i$=a1, a2, b, c, d) for the metabolites between the entry point metabolites shown in *Figure 1A* to the precursor pools. These processes involves many steps, so these values are expected to be quite large. Here, we combine the contributions of $\kappa_t$ and $\kappa_i$ ($i$=a1, a2, b, c, d) by defining a composite parameter:

$$\Omega \equiv 1/\kappa_t + \sum_{i}^{a1,a2,b,c,d} \eta_i/\kappa_i. \tag{S162}$$

We proceed to estimate the values of $\Omega$ and $\varphi$ using experimental data (*Basan et al., 2015*) for wild-type strains on the $J_{\text{acetate}}^{(M)}$-$\lambda$ relation (*Figure 1C*), and then all the remaining model parameters are set accordingly.

For the case of $w_0 = 0$, where all $k_{\text{cat}}$ values follow a Gaussian distribution with an extrinsic noise of 25% CV (which is the general setting we use unless otherwise specified), we have $\varphi = 10.8$ and $\Omega = 1345$ (s). Accordingly, we obtain $\eta_E = 14.78$, $\mu_{\lambda_C} = 0.92\left(\text{h}^{-1}\right)$, and $\sigma_{\lambda_C} = 0.12\mu_{\lambda_C}$, where the CV of the extrinsic noise for $\Omega$ is estimated using the averaged CV of other $\kappa_i$. For the translation inhibition effect of Cm, we estimate the values for $\iota$ as $\iota_{w_0=0}^{(2\mu\text{m Cm})} = 1.15$, $\iota_{w_0=0}^{(4\mu\text{m Cm})} = 2.33$, and $\iota_{w_0=0}^{(8\mu\text{m Cm})} = 6.25$, where the superscript stands for the concentration of Cm, and the subscript represents the choice of $w_0$.

For pyruvate, with the value of $\eta_E$, we get $\varphi_{\text{py}} = 14.82$. However, there is still a lack of proteome data to determine the value of $\kappa_9$, which involves many steps in the metabolic network and thus can be considerably large. Here we define another composite parameter, $\Omega'_{\text{Gg}} \equiv \left(\eta_b + \eta_c\right)/\kappa_8 + \eta_{a1}/\kappa_9$, and estimate its value as $\Omega'_{\text{Gg}} = 690$ (s) from growth rate data for *E. coli* measured under the relevant nutrient conditions (*Basan et al., 2015*), where the subscript 'Gg' stands for glucogenesis. Then, $\mu_{\lambda_C^{(\text{py})}} = 0.67\left(\text{h}^{-1}\right)$, and $\sigma_{\lambda_C^{(\text{py})}} = 0.10\mu_{\lambda_C^{(\text{py})}}$, where the same CV of extrinsic noise for $\Omega$ applies to $\Omega'_{\text{Gg}}$.

For the case of a Group A carbon source mixed with 21 amino acids (21AA, with saturated concentrations), we have $\varphi_{21\text{AA}} = 14.2$. Comparing *Equation S32* with *Equation S112*, the parameter $\Omega$ should change to $\Omega_{21\text{AA}} \equiv 1/\kappa_t + \eta_{a1}/\kappa_{a1} + \sum_{i}^{a2,b,c,d} \eta_i/\kappa_i^{(21\text{AA})}$. Obviously, $1/\kappa_t < \Omega_{21\text{AA}} < \Omega$, and we estimate $\Omega_{21\text{AA}} = 1000$ (s) from the growth rate data for *E. coli* measured under the relevant nutrient conditions (*Wallden et al., 2016*). Then, we have $\mu_{\lambda_C^{(21\text{AA})}} = 1.13\left(\text{h}^{-1}\right)$, and $\sigma_{\lambda_C^{(21\text{AA})}} = 0.12\mu_{\lambda_C^{(21\text{AA})}}$.

For the case of a Group A carbon source mixed with 7 amino acids (7AA: His, Iso, Leu, Lys, Met, Phe, and Val), similar to the roles of $\varphi_{21\text{AA}}$ and $\Omega_{21\text{AA}}$, we define $\varphi_{7\text{AA}}$ and $\Omega_{7\text{AA}}$. Using the mass fraction of the 7AA combined with *Equation S18*, we have $\varphi_{7\text{AA}} = 11.6$. For the value of $\Omega_{7\text{AA}}$, evidently, $\Omega_{21\text{AA}} < \Omega_{7\text{AA}} < \Omega$, and we estimate $\Omega_{7\text{AA}} = 1215$ (s) from growth rate data for *E. coli* measured under the relevant culture media (*Basan et al., 2015*). Then, $\mu_{\lambda_C^{(7\text{AA})}} = 0.98\left(\text{h}^{-1}\right)$, and $\sigma_{\lambda_C^{(7\text{AA})}} = 0.12\mu_{\lambda_C^{(7\text{AA})}}$.

For the case of $w_0 = 2.5\left(\text{h}^{-1}\right)$, we have $\varphi = 8.3$, and thus $\eta_E = 12.28$, while other parameters such as $\Omega$, $\mu_{\lambda_C}$ and $\sigma_{\lambda_C}$ remain the same as for $w_0 = 0$. Nevertheless, the values for $\iota$ under translation

inhibition by Cm are influenced by the choice of $w_0$, where the values of $\iota$ change to $\iota_{w_0=2.5}^{(2\mu m\ Cm)} = 1.05$, $\iota_{w_0=2.5}^{(4\mu m\ Cm)} = 2.00$, and $\iota_{w_0=2.5}^{(8\mu m\ Cm)} = 5.40$.

From Appendix 8.1–8.2, combined with *Equation S114*, the distributions of $\lambda_r^{(21AA)}$ and $\lambda_f^{(21AA)}$ can be approximated by Gaussian distributions:

$$
\begin{cases}
\lambda_r^{(21AA)} \sim \mathcal{N}\left(\mu_{\lambda_r^{(21AA)}}, \sigma_{\lambda_r^{(21AA)}}^2\right), \\
\lambda_f^{(21AA)} \sim \mathcal{N}\left(\mu_{\lambda_f^{(21AA)}}, \sigma_{\lambda_f^{(21AA)}}^2\right),
\end{cases}
\tag{S163}
$$

where $\mu_{\lambda_r^{(21AA)}}$ and $\mu_{\lambda_f^{(21AA)}}$ stand for the mean values, while $\sigma_{\lambda_r^{(21AA)}}$ and $\sigma_{\lambda_f^{(21AA)}}$ represent the standard deviations. For the case of glucose mixed with 21AA (labeled as 'Glucose + 21AA'), the distribution of the growth rate $\lambda_{glucose}^{(21AA)}$ follows *Equation S157*. With $\Omega_{21AA} = 1000$ (s), we have $\mu_{\lambda_{glucose,r}^{(21AA)}} = 1.34$ $\left(h^{-1}\right)$, $\mu_{\lambda_{glucose,f}^{(21AA)}} = 1.46\left(h^{-1}\right)$ (both definitions follow *Equation S163*), and $\rho_{rf} \approx 1.0$ (obtained from numerical results).

For the case of succinate mixed with 21AA (labeled as 'Succinate +21AA'), the respiration pathway is always more efficient since succinate lies within the TCA cycle. Thus, the cell growth rate (defined as $\lambda_{succinate}^{(21AA)}$) would take the value of the respiration one and follows a Gaussian distribution:

$$
\lambda_{succinate}^{(21AA)} \sim \mathcal{N}\left(\mu_{\lambda_{succinate}^{(21AA)}}, \sigma_{\lambda_{succinate}^{(21AA)}}^2\right).
\tag{S164}
$$

For the case where acetate is the sole carbon source, the cells exclusively use the respiration pathway, and the growth rate (defined as $\lambda_{acetate}$) follows a Gaussian distribution:

$$
\lambda_{acetate} \sim \mathcal{N}\left(\mu_{\lambda_{acetate}}, \sigma_{\lambda_{acetate}}^2\right).
\tag{S165}
$$

Using the measured growth rate data (*Wallden et al., 2016*), we estimate $\mu_{\lambda_{succinate}^{(21AA)}} = 0.67\left(h^{-1}\right)$ and $\mu_{\lambda_{acetate}} = 0.253\left(h^{-1}\right)$. To illustrate the distribution of growth rates $\lambda_{glucose}^{(21AA)}$, $\lambda_{succinate}^{(21AA)}$ and $\lambda_{acetate}$ shown in *Appendix 1—figure 2B*, if no other source of noise existed, extrinsic noise with a CV of 40% would be required for each $k_{cat}$ value. Then, $\sigma_{\lambda_{glucose,r}^{(21AA)}} \approx 0.21\mu_{\lambda_{glucose,r}^{(21AA)}}$, $\sigma_{\lambda_{glucose,f}^{(21AA)}} \approx 0.23\mu_{\lambda_{glucose,f}^{(21AA)}}$, $\sigma_{\lambda_{succinate}^{(21AA)}} = 0.22\mu_{\lambda_{succinate}^{(21AA)}}$, and $\sigma_{\lambda_{acetate}} = 0.22\mu_{\lambda_{acetate}}$. Allowing for the possibility that intrinsic noise may also play a non-negligible role in the observed single-cell growth rate (which is not a long-term average), we still use extrinsic noise with a CV of 25% for the model results of *E. coli*, except for those shown in *Appendix 1—figure 2B*.

## Appendix 10

### Explanation of the Crabtree effect in yeast and the Warburg effect in tumors

Our model, along with the analysis presented in Appendix 3, can be extended with modifications to explain the Crabtree effect in yeast and the Warburg effect in tumors. In both cases, the optimization objective remains maximizing the cell growth rate. Consequently, yeast and tumor cells use the most efficient pathway for ATP production at the single-cell level.

For model applications in yeast or tumor cell metabolism, the fermentation flux shifts from acetate secretion to ethanol and lactate secretion, respectively (see *Appendix 1—figure 5A and B*). The respiration and biomass generation pathways remain largely similar to those of *E. coli*, except that the biochemical reactions within the TCA cycle and respiratory chain occur in the mitochondria (see *Appendix 1—figure 5C and D*). This leads to an increased enzyme cost for the respiration pathway due to energy currency exchanges between NADH or FADH$_2$ and ATP in the mitochondria. The coarse-grained models for Group A carbon source utilization in yeast and mammalian cells are shown in *Appendix 1—figure 5E and F*, where $M_3$ represents pyruvate. In yeast and mammalian cells, the stoichiometric coefficients for ATP production (i.e. $\beta_i$) are identical to each other but differ from those of *E. coli* (see *Figure 1B* and *Appendix 1—figure 5C and D*), with $\beta_1 = 5$, $\beta_2 = 1$, $\beta_3 = 5$, $\beta_4 = 7.5$, $\beta_6 = -2.5$, and $\beta_{a1} = 5$ (*Nelson and Cox, 2008*). Hence, the stoichiometric coefficients of ATP production per glucose in each pathway are $\beta_r^{(A)} = 32$ and $\beta_f^{(A)} = 2$, respectively, where $\beta_r^{(A)} = \beta_1 + 2\left(\beta_2 + \beta_3 + \beta_4\right)$ and $\beta_f^{(A)} = \beta_1 + 2\left(\beta_2 + \beta_6\right)$.

The impact of maintenance energy in yeast and tumor cells is significantly higher than that in *E. coli* (*Locasale and Cantley, 2010*). Therefore, *Equation S25* changes to (see *Equation S59*):

$$J_E = r_E \cdot J_{BM} + w_0 \cdot \frac{M_{carbon}}{m_0}, \tag{S166}$$

where $w_0$ is the aforementioned maintenance energy coefficient. Thus, we have (see *Equation S60*):

$$J_E^{(N)} = \eta_E \cdot \lambda + w_0. \tag{S167}$$

To account for the protein cost of energy currency exchanges in the mitochondria, we introduce $\phi_{MT}$ and $\kappa_{MT}$ to represent the proteomic mass fraction of the enzymes and the effective substrate quality of related metabolites in the mitochondria, respectively. Note that the energy currency exchanges between NADH or FADH$_2$ and ATP only occur during respiration, as there is no net NADH or FADH$_2$ generation during fermentation (see *Appendix 1—figure 5C and D*). Combined with *Equation S167*, *Equation S25* changes to:

$$\begin{cases} \phi_A \cdot \kappa_A = \phi_1 \cdot \kappa_1 + \phi_{a1} \cdot \kappa_{a1}, \\ 2\phi_1 \cdot \kappa_1 = \phi_2 \cdot \kappa_2 + \phi_5 \cdot \kappa_5 + \phi_{a2} \cdot \kappa_{a2}, \\ \phi_2 \cdot \kappa_2 = \phi_3 \cdot \kappa_3 + \phi_6 \cdot \kappa_6 + \phi_b \cdot \kappa_b, \\ \phi_5 \cdot \kappa_5 + \phi_4 \cdot \kappa_4 = \phi_3 \cdot \kappa_3 + \phi_d \cdot \kappa_d, \\ \phi_3 \cdot \kappa_3 = \phi_4 \cdot \kappa_4 + \phi_c \cdot \kappa_c, \\ \phi_{a1} \cdot \kappa_{a1} = \eta_{a1} \cdot \lambda, \phi_{a2} \cdot \kappa_{a2} = \eta_{a2} \cdot \lambda, \phi_b \cdot \kappa_b = \eta_b \cdot \lambda, \phi_c \cdot \kappa_c = \eta_c \cdot \lambda, \phi_d \cdot \kappa_d = \eta_d \cdot \lambda, \\ \beta_1 \cdot \phi_1 \cdot \kappa_1 + \beta_2 \cdot \phi_2 \cdot \kappa_2 + \beta_3 \cdot \phi_3 \cdot \kappa_3 + \beta_4 \cdot \phi_4 \cdot \kappa_4 + \beta_6 \cdot \phi_6 \cdot \kappa_6 + \beta_{a1} \cdot \phi_{a1} \cdot \kappa_{a1} = J_E^{(N)}, \\ J_E^{(N)} = \eta_E \cdot \lambda + w_0, \lambda = \phi_R \cdot \kappa_t, J_r^{(N)} = \phi_4 \cdot \kappa_4 = \phi_{MT} \cdot \kappa_{MT}, J_f^{(N)} = \phi_6 \cdot \kappa_6, \\ \phi_R + \phi_A + \phi_1 + \phi_2 + \phi_3 + \phi_4 + \phi_5 + \phi_6 + \phi_{MT} + \phi_{a1} + \phi_{a2} + \phi_b + \phi_c + \phi_d = \phi_{max}. \end{cases} \tag{S168}$$

Here, *Equation S28* still holds, and we have:

$$\begin{cases} J_r^{(E)} + J_f^{(E)} = \varphi \cdot \lambda + w_0, \\ \dfrac{J_r^{(E)}}{\varepsilon_r} + \dfrac{J_f^{(E)}}{\varepsilon_f} = \phi_{max} - \psi \cdot \lambda, \end{cases} \tag{S169}$$

where $J_r^{(E)}$ and $J_f^{(E)}$ follow *Equation S30*, and $\psi$ and $\varphi$ satisfy *Equation S32 and S33*, respectively. The expression for $\varepsilon_f$ follows *Equation S31*. However, the expression for $\varepsilon_r$ differs from that in *Equation S31*. For yeast and mammalian cells, we have:

$$\begin{cases} \varepsilon_r = \dfrac{\beta_r^{(A)}}{1/\kappa_A + 1/\kappa_1 + 2/\kappa_2 + 2/\kappa_3 + 2/\kappa_4 + 2/\kappa_{\mathrm{MT}}}, \\[4mm] \varepsilon_f = \dfrac{\beta_f^{(A)}}{1/\kappa_A + 1/\kappa_1 + 2/\kappa_2 + 2/\kappa_6}. \end{cases} \tag{S170}$$

At the single-cell level, from *Equation S169*, and similar to *Equation S61-S63*, if $\varepsilon_r > \varepsilon_f$, the optimal growth strategy is:

$$\begin{cases} J_f^{(E)} = 0, \\ J_r^{(E)} = \varphi \cdot \lambda + w_0, \end{cases} \qquad \varepsilon_r > \varepsilon_f, \tag{S171}$$

while if $\varepsilon_f > \varepsilon_r$, the optimal growth strategy is:

$$\begin{cases} J_f^{(E)} = \varphi \cdot \lambda + w_0, \\ J_r^{(E)} = 0. \end{cases} \qquad \varepsilon_r < \varepsilon_f. \tag{S172}$$

In both cases, the growth rate $\lambda$ reaches its maximum value for a given nutrient condition with fixed $\kappa_A$:

$$\lambda(\kappa_A) = \begin{cases} \dfrac{\phi_{\max} - w_0/\varepsilon_r(\kappa_A)}{\varphi/\varepsilon_r(\kappa_A) + \psi(\kappa_A)} & \varepsilon_r(\kappa_A) > \varepsilon_f(\kappa_A), \\[4mm] \dfrac{\phi_{\max} - w_0/\varepsilon_f(\kappa_A)}{\varphi/\varepsilon_f(\kappa_A) + \psi(\kappa_A)} & \varepsilon_r(\kappa_A) < \varepsilon_f(\kappa_A). \end{cases} \tag{S173}$$

From *Equation S170*, when $\kappa_A$ is very small such that $\kappa_A \to 0$, it is evident that for yeast and mammalian cells, we still have:

$$\begin{cases} \varepsilon_r(\kappa_A \to 0) \approx \beta_r^{(A)} \cdot \kappa_A, \\ \varepsilon_f(\kappa_A \to 0) \approx \beta_f^{(A)} \cdot \kappa_A. \end{cases} \tag{S174}$$

Thus,

$$\varepsilon_r(\kappa_A \to 0) > \varepsilon_f(\kappa_A \to 0), \tag{S175}$$

since $\beta_r^{(A)} \gg \beta_f^{(A)}$ still holds. Then, as long as $\varepsilon_r(\kappa_A^{\max}) < \varepsilon_f(\kappa_A^{\max})$, there exists a critical switching point for $\kappa_A$ (denoted as $\kappa_A^{(C)}$; see *Equation S41*), below which respiration is more efficient, while above $\kappa_A^{(C)}$, fermentation becomes more efficient in ATP production per proteome. Combined with *Equation S170*, we have:

$$\kappa_A^{(C)} = \frac{\beta_r^{(A)} - \beta_f^{(A)}}{\beta_f^{(A)}\left(1/\kappa_1 + 2/\kappa_2 + 2/\kappa_3 + 2/\kappa_4 + 2/\kappa_{\mathrm{MT}}\right) - \beta_r^{(A)}\left(1/\kappa_1 + 2/\kappa_2 + 2/\kappa_6\right)}. \tag{S176}$$

Accordingly, we obtain the expressions for $\varepsilon_r\left(\kappa_A^{(C)}\right)$, $\varepsilon_f\left(\kappa_A^{(C)}\right)$ and $\lambda_{\mathrm{C}}$ (i.e. $\lambda\left(\kappa_A^{(C)}\right)$):

$$\begin{cases} \varepsilon_r \left( \kappa_A^{(C)} \right) = \varepsilon_f \left( \kappa_A^{(C)} \right) = \dfrac{\beta_r^{(A)} - \beta_f^{(A)}}{2 \left( 1/\kappa_3 + 1/\kappa_4 + 1/\kappa_{MT} - 1/\kappa_6 \right)}, \\[2ex] \lambda_C = \dfrac{\phi_{max} - w_0 / \varepsilon_{rlf} \left( \kappa_A^{(C)} \right)}{\varphi / \varepsilon_{rlf} \left( \kappa_A^{(C)} \right) + \psi \left( \kappa_A^{(C)} \right)}, \end{cases} \tag{S177}$$

Consequently, yeast and tumor cells would preferentially use respiration under starvation conditions (where $\varepsilon_r > \varepsilon_f$), yet switch to aerobic glycolysis when nutrients are abundant (where $\varepsilon_r < \varepsilon_f$) for optimal cell growth. This qualitatively illustrates the Crabtree effect in yeast and the Warburg effect in tumors.

At the cell population level, cell heterogeneity resulting from intrinsic and extrinsic noise causes the turnover numbers (i.e. $k_{cat}$) of enzymes and the critical growth rates at the transition point ($\lambda_C$) to follow distributions, which we assume to be Gaussian (see *Equation S45*, Appendices 3.3 and 8.1). Due to the higher level of heterogeneity observed in tumor cells (*Duraj et al., 2021*; *Shibao et al., 2018*; *Hanahan and Weinberg, 2011*; *Hensley et al., 2016*) and yeast (*Bagamery et al., 2020*) compared to *E. coli*, the extent of noise—and thus the CVs of $k_{cat}$ and $\lambda_C$—in yeast and tumor cells are expected to be larger than those in *E. coli*. The growth rate dependence of the normalized energy fluxes is as follows:

$$\begin{cases} J_f^{(E)} (\lambda) = \dfrac{1}{2} (\varphi \cdot \lambda + w_0) \cdot \left[ \mathrm{erf} \left( \dfrac{\lambda - \mu_{\lambda_C}}{\sqrt{2} \sigma_{\lambda_C}} \right) + 1 \right], \\[2ex] J_r^{(E)} (\lambda) = \dfrac{1}{2} (\varphi \cdot \lambda + w_0) \cdot \left[ 1 - \mathrm{erf} \left( \dfrac{\lambda - \mu_{\lambda_C}}{\sqrt{2} \sigma_{\lambda_C}} \right) \right], \end{cases} \tag{S178}$$

where $\mu_{\lambda_C}$ and $\sigma_{\lambda_C}$ are the mean and standard deviation of $\lambda_C$, respectively, similar to the case of *E. coli*. Therefore, the growth rate dependence of the normalized fluxes is:

$$\begin{cases} J_f^{(N)} (\lambda) = \dfrac{\varphi \cdot \lambda + w_0}{\beta_f^{(A)}} \cdot \left[ \mathrm{erf} \left( \dfrac{\lambda - \mu_{\lambda_C}}{\sqrt{2} \sigma_{\lambda_C}} \right) + 1 \right], \\[2ex] J_r^{(N)} (\lambda) = \dfrac{\varphi \cdot \lambda + w_0}{\beta_r^{(A)}} \cdot \left[ 1 - \mathrm{erf} \left( \dfrac{\lambda - \mu_{\lambda_C}}{\sqrt{2} \sigma_{\lambda_C}} \right) \right]. \end{cases} \tag{S179}$$

Combined with *Equation S160*, *Equation S179* can be compared to experimental results, although in practice, it is difficult to tune the growth rate of tumor cells in vivo in experiments.

Recently, *Shen et al., 2024* reported that in many yeast and tumor cells, the measured proteome efficiencies in respiration at the cell population level are higher than the corresponding proteome efficiencies in fermentation, even though aerobic glycolysis fermentation fluxes still occur. This finding apparently contradicts prevalent explanations (*Basan et al., 2015*; *Chen and Nielsen, 2019*), which assert that overflow metabolism originates from the proteome efficiency in fermentation always being higher than in respiration.

Our model can resolve the puzzle above based on two important features: First, our model predicts that as long as ATP generation per glucose in respiration is higher than in fermentation (i.e. $\beta_r^{(A)} > \beta_f^{(A)}$), which definitely holds true for all organisms, the proteome efficiency in respiration is higher than that in fermentation when the nutrient quality $\kappa_A$ is low (see *Equations S37-S38 and S174-S175*). Second, and importantly, due to cell heterogeneity at the population level, a subset of cells exhibiting greater proteome efficiency in fermentation compared to respiration could exist, even if the proteome efficiency at the cell population level in respiration is higher than in fermentation.

To facilitate comparison between our model and the experiments of *Shen et al., 2024*, we define $\mathrm{Pr}_f$ as the proportion of ATP generated from fermentation, and $\bar{\Delta}$ as the proteome efficiency difference between respiration and fermentation, with

$$\text{Pr}_f \equiv \frac{J_f^{(\text{E})}}{J_f^{(\text{E})} + J_r^{(\text{E})}}, \tag{S180}$$

and

$$\bar{\Delta} \equiv 1/\varepsilon_r - 1/\varepsilon_f. \tag{S181}$$

At the cell population level, $\varepsilon_r$, $\varepsilon_f$, $1/\varepsilon_r$, and $1/\varepsilon_f$ roughly follow Gaussian distributions (see Appendix 8.1 and *Equation S170*), with

$$\begin{cases} \varepsilon_r \sim \mathcal{N}\left(\mu_{\varepsilon_r}, \sigma_{\varepsilon_r}^2\right), \varepsilon_f \sim \mathcal{N}\left(\mu_{\varepsilon_f}, \sigma_{\varepsilon_f}^2\right), \\ 1/\varepsilon_r \sim \mathcal{N}\left(\mu_{1/\varepsilon_r}, \sigma_{1/\varepsilon_r}^2\right), 1/\varepsilon_f \sim \mathcal{N}\left(\mu_{1/\varepsilon_f}, \sigma_{1/\varepsilon_f}^2\right). \end{cases} \tag{S182}$$

Here, $\sigma_{\varepsilon_r}$, $\sigma_{\varepsilon_f}$, $\sigma_{1/\varepsilon_r}$, $\sigma_{1/\varepsilon_f}$, and $\mu_{\varepsilon_r}$, $\mu_{\varepsilon_f}$, $\mu_{1/\varepsilon_r}$, $\mu_{1/\varepsilon_f}$ are the standard deviations and mean values of $\varepsilon_r$, $\varepsilon_f$, $1/\varepsilon_r$ and $1/\varepsilon_f$, respectively. Thus,

$$\begin{cases} \mu_{\varepsilon_r} = \langle \varepsilon_r \rangle, \\ \mu_{\varepsilon_f} = \langle \varepsilon_f \rangle, \end{cases} \tag{S183}$$

where the angle bracket '$\langle \rangle$' represents the average over the cell population, and $\langle \varepsilon_r \rangle$ and $\langle \varepsilon_f \rangle$ are the population-averaged values of $\varepsilon_r$ and $\varepsilon_f$, respectively, which are both measurable in experiments. From the derivations shown in Appendix 8.1, we approximately have

$$\begin{cases} \mu_{1/\varepsilon_r} = 1/\mu_{\varepsilon_r} = 1/\langle \varepsilon_r \rangle, \\ \mu_{1/\varepsilon_f} = 1/\mu_{\varepsilon_f} = 1/\langle \varepsilon_f \rangle. \end{cases} \tag{S184}$$

Here, we use $\chi_{\varepsilon_r}$, $\chi_{\varepsilon_f}$, $\chi_{1/\varepsilon_r}$ and $\chi_{1/\varepsilon_f}$ to represent the CVs of $\varepsilon_r$, $\varepsilon_f$, $1/\varepsilon_r$ and $1/\varepsilon_f$, respectively, with

$$\begin{cases} \chi_{\varepsilon_r} = \sigma_{\varepsilon_r}/\mu_{\varepsilon_r}, \chi_{\varepsilon_f} = \sigma_{\varepsilon_f}/\mu_{\varepsilon_f}, \\ \chi_{1/\varepsilon_r} = \sigma_{1/\varepsilon_r}/\mu_{1/\varepsilon_f}, \chi_{1/\varepsilon_f} = \sigma_{1/\varepsilon_f}/\mu_{1/\varepsilon_f} \end{cases} \tag{S185}$$

Similar to *Equation S151*, the CVs of $1/\varepsilon_r$ and $1/\varepsilon_f$ are roughly equal to those of $\varepsilon_r$ and $\varepsilon_f$, respectively. Thus,

$$\begin{cases} \chi_{1/\varepsilon_r} \approx \chi_{\varepsilon_r}, \\ \chi_{1/\varepsilon_f} \approx \chi_{\varepsilon_f}. \end{cases} \tag{S186}$$

Combining *Equations S181 and S182*, and using the properties of Gaussian distributions, $\bar{\Delta}$ follows a Gaussian distribution:

$$\bar{\Delta} \sim \mathcal{N}\left(\mu_{\bar{\Delta}}, \sigma_{\bar{\Delta}}^2\right), \tag{S187}$$

where $\mu_{\bar{\Delta}}$ and $\sigma_{\bar{\Delta}}$ are the mean and standard deviation of $\bar{\Delta}$, respectively. Evidently, we have

$$\begin{cases} \mu_{\bar{\Delta}} = \mu_{1/\varepsilon_r} - \mu_{1/\varepsilon_f}, \\ \sigma_{\bar{\Delta}}^2 = \sigma_{1/\varepsilon_r}^2 + \sigma_{1/\varepsilon_f}^2. \end{cases} \tag{S188}$$

Then, we proceed to calculate the relation between $\text{Pr}_f$ and $\bar{\Delta}$ using *Equation S187*, and hence we obtain:

$$\text{Pr}_f = \int_0^{+\infty} \frac{1}{\sigma_{\bar{\Delta}}\sqrt{2\pi}} e^{-\frac{1}{2}\left(\frac{x - \mu_{\bar{\Delta}}}{\sigma_{\bar{\Delta}}}\right)^2} dx = \frac{1}{2}\left[\text{erf}\left(\frac{\mu_{\bar{\Delta}}}{\sqrt{2}\sigma_{\bar{\Delta}}}\right) + 1\right]. \tag{S189}$$

Combining *Equations S180, S183-S185, and S188-S189*, we have:

$$\frac{J_f^{(E)}}{J_f^{(E)} + J_r^{(E)}} = \frac{1}{2}\left[\mathrm{erf}\left(\frac{1 - \langle\varepsilon_r\rangle / \langle\varepsilon_f\rangle}{\sqrt{2}\cdot\sqrt{\chi_{\varepsilon_r}^2 + \chi_{\varepsilon_f}^2\cdot\left(\langle\varepsilon_r\rangle / \langle\varepsilon_f\rangle\right)^2}}\right) + 1\right].$$  (S190)

Note that the normalized energy fluxes $J_r^{(E)}$ and $J_f^{(E)}$ are proportional to the measured ATP fluxes generated in respiration and fermentation, respectively. Hence, *Equation S190* can be directly compared to experimental data. For yeast and tumor cells, due to a higher level of heterogeneity, the CVs of $\varepsilon_r$ and $\varepsilon_f$, i.e., $\chi_{\varepsilon_r}$ and $\chi_{\varepsilon_f}$, could be significantly higher than the corresponding values in *E. coli*, though their exact values are unknown. Consequently, we plot theoretical results with the values of $\chi_{\varepsilon_r}$ and $\chi_{\varepsilon_f}$ chosen as 0.25, 0.40, and 0.58 to compare with the experimental data for yeast and in vivo mouse tumors (*Bartman et al., 2023*; *Shen et al., 2024*). In *Figure 5A, B*, we observe that the theoretical results using $\chi_{\varepsilon_r} = \chi_{\varepsilon_f} = 0.58$ agree well with the experimental data (*Bartman et al., 2023*; *Shen et al., 2024*), both on a log scale and linear scale. This demonstrates that our model has the potential to quantitatively illustrate the Crabtree effect in yeast and the Warburg effect in tumors.

## Appendix 11

### Notes on the application of reference data

#### Data calibration

Throughout our manuscript, we use experimental data from the original references, except for two calibrations. The first calibration is noted in the footnote of *Appendix 1—table 2*. With this calibration, the $J_{\text{acetate}}^{(\text{M})}$-$\lambda$ data for *E. coli* (*Basan et al., 2015*) in *Appendix 1—table 2* align with the curve shown in *Figure 1C*, which includes experimental data for *E. coli* from other sources. The second calibration applies to the data shown in *Figures 3F and 1C* (chemostat data for *E. coli*). The unit in the original reference (*Holms, 1996*) is mmol/(dry mass)g/h. To convert this to the unit mM/$OD_{600}$/h, used in our text, the conversion factor should be 0.18. Here, we deduce that only 60% of the measured dry biomass in centrifuged material is effective when calibrating with other experimental results. Therefore, there is a calibration factor of 0.6, and the conversion factor changes to 0.3.

#### Data from the inducible strains

Some of the experimental data in the original references (*Basan et al., 2015*; *Hui et al., 2015*) were obtained using *E. coli* strains with titratable systems (e.g. titratable ptsG, LacY). The $J_{\text{acetate}}^{(\text{M})}$-$\lambda$ relation of these inducible strains generally aligns with the same curve as that of wild-type *E. coli* (*Figure 1C*). Since evolutionary treatment was not applied to the inducible strains, we approximate titration perturbation as a technique that mimics culturing the strains in a less efficient Group A carbon source.

#### Experimental data sources

The batch culture data for *E. coli* shown in *Figure 1C* (labeled as minimum/rich media or inducible strains) and *Appendix 1—figure 2C* were taken from the source data of the reference's figure 1 (*Basan et al., 2015*). The chemostat data for *E. coli* shown in *Figure 1C* were taken from the reference's table 7 (*Holms, 1996*). The data for *E. coli* shown in *Figure 1D* were taken from the reference's extended data figure 3a (*Basan et al., 2015*), with the calibration specified in the footnote to *Appendix 1—table 2*.

The data for *E. coli* shown in *Figure 2A* were adopted from the reference's extended data figure 4a–b (*Basan et al., 2015*). The data for *E. coli* shown in *Figure 2B* were taken from the source data of the reference's figure 2a (*Basan et al., 2015*). The data for *E. coli* shown in *Figure 2C* were taken from the source data of the reference's figure 3a (*Basan et al., 2015*). The data for *E. coli* shown in *Figure 3A–B* were taken from the source data of the reference's figure 3d (*Basan et al., 2015*).

The data for *E. coli* shown in *Figure 3C–D* and *Appendix 1—figure 2D-E* were taken from the source data of the reference's figure 3c (*Basan et al., 2015*). The data for *E. coli* shown in *Figure 3F* were taken from the reference's table 7 (*Holms, 1996*), with a calibration factor specified in the above paragraph ('Data calibration').

The data for *E. coli* shown in *Figure 4A–B* and *Appendix 1—figure 3A-D* were taken from the reference's table S2 with the label 'C-lim' (*Hui et al., 2015*). We excluded the reference's data with $\lambda$=0.45205 h$^{-1}$ as there are other unconsidered factors involved during slow growth (*Dai et al., 2017*) (for $\lambda<0.5$ h$^{-1}$), and we suspect that unknown calibration factors may exist. The data for *E. coli* shown in *Figure 4C–D* and *Appendix 1—figure 3E-N* were adopted from the reference's extended data figure 6-7 (*Basan et al., 2015*).

The batch culture data for yeast shown in *Figure 5* were derived from the source data of the reference's extended figure 4c-d (*Shen et al., 2024*). The chemostat data for yeast shown in *Figure 5* were derived from the source data of the reference's figure 3d-e (*Shen et al., 2024*), where glucose is the limiting nutrient. We excluded the reference's data for *I. orientalis* under condition C2, where the ATP flux was abnormally small. The mouse tumor in vivo data shown in *Figure 5* were derived from the source data of the reference's figure 4e-g (*Shen et al., 2024*), which were originally reported by *Bartman et al., 2023*, the same research group as *Shen et al., 2024*. We did not include the cancer cell line data shown in figure 4a-c of *Shen et al., 2024* since it appears that the proteomic data and flux data were obtained from two different references with inconsistent culturing conditions.

The gene names of *E. coli* depicted in *Appendix 1—figure 1B* were identified using the KEGG database. The data for *E. coli* shown in *Appendix 1—figure 2G* were drawn from *Appendix 1—table 1*, which includes the original references themselves. The flux data for *E. coli* presented in

*Appendix 1—table 2* were obtained from the reference's extended data figure 3a (*Basan et al., 2015*), with the calibration specified in the footnote. The proteome data for *E. coli* shown in *Appendix 1—table 2* were taken from the reference's supplementary Table N5 (*Basan et al., 2015*).

