## [Editor Report · eLife Assessment]

This **valuable** study tackles the well-established overflow metabolism issue by applying a coarse-grained metabolic flux model to predict how individual cells execute various energy strategies, such as respiration versus fermentation. The model's population average is **convincing** enough to align with experimental observations on overflow metabolism. The potential source of metabolic or proteomic heterogeneity of individual cells remains an open question to be studied. How individual cells adjust their metabolic strategies also requires future study of the underlying mechanisms. Overall, this work provides a key aspect on cell-to-cell variability on general metabolic response.

---

## [Referee Report · Reviewer #1 (Public review)]

Summary:

Cell metabolism exhibits a well-known behavior in fast-growing cells, which employ seemingly wasteful fermentation to generate energy even in the presence of sufficient environmental oxygen. This phenomenon is known as Overflow Metabolism or the Warburg effect in cancer. It is present in a wide range of organisms, from bacteria and fungi to mammalian cells.

In this work, starting with a metabolic network for *Escherichia coli* based on sets of carbon sources, and using a corresponding coarse-grained model, the author applies some well-based approximations from the literature and algebraic manipulations. These are used to successfully explain the origins of Overflow Metabolism, both qualitatively and quantitatively, by comparing the results with *E. coli* experimental data.

By modeling the proteome energy efficiencies for respiration and fermentation, the study shows that these parameters are dependent on the carbon source quality constants K_i (p.115 and 116). It is demonstrated that as the environment becomes richer, the optimal solution for proteome energy efficiency shifts from respiration to fermentation. This shift occurs at a critical parameter value K_A(C).

This counterintuitive result qualitatively explains Overflow Metabolism.

Quantitative agreement is achieved through the analysis of the heterogeneity of the metabolic status within a cell population. By introducing heterogeneity, the critical growth rate is assumed to follow a Gaussian distribution over the cell population, resulting in accordance with experimental data for *E. coli*. Overflow metabolism is explained by considering optimal protein allocation and cell heterogeneity.

The obtained model is extensively tested through perturbations: (1) Introduction of overexpression of useless proteins; (2) Studying energy dissipation; (3) Analysis of the impact of translation inhibition with different sub-lethal doses of chloramphenicol on *Escherichia coli*; (4) Alteration of nutrient categories of carbon sources using pyruvate. All model perturbations results are corroborated by *E. coli* experimental results.

Strengths:

In this work, the author effectively uses modeling techniques typical of Physics to address complex problems in Biology, demonstrating the potential of interdisciplinary approaches to yield novel insights. The use of *Escherichia coli* as a model organism ensures that the assumptions and approximations are well-supported in existing literature. The model is convincingly constructed and aligns well with experimental data, lending credibility to the findings. In this version, the extension of results from bacteria to yeast and cancer is substantiated by a literature base, suggesting that these findings may have broad implications for understanding diverse biological systems.

Weaknesses:

The author explores the generalization of their results from bacteria to cancer cells and yeast, adapting the metabolic network and coarse-grained model accordingly. In the previous version this generalization was not completely supported by references and data from the literature. This drawback, however, has been treated in this current version, where the authors discuss in much more detail and give references supporting this generalization.

Comments on revisions:

I have no specific comments for the authors. My previous comments were all addressed, discussed and explained.

---

## [Author Response]

The following is the authors’ response to the previous reviews

**Reviewer #1 (Public Review):**
Summary:Cell metabolism exhibits a well-known behavior in fast-growing cells, which employ seemingly wasteful fermentation to generate energy even in the presence of sufficient environmental oxygen. This phenomenon is known as Overflow Metabolism or the Warburg effect in cancer. It is present in a wide range of organisms, from bacteria and fungi to mammalian cells.In this work, starting with a metabolic network for *Escherichia coli* based on sets of carbon sources, and using a corresponding coarse-grained model, the author applies some well-based approximations from the literature and algebraic manipulations. These are used to successfully explain the origins of Overflow Metabolism, both qualitatively and quantitatively, by comparing the results with *E. coli* experimental data.By modeling the proteome energy efficiencies for respiration and fermentation, the study shows that these parameters are dependent on the carbon source quality constants K_i (p.115 and 116). It is demonstrated that as the environment becomes richer, the optimal solution for proteome energy efficiency shifts from respiration to fermentation. This shift occurs at a critical parameter value K_A(C).This counter intuitive results qualitatively explains Overflow Metabolism.Quantitative agreement is achieved through the analysis of the heterogeneity of the metabolic status within a cell population. By introducing heterogeneity, the critical growth rate is assumed to follow a Gaussian distribution over the cell population, resulting in accordance with experimental data for *E. coli*. Overflow metabolism is explained by considering optimal protein allocation and cell heterogeneity.The obtained model is extensively tested through perturbations: (1) Introduction of overexpression of useless proteins; (2) Studying energy dissipation; (3) Analysis of the impact of translation inhibition with different sub-lethal doses of chloramphenicol on *Escherichia coli*; (4) Alteration of nutrient categories of carbon sources using pyruvate. All model perturbations results are corroborated by *E. coli* experimental results.Strengths:In this work, the author effectively uses modeling techniques typical of Physics to address complex problems in Biology, demonstrating the potential of interdisciplinary approaches to yield novel insights. The use of *Escherichia coli* as a model organism ensures that the assumptions and approximations are well-supported in existing literature. The model is convincingly constructed and aligns well with experimental data, lending credibility to the findings. In this version, the extension of results from bacteria to yeast and cancer is substantiated by a literature base, suggesting that these findings may have broad implications for understanding diverse biological systems.

We appreciate the reviewer’s exceptionally positive comments. The manuscript has been significantly improved thanks to the reviewer’s insightful suggestions.

Weaknesses:The author explores the generalization of their results from bacteria to cancer cells and yeast, adapting the metabolic network and coarse-grained model accordingly. In previous version this generalization was not completely supported by references and data from the literature. This drawback, however, has been treated in this current version, where the authors discuss in much more detail and give references supporting this generalization.

We appreciate the reviewer’s recognition of our revisions and the insightful suggestions provided in the previous round, which have greatly strengthened our manuscript.

**Reviewer #2 (Public Review):**
In this version of manuscript, the author clarified many details and rewrote some sections. This substantially improved the readability of the paper. I also recognized that the author spent substantial efforts in the Appendix to answer the potential questions.

We thank the reviewer for the positive comments and the suggestions to improve our manuscript.

Unfortunately, I am not currently convinced by the theory proposed in this paper. In the next section, I will first recap the logic of the author and explain why I am not convinced. Although the theory fits many experimental results, other theories on overflow metabolism are also supported by experiments. Hence, I do not think based on experimental data we could rule in or rule out different theories.

We thank the reviewer for both the critical and constructive comments.

Regarding the comments on the comparison between theoretical and experimental results, we would like to first emphasize that no prior theory has resolved the conflict arising from the proteome efficiencies measured in *E. coli* and eukaryotic cells. Specifically, prevalent explanations (Basan et al., *Nature* 528, 99–104 (2015); Chen and Nielsen, *PNAS* 116, 17592–17597 (2019)) hold that overflow metabolism results from proteome efficiency in fermentation consistently being higher than that in respiration. While it was observed in *E. coli* that proteome efficiency in fermentation exceeds that in respiration when cells were cultured in lactose at saturated concentrations (Basan et al., *Nature* 528, 99-104 (2015)), more recent findings (Shen et al., *Nature Chemical Biology* 20, 1123–1132 (2024)) show that the measured proteome efficiency in respiration is actually higher than in fermentation for many yeast and cancer cells, despite the presence of aerobic glycolytic fermentation flux. To the best of our knowledge, no prior theory has explained these contradictory experimental results. Notably, our theory resolves this conflict and quantitatively explains both sets of experimental observations (Basan et al., *Nature* 528, 99-104 (2015); Shen et al., *Nature Chemical Biology* 20, 1123–1132 (2024)) by incorporating cell heterogeneity and optimizing cell growth rate through protein allocation.

Furthermore, rather than merely fitting the experimental results, as explained in Appendices 6.2, 8.1-8.2 and summarized in Appendix-tables 1-3, nearly all model parameters important for our theoretical predictions for *E. coli* were derived from in vivo and in vitro biochemical data reported in the experimental literature. For comparisons between model predictions and experimental results for yeast and cancer cells (Shen et al., *Nature Chemical Biology* 20, 1123–1132 (2024)), we intentionally derived Eq. 6 to ensure an unbiased comparison.

Finally, in response to the reviewer’s suggestion, we have revised the expressions in our manuscript to present the differences between our theory and previous theories in a more modest style.

Recap: To explain the origin of overflow metabolism, the author uses the following logic:(1) There is a substantial variability of single-cell growth rate(2) The flux (J_r^E) and (J_f^E) are coupled with growth rate by Eq. 3(3) Since growth rate varies from cells to cells, flux (J_r^E) and (J_f^E) also varies (4) The variabilities of above fluxes in above create threshold-analog relation, and hence overflow metabolism.

We thank the reviewer for the clear summary. We apologize for not explaining some points clearly enough in the previous version of our manuscript, which may have led to misunderstandings. We have now revised the relevant content in the manuscript to clarify our reasoning. Specifically, we have applied the following logic in our explanation:

(a) The solution for the optimal growth strategy of a cell under a given nutrient condition is a binary choice between respiration and fermentation, driven by comparing their proteome efficiencies (ε*r* and ε*f*).

(b) Under nutrient-poor conditions, the nutrient quality (*κA*) is low, resulting in the proteome efficiency of respiration being higher than that of fermentation (i.e., ε*r* > ε*f*), so the cell exclusively uses respiration.

(c) In rich media (with high *κA*), the proteome efficiency of fermentation increases more rapidly and surpasses that of respiration (i.e., ε*f* > ε*r*), hence the cell switches to fermentation.

(d) Heterogeneity is introduced: variability in the *κ*_cat_ of catalytic enzymes from cell to cell. This leads to heterogeneity (variability) in ε*r* and ε*f* within a population of cells under the same nutrient condition.

(e) The critical value of nutrient quality for the switching point (\begin{document}$\kappa_{A}^{(C)}$\end{document}, where ε*r* = ε*f*) changes from a single point to a distribution due to cell heterogeneity. This results in a distribution of the critical growth rate λ_C_ (defined as \begin{document}$\lambda_{\mathrm{C}} \equiv \lambda\left(\kappa_{A}^{(\mathrm{C})}\right)$\end{document}) within the cell population.

(f) The change in culturing conditions (with a highly diverse range of *κA*) and heterogeneity in the critical growth rate λ_C_ (a distribution of values) result in the threshold-analog relation of overflow metabolism at the cell population level.

Steps (a)-(c) were applied to qualitatively explain the origin of overflow metabolism, while steps (d)-(f) were further used to quantitatively explain the threshold-analog relation observed in the data on overflow metabolism.

Regarding the reviewer’s recap, which seems to have involved some misunderstandings, we first emphasize that the major change in cell growth rate for the threshold-analog relation of overflow metabolism—particularly as it pertains to logic steps (1), (3) and (4)—is driven by the highly varied range of nutrient quality (*κA*) in the culturing conditions, rather than by heterogeneity between cells. For the batch culture data, the nutrient type of the carbon source differs significantly (e.g., Fig.1 in Basan et al., *Nature* 528, 99-104 (2015), wild-type strains). In contrast, for the chemostat data, the concentration of the carbon source varies greatly due to the highly varied dilution rate (e.g., Table 7 in Holms, *FEMS Microbiology Reviews* 19, 85-116 (1996)). Both of these factors related to nutrient conditions are the major causes of the changes in cell growth rate in the threshold-analog relation.

Second, Eq. 3, as mentioned in logic step (2), represents a constraint between the fluxes (\begin{document}$J_{r}^{(\mathrm{E})}$\end{document} and \begin{document}$J_{f}^{(\mathrm{E})}$\end{document}) and the growth rate (λ) for a single nutrient condition (with a given value of *κA* ideally) rather than for varied nutrient conditions. For a single cell in each nutrient condition, the optimal growth strategy is binary, between respiration and fermentation.

Finally, for the threshold-analog relation of overflow metabolism, the switch from respiration to fermentation is caused by the increased nutrient quality in the culturing conditions, rather than by cell heterogeneity as indicated in logic step (4). Upon nutrient upshifts, the proteome efficiency of fermentation surpasses that of respiration, causing the optimal growth strategy for the cell to switch from respiration to fermentation. The role of cell heterogeneity is to transform the growth rate-dependent fermentation flux in overflow metabolism from a digital response to a threshold-analog relation under varying nutrient conditions.

My opinion:The logic step (2) and (3) have caveats. The variability of growth rate has large components of cellular noise and external noise. Therefore, variability of growth rate is far from 100% correlated with variability of flux (J_r^E) and (J_f^E) at the single-cell level. Single-cell growth rate is a complex, multivariate functional, including (Jr^E) and (J_f^E) but also many other variables. My feeling is the correlation could be too low to support the logic here.One example: ribosomal concentration is known to be an important factor of growth rate in bulk culture. However, the "growth law" from bulk culture cannot directly translate into the growth law at single-cell level [Ref1,2]. This is likely due to other factors (such as cell aging, other muti-stability of cellular states) are involved.Therefore, I think using Eq.3 to invert the distribution of growth rate into the distribution of (Jr^E) and (J_f^E) is inapplicable, due to the potentially low correlation at single-cell level. It may show partial correlations, but may not be strong enough to support the claim and create fermentation at macroscopic scale.Overall, if we track the logic flow, this theory implies overflow metabolism is originated from variability of k_cat of catalytic enzymes from cells to cells. That is, the author proposed that overflow metabolism happens macroscopically as if it is some "aberrant activation of fermentation pathway" at the single-cell level, due to some unknown partially correlation from growth rate variability.

We thank the reviewer for raising these questions and for the insights. We apologize for any lack of clarity in the previous version of our manuscript that may have caused misunderstandings. We have revised the manuscript to address all points, and below are our responses to the questions, some of which seem to involve misunderstandings.

First, in our theory, the qualitative behavior of overflow metabolism—where cells use respiration under nutrient-poor conditions (low growth rate) and fermentation in rich media (high growth rate)—does not arise from variability between cells, as the reviewer seems to have interpreted. Instead, it originates from growth optimization through optimal protein allocation under significantly different nutrient conditions. Specifically, the proteome efficiency of fermentation is lower than that of respiration (i.e. ε*f* < ε*r*) under nutrient-poor conditions, making respiration the optimal strategy in this case. However, in rich media, the proteome efficiency of fermentation surpasses that of respiration (i.e. ε*f* < ε*r*), leading the cell to switch to fermentation for growth optimization. To implement the optimal strategy, as clarified in the revised manuscript and discussed in Appendix 2.4, a cell should sense and compare the proteome efficiencies between respiration and fermentation, choosing the pathway with the higher efficiency, rather than sensing the growth rate, which can fluctuate due to stochasticity. Regarding the role of cell heterogeneity in overflow metabolism, as discussed in our previous response, it is twofold: first, it quantitatively illustrates the threshold-analog response of growth rate-dependent fermentation flux, which would otherwise be a digital response without heterogeneity during growth optimization; second, it enables us to resolve the paradox in proteome efficiencies observed in *E. coli* and eukaryotic cells, as raised by Shen et al. (Shen et al., *Nature Chemical Biology* 20, 1123–1132 (2024)).

Second, regarding logic step (2) in the recap, the reviewer thought we had coupled the growth rate (λ) with the respiration and fermentation fluxes (\begin{document}$J_{r}^{(\mathrm{E})}$\end{document} and \begin{document}$J_{f}^{(\mathrm{E})}$\end{document}) through Eq. 3, and used Eq. 3 to invert the distribution of growth rate into the distribution of respiration and fermentation fluxes. We need to clarify that Eq. 3 represents the constraint between the fluxes and the growth rate under a single nutrient condition, rather than describing the relation between growth rate and the fluxes (\begin{document}$J_{r}^{(\mathrm{E})}$\end{document} and *κA*), without considering optimal protein allocation, the cell growth rate varies with the fluxes according to Eq.3 by adjusting the proteome allocation between respiration and fermentation (*ϕr* and \begin{document}$J_{f}^{(\mathrm{E})}$\end{document}) under varied nutrient conditions. In a given nutrient condition (with a fixed value of *ϕf*). However, once growth optimization is applied, the optimal protein allocation strategy for a cell is limited to either pure respiration (with *ϕf* =0 and *ϕr* =0 and *κA*). Furthermore, under varying nutrient conditions (with different values of \begin{document}$J_{f}^{(\mathrm{E})}=\varepsilon_{f} \cdot \phi_{f}=0$\end{document}) or pure fermentation (with *κA*), both proteome efficiencies of respiration and fermentation (*εr* and (*εf*) change with nutrient quality \begin{document}$J_{r}^{(\mathrm{E})}=\varepsilon_{r} \cdot \phi_{r}=0$\end{document}), depending on the nutrient condition or the value of *κA* (see Eq. 4). Thus, Eq. 3 does not describe the relation between growth rate (λ) and the fluxes (\begin{document}$J_{r}^{(\mathrm{E})}$\end{document} and \begin{document}$J_{f}^{(\mathrm{E})}$\end{document}) under nutrient variations.

Thirdly, regarding reviewer’s concerns on logic step (3) in the recap, as well as the example where ribosome concentration does not correlate well with cell growth rate at the single-cell level, we fully agree with reviewer that, due to factors such as stochasticity and cell cycle status, the growth rate fluctuates constantly for each cell. Consequently, it would not be fully correlated with cell parameters such as ribosome concentration or respiration/fermentation flux. We apologize for our oversight in not discussing suboptimal growth conditions in the previous version of the manuscript. In response, we have added a paragraph to the discussion section and a new Appendix 2.4, titled “Dependence of the model on optimization principles,” to address these issues in detail. Specifically, recent experimental studies (Dai et al., *Nature microbiology* 2, 16231 (2017); Li et al., *Nature microbiology* 3, 939–947 (2018)) show that the inactive portion of ribosomes (i.e. ribosomes not bound to mRNAs) can vary under different culturing conditions. The reviewer also pointed out that ribosome concentration does not correlate well with cell growth rate at single-cell level. In this regard, we have cited Pavlou et al. (Pavlou et al., *Nature Communications* 16, 285 (2025)) instead of the references provided by the reviewer (Ref1 and Ref2), with our rationale outlined in the final section of the author response. These findings (Dai et al, (2017); Li et al., (2018); Pavlou et al., (2025)) suggest that ribosome allocation may be suboptimal under many culturing conditions, likely as cells prepare for potential environmental changes (Li et al., *Nature microbiology* 3, 939–947 (2018)). However, since our model's predictions regarding the binary choice between respiration and fermentation are based solely on comparing proteome efficiency between these two pathways, the optimal growth principle in our model can be relaxed. Specifically, efficient protein allocation is required only for enzymes rather than ribosomes, allowing our model to remain applicable under suboptimal growth conditions. Furthermore, protein allocation via the ribosome occurs at the single-cell level rather than at the population level. The strong linear correlation between ribosomal concentration and growth rate at the population level under nutrient variations suggests that each cell optimizes its protein allocation individually. Therefore, the principle of growth optimization still applies to individual cells, although factors like stochasticity, nutrient variation preparations, and differences in cell cycle stages may complicate this relationship, resulting in only a rough linear correlation between ribosome concentration and growth rate at the single-cell level (with with *R2* = 0.64 reported in Pavlou et al., (2025)).

Lastly, regarding the reviewer concerns about the heterogeneity of fermentation and respiration at macroscopic scale, we first clarify in the second paragraph of this response that the primary driving force for cells to switch from respiration to fermentation in the context of overflow metabolism is the increased nutrient quality under varying culturing conditions, which causes the proteome efficiency of fermentation to surpass that of respiration. Under nutrient-poor conditions, our model predicts that all cells use respiration, and therefore no heterogeneity for the phenotype of respiration and fermentation arises in these conditions. However, in a richer medium, particularly one that does not provide optimal conditions but allows for an intermediate growth rate, our model predicts that some cells opt for fermentation while others continue with respiration due to cell heterogeneity (with ε*f* > ε*r* for some cells engaging in fermentation and ε*r* > ε*f* for the other cells engaging in respiration within the same medium). Both of these predictions have been validated in isogenic singlecell experiments with *E. coli* (Nikolic et al., *BMC Microbiology* 13, 258 (2013)) and *S. cerevisiae* (Bagamery et al., *Current Biology* 30, 4563–4578 (2020)). The single-cell experiments by Nikolic et al. with *E. coli* in a rich medium of intermediate growth rate clearly show a bimodal distribution in the expression of genes related to overflow metabolism (see Fig. 5 in Nikolic et al., *BMC Microbiology* 13, 258 (2013)), where one subpopulation suggests purely fermentation, while the other suggests purely respiration. In contrast, in a medium with lower nutrient concentration (and consequently lower nutrient quality), only the respirative population exists (see Fig. 5 in Nikolic et al., *BMC Microbiology* 13, 258 (2013)). These experimental results from *E. coli* (Nikolic et al., *BMC Microbiology* 13, 258 (2013)) are fully consistent with our model predictions. Similarly, the single-cell experiments with *S. cerevisiae* by Bagamery et al. clearly identified two subpopulations of cells with respect to fermentation and respiration in a rich medium, which also align well with our model predictions regarding heterogeneity in fermentation and respiration within a cell population in the same medium.

Compared with other theories, this theory does not involve any regulatory mechanism and can be regarded as a "neutral theory". I am looking forward to seeing single cell experiments in the future to provide evidences about this theory.

We thank the reviewer for raising these questions and for the valuable insights. Regarding the regulatory mechanism, we have now added a paragraph in the discussion section of our manuscript and Appendix 2.4 to address this point. Specifically, our model predicts that a cell can implement the optimal strategy by directly sensing and comparing the proteome efficiencies of respiration and fermentation, choosing the pathway with the higher efficiency. At the gene regulatory level, a growing body of evidence suggests that the cAMP-CRP system plays an important role in sensing and executing the optimal strategy between respiration and fermentation (Basan et al., *Nature* 528, 99-104 (2015); Towbin et al., *Nature Communications* 8, 14123 (2017); Valgepea et al., *BMC Systems Biology* 4, 166 (2010); Wehrens et al., *Cell Reports* 42, 113284 (2023)). However, it has also been suggested that the cAMP-CRP system alone is insufficient, and additional regulators may need to be identified to fully elucidate this mechanism (Basan et al., *Nature* 528, 99-104 (2015); Valgepea et al., *BMC Systems Biology* 4, 166 (2010)).

Regarding the single-cell experiments that provide evidence for this theory, we have shown in the previous paragraphs of this response that the heterogeneity between respiration and fermentation, as predicted by our model for isogenic cells within the same culturing condition, has been fully validated by single-cell experiments with *E. coli* (Fig. 5 from Nikolic et al., *BMC Microbiology* 13, 258 (2013)) and *S. cerevisiae* (Fig. 1 and the graphical abstract from Bagamery et al., *Current Biology* 30, 4563–4578 (2020)). We have now revised the discussion section of our manuscript to make this point clearer.

[Ref1] https://www.biorxiv.org/content/10.1101/2024.04.19.590370v2[Ref2] https://www.biorxiv.org/content/10.1101/2024.10.08.617237v2

We thank the reviewer for providing insightful references. Regarding the two specific references, Ref1 directly addresses the deviation in the linear relationship between growth rate and ribosome concentration (“growth law”) at the single-cell level. However, since the authors of Ref1 determined the rRNA abundance in each cell by aligning sequencing reads to the genome, this method inevitably introduces a substantial amount of measurement noise. As a result, we chose not to cite or discuss this preprint in our manuscript. Ref2 appears to pertain to a different topic, which we suspect may be a copy/paste error. Based on the reviewer’s description and the references in Ref1, we believe the correct Ref2 should be Pavlou et al., *Nature Communications* 16, 285 (2025) (with the biorxiv preprint link: https://www.biorxiv.org/content/10.1101/2024.04.26.591328v1). In this reference, it is stated that the relationship between ribosome concentration and growth rate only roughly aligns with the “growth law” at the single-cell level (with *R2* = 0.64), exhibiting a certain degree of deviation. We have now cited and incorporated the findings of Pavlou et al. (Pavlou et al., *Nature Communications* 16, 285 (2025)) in both the discussion section of our manuscript and Appendix 2.4. Overall, we agree with Pavlou et al.’s experimental results, which suggest that ribosome concentration does not exhibit a strong linear correlation with cell growth rate at the single-cell level. However, we remain somewhat uncertain about the extent of this deviation, as Pavlou et al.’s experimental setup involved alternating nutrients between acetate and glucose, and the lapse of five generations may not have been long enough for the growth to be considered balanced. Furthermore, as observed in Supplementary Movie 1 of Pavlou et al., some of the experimental cells appeared to experience growth limitations due to squeezing pressure from the pipe wall of the mother machine, which could further increase the deviation from the “growth law” at the single-cell level.

**Recommendations for the authors:**

**Reviewer #1 (Recommendations for the authors):**
I have no specific comments for the authors related to this last version of the paper. I believe the authors have properly improved the previous version of the manuscript.

Response: We thank the reviewer for the highly positive comments and for recognizing the improvements made in the revised version of our manuscript.